# Novel automated inversion algorithm for temperature reconstruction using gas isotopes from ice cores

Michael Döring[1,2*] and Markus Leuenberger[1,2]

[1]Climate and Environmental Physics, University of Bern, Switzerland
[2]Oeschger Centre for Climate Change Research (OCCR), Bern, Switzerland

*Correspondence to*: Michael Döring (doering@climate.unibe.ch)

**Keywords:** temperature reconstruction, ice core, nitrogen isotope, argon isotope, inverse model, firn model, accumulation rate

**Abstract.** Greenland past temperature history can be reconstructed by forcing the output of a firn-densification and heat-diffusion model to fit multiple gas-isotope data ($\delta^{15}N$ or $\delta^{40}Ar$ or $\delta^{15}N_{excess}$) extracted from ancient air in Greenland ice cores using published accumulation-rate (Acc) data-sets. We present here a novel methodology to solve this inverse problem, by designing a fully-automated algorithm. To demonstrate the performance of this novel approach, we begin by intentionally constructing synthetic temperature-histories and associated $\delta^{15}N$ datasets, mimicking real Holocene data that we use as "true

values" (targets) to be compared to the output of the algorithm. This allows us to quantify uncertainties originating from the algorithm itself. The presented approach is completely automated and therefore minimizes the "subjective" impact of manual parameter-tuning, leading to reproducible temperature-estimates. In contrast to many other ice-core-based temperature-reconstruction methods, the presented approach is completely independent from ice-core stable-water-isotopes, providing the opportunity to validate water-isotope-based reconstructions or reconstructions where water isotopes are used together with

$\delta^{15}N$ or $\delta^{40}Ar$. We solve the inverse problem T($\delta^{15}N$, Acc) by using a combination of a Monte-Carlo-based iterative approach and the analysis of remaining mismatches between modelled and target data, based on cubic-spline-filtering of random numbers and the laboratory-determined temperature-sensitivity for nitrogen isotopes. Additionally, the presented reconstruction approach was tested by fitting measured $\delta^{40}Ar$ and $\delta^{15}N_{excess}$ data, which leads as well to a robust agreement between modelled and measured data. The obtained final mismatches follow a symmetric standard-distribution-function. For

the study on synthetic data, 95 % of the mismatches compared to the synthetic target-data are in an envelope between 3.0 permeg to 6.3 permeg for $\delta^{15}N$ and 0.23 K to 0.51 K for temperature ($2\sigma$, respectively). In addition to Holocene temperature-reconstructions, the fitting approach can also be used for glacial temperature-reconstructions. This is shown by fitting of NGRIP $\delta^{15}N$ data for two Dansgaard-Oeschger events using the presented approach, leading to results comparable to other studies.

# 1 Introduction

Holocene climate variability is of key interest to our society, since it represents a time of moderate natural variations prior to anthropogenic disturbance, often referred to as a baseline for today's increasing greenhouse effect driven by mankind. Yet, high-resolution studies are still very sparse and therefore limit the investigation of decadal and even centennial climate variations over the course of the Holocene. One of the first studies about changes in the Holocene climate was conducted in the early 1970s by Denton and Karle´n (1973). The authors investigated rapid changes in glacier extents around the globe potentially resulting from variations of Holocene climatic conditions. Mayewski et al. (2004) used these data as the base of a multiproxy study identifying rapid climate changes (so called RCCs) globally distributed over the whole Holocene time period. Although not all proxy data are showing an equal behaviour in timing and extent during the quasi-periodic RCC patterns, the authors found evidence for a highly variable Holocene climate controlled by multiple mechanisms, which significantly affects ecosystems (Beaulieu et al., 2017; Crausbay et al., 2017; Pál et al., 2016) and human societies (Holmgren et al., 2016; Lespez et al., 2016). Precise high-resolution temperature-estimates can contribute significantly to the understanding of these mechanisms. Ice-core proxy-data offer multiple paths for reconstructing past climate and temperature variability. The studies of Cuffey et al. (1995), Cuffey and Clow (1997) and Dahl-Jensen et al. (1998) demonstrate the usefulness of inverting the measured borehole-temperature profile for surface-temperature-history estimates for the investigated drilling site using a coupled heat- and ice-flow model. Because of smoothing effects due to heat-diffusion within an ice sheet, this method is unable to resolve fast temperature oscillations and leads to a rapid reduction of the time resolution towards the past. Another approach to reconstruct past temperature is based on the calibration of water-stable-isotopes of oxygen and hydrogen ($\delta^{18}O_{ice}$, $\delta D_{ice}$) from ice-core water-samples assuming a constant (and mostly linear) relationship between temperature and isotopic composition due to fractionation effects during ocean evaporation, cloud formation and snow and ice precipitation (Johnsen et al., 2001; Stuiver et al., 1995). This method provides a rather robust tool for reconstructing past temperature for times where large temperature excursions occur when an adequate relationship is used (Dansgaard-Oeschger events, Glacial-Interglacial transitions (Dansgaard et al., 1982; Johnsen et al., 1992)). Also, in the Holocene where Greenland temperature variations are comparatively small, seasonal changes of precipitation as well as of evaporation conditions at the source region may contribute to water-isotope-data variations (Huber et al., 2006; Kindler et al., 2014; Werner et al., 2001). A relatively new method for ice-core-based temperature reconstructions uses the thermal fractionation of stable isotopes of air compounds (nitrogen and argon) within a firn layer of an ice sheet (Huber et al., 2006; Kindler et al., 2014; Kobashi et al., 2011; Orsi et al., 2014; Severinghaus et al., 1998, 2001). The measured nitrogen- and argon-isotope records of air enclosed in bubbles in an ice core can be used as a paleothermometer due to (i) the stability of isotopic compositions of nitrogen and argon in the atmosphere at orbital timescales and (ii) the fact that changes are only driven by firn processes (Leuenberger et al., 1999; Mariotti, 1983; Severinghaus et al., 1998). To robustly reconstruct the surface temperature for a given drilling site, the use of firn models describing gas- and heat-diffusion throughout the ice sheet is necessary to decompose the gravitational from the thermal-diffusion influence on the isotope signals.

This work addresses two issues relevant for temperature reconstructions based on nitrogen and argon isotopes. First, we introduce a novel, entirely automated approach for inverting gas-isotope data to surface-temperature estimates. For that, we force the output of a firn-densification and heat-diffusion model to fit gas-isotope data. This methodology can be used for many different optimization tasks not restricted to ice-core data. As we will show, the approach works besides $\delta^{15}N$ for all relevant gas-isotope quantities ($\delta^{15}N$, $\delta^{40}Ar$, $\delta^{15}N_{excess}$) and for Holocene and glacial data as well. Furthermore, the possibility of fitting all relevant gas-isotope quantities, individually or combined, makes it possible for the first time to validate the temperature solution gained from one single isotope species by comparison to the solution calculated from other isotope quantities. This approach is a completely new method which enables the automated fitting of gas-isotope data without any manual tuning of parameters, minimizing any potential "subjective" impacts on temperature estimates as well as working hours. Also, except for the model spin-up, the presented temperature-reconstruction approach is completely independent from water stable-isotopes ($\delta^{18}O_{ice}$, $\delta D_{ice}$), which provides the opportunity to validate water-isotope-based reconstructions (e.g. Masson-Delmotte, 2005) or reconstructions where water isotopes are used together with $\delta^{15}N$ or $\delta^{40}Ar$ (e.g. Capron et al., 2010; Huber et al., 2006; Landais et al., 2004). To our knowledge, there are only two other reconstruction methods independent from water stable-isotopes that have been applied to Holocene gas-isotope data, without a priori assumption on the shape of a temperature change. The studies from Kobashi et al. (2008a, 2017) use the second order parameter $\delta^{15}N_{excess}$ to calculate firn-temperature gradients, which are later temporally integrated from past to future over the time-series of interest using the firn-densification and heat-diffusion model from Goujon et al. (2003). Additionally Orsi et al. (2014) use a linearized firn-model approach together with $\delta^{15}N$ and $\delta^{40}Ar$ data to extract surface-temperature histories. The method presented here can be used when no $\delta^{40}Ar$ data are available, which is often the case because $\delta^{40}Ar$ is a more analytically challenging measurement and is not as commonly measured as $\delta^{15}N$ and further allows a comparison among solutions obtained from any of the available isotope quantities.

Second, we investigate the accuracy of our novel fitting approach by examining the method on different synthetic nitrogen-isotope and temperature scenarios. The aim of this work is to study the uncertainties emerging from the algorithm itself. Furthermore the focal question in this study is: what is the minimal mismatch in $\delta^{15}N$ for Holocene-like data we can reach and what is the implication for the final temperature mismatches. Studying and moreover answering these questions makes it mandatory to create well defined $\delta^{15}N$ targets and related temperature histories. It is impossible to answer these questions without using synthetic data in a methodology study. The aim is to evaluate the accuracy and associated uncertainty of the inverse method itself to then later apply this method to real $\delta^{15}N$, $\delta^{40}Ar$ or $\delta^{15}N_{excess}$ datasets, for which of course the original driving temperature histories are unknown.

## 2. Methods and data

### 2.1 Reconstruction approach

The problem that we deal with is an inverse problem, since the effect, observed as $\delta^{15}N$ variations, is dependent on its drivers, i.e. temperature and accumulation-rate changes. Hence, the temperature that we would like to reconstruct depends on $\delta^{15}N$ and accumulation-rate changes. To solve this inverse problem, the firn-densification and heat-diffusion model (from now on referred to as firn model), which is a non-linear transfer function of temperature and accumulation rate to firn states and relates to $\delta^{15}N$ values, is run iteratively to match the modelled and measured $\delta^{15}N$ values (or other gas species). The automated procedure is significantly more efficient and less time-consuming than a manual approach. The Holocene temperature-reconstruction is implemented by the following four steps (Fig. 01):

Step 1: A prior temperature input (first guess) is constructed, which serves as the starting point for the optimization.

Step 2: A long-term solution which passes through the $\delta^{15}N$ data (here synthetic target data) is generated following a Monte-Carlo approach. It is assumed that the smooth solution contains all long-term temperature trends (centuries to millennial) as well as firn-column-height changes (temperature and accumulation-rate dependent) that drive the gravitational background signal in $\delta^{15}N$.

Step 3: The long-term temperature solution is complemented by superimposing short-term information directly extracted from the $\delta^{15}N$ data (here synthetic target data). This step adds short-term temperature changes (decadal) in the same time resolution as the data.

Step 4: The gained temperature solution is further corrected using information extracted from the mismatch between the synthetic target and modelled $\delta^{15}N$ time-series.

The functionality of the presented inversion algorithm is schematically displayed in Fig. 01. It guides the reader through chapters and documents which variables, listed in Table 01, are in use. In the following a detailed description of each step is given.

### Step 1: prior input

The starting point of the optimization procedure is the first-guess. To construct the first-guess temperature-input $T_{g,0}(t)$, a constant temperature of -29.6 °C is used for the complete Holocene section, which corresponds to the last value of the temperature spin-up (Fig. 02b).

**Step 2: Monte-Carlo-type input-generator - Generating long-term solutions**

During the second step of the optimization, the prior temperature-input $T_{g,0}(t)$ from step 1 is iteratively (j) changed following a Monte-Carlo approach. The basic idea of the Monte-Carlo approach is to generate smooth temperature-inputs $T_{mc,j}(t)$ by superimpose low-pass-filtered values $\vec{P}_j$ of uniformly-distributed random values $\vec{P}_{r,j}$ on the prior input $T_{mc,j-1}$. Then, the new input is fed to the firn model and the mismatch $D_{\delta15N,mc,j}$ (with $X \equiv \delta^{15}N_{mc,j}$) between the modelled $\delta^{15}N_{mc,j}$ (here $X_{mod}$), calculated from the model output, and the synthetic $\delta^{15}N_{syn}$ (here $X_{target}$) is computed for every time step (i) of the target data $\delta^{15}N_{syn}$ according to:

$$D_X = \frac{1}{n}\sum_{i=1}^{n}|D_{X,i}| = \frac{1}{n}\sum_{i=1}^{n}|X_{target,i} - X_{mod,i}| \tag{1}$$

(Note: If not otherwise stated, all mismatches in this study labelled with "D" are calculated similar to eq. (1))

$D_{\delta15N,mc}$ serves as the criterion which is minimised during the optimization in step 2. If the mismatch $D_{\delta15N,mc,j}$ decreases compared to the prior input ($T_{mc,j-1}$, $D_{\delta15N,mc,j-1}$), the new input is saved and used as new guess ($T_{g,j} = T_{mc,j}$). This procedure is repeated until convergence is achieved leading to the final long-term temperature $T_{mc,fin}(t)$. Table 02 lists the number of improvements and iterations performed for the different synthetic datasets.

The perturbation of the current guess $T_{g,j}$ is conducted in the following way: Let $\vec{T}_{g,0} = T_{g,0}(t)$ be the vector containing the prior temperature-input. A second vector $\vec{P}_{r,1}$ with the same number of elements $n_{mc}$ as $\vec{T}_{g,0}$ is generated containing $n_{mc}$ uniformly-distributed random numbers within the limits of an also randomly (equally-distributed) chosen standard deviation s. s is chosen from a range of 0.05-0.50, which means that the maximum allowed perturbation of a single temperature value $T(t_0)$ is in a range of ±5 % to ±50 %. Creating the synthetic frequencies, $\vec{P}_{r,1}$ is low-pass filtered using cubic-spline-filtering (Enting, 1987) with an equally distributed random cut-off-period (COP) in the range of 500 yr to 2000 yr generating the vector $\vec{P}_1$. Hereby the low-pass filtering of $\vec{P}_{r,1}$ reduces the amplitudes of the perturbation of $\vec{T}_{g,0}$. The new surface temperature input $\vec{T}_{mc,1}$ is calculated from $\vec{P}_1$ according to:

$$\vec{T}_{mc,1} = \vec{T}_{g,0}^{T} \cdot (\hat{1} + \vec{P}_1) \tag{2}$$

The superscript "T" stands for transposed and $\hat{1}$ is the n by 1 matrix of ones.

This approach provides a high potential for parallel computing. In this study, an eight-core computer was used, generating and running eight different inputs of $\vec{T}_{mc}$ simultaneously, minimizing the time to find an improved solution. For example, during the 706 iterations for scenario S2, about 5600 different inputs were created and tested, leading to 351 improvements (see Table 02). Since it is possible to find more than one improvement per iteration step due to the parallelization on eight CPU's, the solution giving the minimal misfit $D_{\delta15N,mc,j}$ is chosen as new first-guess for the next iteration step. This leads to a decrease of the used improvements for the optimization (e.g. for S2, 172 of the 351 improvements were used). Additionally, a first gas-age scale ($\Delta age_{mc,fin}(t)$) is extracted from the model using the last improved conditions, which will then be used in step 3.

**Step 3: Adding short-term (high frequency) information**

In step 3 the missing short-term temperature history providing a suitable fit between modelled and synthetic $\delta^{15}N$ data is directly extracted from the pointwise mismatch $D_{\delta15N,mc,fin}(t)$, between the modelled $\delta^{15}N_{mc,fin}(t)$ obtained in step 2 and the synthetic $\delta^{15}N_{syn}$ target. Note that for a real reconstruction, this mismatch is calculated using the measured $\delta^{15}N_{meas}$ dataset instead of the synthetic one. $D_{\delta15N,mc,fin}(t)$ can be interpreted in first order as the detrended high-frequency signal of the synthetic $\delta^{15}N_{syn}$ target. $D_{\delta15N,mc,fin}(t)$ is transferred to the gas-age scale using $\Delta age_{mc,fin}(t)$ provided by the firn-model output for the smooth temperature input $T_{mc,fin}(t)$. This is needed to insure synchroneity between the high-frequency temperature variations $\Delta T(t)$ extracted from the mismatch $D_{\delta15N,mc,fin}(t)$ on the ice-age scale and the smooth temperature solution $T_{mc,fin}(t)$. Additionally, the signal is shifted by about 10 yr towards modern values to account for gas diffusion from the surface to the lock-in-depth (Schwander et al., 1993), which is not yet implemented in the firn model. This is necessary for adding the calculated short-term temperature changes $\Delta T(t)$ to the smooth signal $T_{mc,fin}(t)$. The $\Delta T$ values are calculated according to eq. (3):

$$\Delta T_i = \frac{D_{\delta15N,mc,fin,i}}{\Omega_{N_2,i}}, \tag{3}$$

using the thermal-diffusion sensitivity $\Omega_{N_2,i}$ for nitrogen-isotope fractionation from Grachev and Severinghaus (2003):

$$\Omega_{N_2,i} = \frac{8.656\ ‰}{\overline{T}_i} - \frac{1232\ ‰\cdot K}{\overline{T}_i^2} \tag{4}$$

$\overline{T}_i$ is the mean firn temperature in Kelvin which is calculated by the firn model for each time point i. To reconstruct the final (high frequency) temperature input $T_{hf}(t)$, the extracted short-term temperature signal $\Delta T(t)$ is simply added to the long-term temperature input $T_{mc,fin}(t)$:

$$T_{hf,i} = T_{mc,fin,i} + \Delta T_i \tag{5}$$

**Step 4: final correction of the surface temperature solution**

For a further improvement of the remaining $\delta^{15}N$ and resulting surface-temperature misfits ($D_{\delta15N,hf}(t)$, $D_{T,hf}(t)$), it is important to find a correction method that contains information that is also available when using measured data. The benefit of the synthetic data study is that several later unknown quantities can be calculated, and used for improving the reconstruction approach (see Sect. 3 and 4). For instance, it is possible to split the synthetic $\delta^{15}N_{syn}$ data in the gravitational and thermo-diffusion parts or to use the temperature misfit, which is unknown in reality. The idea underlying the correction algorithm explained hereafter is that the remaining misfits of $\delta^{15}N$ ($D_{\delta15N,hf}(t)$) and temperature ($D_{T,hf}(t)$) are connected to the Monte-Carlo (step 2) and high-frequency part (step 3) of the reconstruction algorithm. In the present inversion framework, it is not possible to find a long-term solution $\delta^{15}N_{mc,fin}$ (or $T_{mc,fin}$) which exactly passes through the $\delta^{15}N_{syn}$ (or $T_{syn}$) target in the middle of the variance in all parts of the time-series. This leads to a slightly over- or underestimation of $\delta^{15}N_{mc,fin}(t)$ and their corresponding temperature values $T_{mc,fin}(t)$. For example, a slightly too low (or too high) smooth temperature estimate $T_{mc,fin}$ leads to a small increase (or decrease) of the firn-column-height, creating a wrong gravitational background signal in

$\delta^{15}N_{mc,fin}$ on a later point in time (because the firn column needs some time to react). An additional error in the thermal-diffusion signal is also created due to the high-frequency part of the reconstruction (step 3), because the high-frequency information is directly extracted from the deviation of the synthetic target $\delta^{15}N_{syn}(t)$ and the modelled $\delta^{15}N_{mc,fin}(t)$ from the final long-term solution $T_{mc,fin}(t)$ of the Monte-Carlo part. Therefore, this error is transferred into the next step of the reconstruction and partly creates the remaining deviations.

To investigate this problem, the deviations $D_{\delta15N,mc,fin}(t)$ of the synthetic target data $\delta^{15}N_{syn}$ to $\delta^{15}N_{mc,fin}$ of the Monte-Carlo part are numerically integrated over a time window of 200 yr (Sect. 4, Supple. S3), and thereafter the window is shifted from past to future in 1 yr steps resulting in a time-series called IF(t). IF(t) equals a 200 yr running-mean of $D_{\delta15N,mc,fin}(t)$. For t, the mid position of the window is allocated. The time evolution of IF(t) is a measure for the deviation of the long-term solution $\delta^{15}N_{mc,fin}(t)$ (or $T_{mc,fin}(t)$) from the perfect middle passage through the target data $\delta^{15}N_{syn}(t)$ (or $T_{syn}(t)$) and for the slightly over- and underestimation of the resulting temperature.

$$IF(t) = \frac{1}{200} \int_{t-100}^{t+100} \left( \delta^{15}N_{syn}(t) - \delta^{15}N_{mc,fin}(t) \right) dt = \frac{1}{200} \int_{t-100}^{t+100} D_{\delta15N,mc,fin}(t) \, dt \qquad (6)$$

Next, the sample-cross-correlation-function (xcf) (Box et al., 1994) is applied to IF(t) and the remaining misfits $D_{\delta15N,hf}(t)$ of $\delta^{15}N$ after the high-frequency part. The xcf shows two extrema (Fig. 03a), a maximum ($xcf_{max}$) and a minimum ($xcf_{min}$) at two certain lags ($lag_{max,\delta15N}$ at $xcf_{max,\delta15N}$ and $lag_{min,\delta15N}$ at $xcf_{min,\delta15N}$). Now, the same analysis is conducted for IF(t) versus the temperature mismatch $D_{T,hf}(t)$ (Fig. 03b), which shows an equal behaviour (two extrema, $lag_{max,T}$ at $xcf_{max,T}$ and $lag_{min,T}$ at $xcf_{min,T}$). Comparing the two cross correlations show that $lag_{max,\delta15N}$ equals the negative $lag_{min,T}$ and $lag_{min,\delta15N}$ corresponds to the negative $lag_{max,T}$ (Fig. 03d,e). The idea for the correction is that the extrema in the cross-correlation IF(t) vs. $D_{\delta15N,hf}(t)$ with the positive lag (positive means here that $D_{\delta15N,hf}(t)$ has to be shifted to past values relative to IF(t)) creates the misfit of temperature $D_{T,hf}(t)$ on the negative lag (modern direction) of IF(t) vs. $D_{T,hf}(t)$ and vice versa. So IF(t) yields information about the cause and allows us to correct this effect between the remaining mismatches $D_{\delta15N,hf}(t)$ and $D_{T,hf}(t)$ over the whole time-series. The lags are not sharp signals, due to the fact that (i) the cross-correlations are conducted over the whole analysed record, leading to an averaging of this cause and effect relationship as well as that (ii) IF(t) is a smoothed quantity itself. The correction of the reconstructed temperature after the high-frequency part is conducted in the following way: From the two linear relationships between IF(t) and $D_{\delta15N,hf}(t)$ at the two lags ($lag_{max,\delta15N}$ at $xcf_{max,\delta15N}$, $lag_{min,\delta15N}$ at $xcf_{min,\delta15N}$) two sets of $\delta^{15}N$ correction values ($\Delta\delta^{15}N_{max}(t)$ from $xcf_{max,\delta15N}$ and $\Delta\delta^{15}N_{min}(t)$ from $xcf_{min,\delta15N}$) are calculated. Then the lags are being inverted (Fig. 03c,e) shifting the two sets of the $\delta^{15}N$ correction values to the attributed lags of the cross correlation between IF(t) and $D_{T,hf}(t)$ (e.g. $\Delta\delta^{15}N_{min}(t)$ to lag from $xcf_{max,T}$ from the cross correlation between IF(t) and $D_{T,hf}(t)$) therefore changing the time assignments of $\Delta\delta^{15}N_{min}(t)$ and $\Delta\delta^{15}N_{max}(t)$ to $\Delta\delta^{15}N_{min}(t+lag_{max,T})$ and $\Delta\delta^{15}N_{max}(t+lag_{min,T})$. Now, the $\Delta\delta^{15}N_{max}(t)$ and $\Delta\delta^{15}N_{min}(t)$ are component-wise summed up leading to the time-series $\Delta\delta^{15}N_{cv}(t)$. From eq. (3) with $\Delta\delta^{15}N_{cv,i}$ instead of $D_{\delta15N,mc,fin,i}$ the corresponding temperature correction values are calculated and added to the high-frequency temperature solution $T_{hf}(t)$ giving the corrected temperature $T_{corr}(t)$. Finally, $T_{corr}(t)$ is used to run the firn model to calculate the corrected $\delta^{15}N_{corr}(t)$ time-series. This cause and effect relationship found in the cross-correlations between IF(t) and

$D_{\delta15N,hf}(t)$, and $IF(t)$ and $D_{T,hf}(t)$, is exemplarily shown in Fig. 03 for scenario S1 and was found for all eight synthetic scenarios. The derived correction algorithm leads to a further reduction of the mismatches of about 40 % in $\delta^{15}N$ and temperature (see Sect. 3.2).

## 2.2 Firn densification and heat diffusion model

Surface-temperature reconstruction relies on firn densification combined with gas- and heat-diffusion (Severinghaus et al., 1998). In this study, the firn-densification and heat-diffusion model, developed by Schwander et al. (1997) is used to reconstruct firn parameters for calculating synthetic $\delta^{15}N$ values depending on the input time-series. It is a semi-empirical model based on the work of Herron and Langway (1980), Barnola et al. (1991), and implemented using the Crank and Nicholson algorithm (Crank, 1975) and was also used for the temperature reconstructions by Huber et al. (2006) and Kindler

et al. (2014). Besides surface-temperature time-series, accurate accumulation-rate data are needed to run the model. The model then calculates the densification and heat-diffusion history of the firn layer and provides parameters for calculating the fractionation of the nitrogen isotopes for each time step, according to the following equations:

$$\delta^{15}N_{grav}\left(z_{LID},t\right) = \left(e^{\frac{\Delta m \cdot g \cdot z_{LID}(t)}{R \cdot \overline{T}(t)}} - 1\right) \cdot 1000 \tag{7}$$

$$\delta^{15}N_{therm}(t) = \left[\left(\frac{T_{surf}(t)}{T_{bottom}(t)}\right)^{\alpha_T} - 1\right] \cdot 1000 \tag{8}$$

$$\delta^{15}N_{mod}(t) = \delta^{15}N_{grav}(t) + \delta^{15}N_{therm}(t) \tag{9}$$

$\delta^{15}N_{grav}(t)$ is the component of the isotopic fractionation due to the gravitational settling (Craig et al., 1988; Schwander, 1989) and depends on the lock-in-depth (LID) $z_{LID}(t)$ and the mean firn temperature $\overline{T}(t)$ (Leuenberger et al., 1999). g is the gravitational acceleration, $\Delta m$ the molar mass-difference between the heavy and light isotopes (equals $10^{-3}$ kg per mol for nitrogen) and R the ideal gas-constant. $z_{LID}$ is defined as a density threshold $\rho_{LID}$, which is slightly sensitive to surface

temperature, following the formula from Martinerie et al. (1994), with a small offset correction of 14 kg m$^{-3}$ to account for the presence of a non-diffusive zone (Schwander et al., 1997):

$$\rho_{LID}(kg \cdot m^{-3}) = \frac{1}{\frac{1}{\rho_{ice}} - 6.95 \cdot 10^{-7} \cdot \overline{T} - 4.3 \cdot 10^{-5}} - 14 \tag{10}$$

where

$$\rho_{ice}(kg \cdot m^{-3}) = 916.5 - 0.14438 \cdot \overline{T} - 1.5175 \cdot 10^{-4} \cdot \overline{T}^2 \tag{11}$$

The thermal-fractionation component of the $\delta^{15}N$ signal (Severinghaus et al., 1998) is calculated using eq. (8), where $T_{surf}(t)$ and $T_{bottom}(t)$ stand for the temperatures at the top and the bottom of the diffusive firn-layer. In contrast to $T_{surf}(t)$ which is an input parameter for the model, $T_{bottom}(t)$ is calculated by the model for each time step. The thermal-diffusion constant $\alpha_T$ was measured by Grachev and Severinghaus (2003) for nitrogen (eq. (12)):

$$\alpha_T = \left(8.656 - \frac{1323 \, K}{\overline{T}}\right) \cdot 10^{-3} \tag{12}$$

The firn model used here behaves purely as a forward model, which means that for the given input time-series the output parameters (here finally $\delta^{15}N_{mod}(t)$) can be calculated, but it is not easily possible to construct from measured isotope data the related surface-temperature or accumulation-rate histories. The goal of the presented study is an automatization of this inverse-modelling procedure for the reconstruction of the rather small Holocene temperature variations.

## 2.3 Measurement, input data and time scale

### Time scale

For the entire study the GICC05 chronology is used (Rasmussen et al., 2014; Seierstad et al., 2014). During the whole reconstruction procedure the two input time-series (surface temperature and accumulation rate) are split into two parts. The first part ranges from 20 yr to 10520 yr b2k (called "Holocene section") and the second one from 10520 yr to 35000 yr b2k ("spin-up section"). The entire accumulation-rate input, as well as the spin-up section of the surface-temperature input remains unchanged during the reconstruction procedure.

### Accumulation-rate data

Besides surface temperatures, accumulation-rate data are needed to drive the firn model. In this study we use the original accumulation rates, reconstructed in Cuffey and Clow (1997) produced using an ice-flow model adapted to the GISP2 location, but adapted to the GICC05 chronology (Rasmussen et al., 2008; Seierstad et al., 2014). A detailed description of the adaption procedure can be found in supplement S1. The raw accumulation-rate data for the main part of the spin-up section (12000 yr to 35000 yr b2k) are linearly interpolated to a 20 yr grid and low-pass filtered with a 200 yr cut-off-period (COP) using cubic-spline-filtering (Enting, 1987). For the Holocene section (20-10520 yr b2k) and the transition part between Holocene and spin-up section (10520 yr to 12000 yr b2k) the raw accumulation-rate data are linearly interpolated to a 1 yr grid to obtain equidistant integer point-to-point distances which are necessary for the reconstruction, and to preserve as much information as possible for this time period (Fig. 02a). Except for these technical adjustments, the accumulation-rate input remains unmodified, assuming high reliability of these data during the Holocene. The accumulation data were reconstructed using annual-layer-counting, and a thinning model which should lead to maximum relative uncertainty of 10 % for the first 1500 m of the 3000 m ice core (Cuffey and Clow, 1997). From the three accumulation-rate scenarios reconstructed in Cuffey and Clow (1997) and adapted here to the GICC05 chronology, the intermediate one is chosen (red curves in Fig. S01). Since the differences between the scenarios are not important for the evaluation of the reconstruction approach, they are not taken into account for this study.

Additionally, two sensitivity experiments were conducted (see supplement S2) in order to investigated (i) the influence of low-pass-filtering of the high-resolution accumulation rates on the model outputs and (ii) the possible contribution of the accumulation-rate variability on the $\delta^{15}N$ data during the Holocene. The first experiment shows that filtering the accumulation-rates with cut-off-periods in the range of 20 yr to 500 yr has nearly no influence on the modelled $\delta^{15}N$ or lock-in-depth as long as the major trends are being conserved. The second experiment leads to the finding that the accumulation-

rate variability explains about 12 % to 30 % of $\delta^{15}N$ variability. 30 % corresponds to the 8.2 kyr event and 12 % for the mean of the whole Holocene period including the 8.2 kyr event. Hence the influence of accumulation changes, excluding the extreme 8.2 kyr event, is generally below 10 % during most parts of the Holocene.

**$\delta^{18}O_{ice}$ data**

Oxygen-isotope data from the GISP2 ice-core-water samples measured at the University of Washington's Quaternary Isotope Laboratory are used to construct the surface-temperature input of the model spin-up (12 yr to 35 kyr b2k, Grootes et al., 1993; Grootes and Stuiver, 1997; Meese et al., 1994; Steig et al., 1994; Stuiver et al., 1995; data availability: Grootes and Stuiver, 1999). The raw $\delta^{18}O_{ice}$ data are filtered and interpolated in the same way as the accumulation-rate data for the spin-
10 up part.

**Surface-temperature spin-up**

The surface-temperature history of the spin-up section (Fig. 02a) is obtained by calibrating the filtered and interpolated $\delta^{18}O_{ice}$ data (eq. (13)) using the values for the temperature sensitivity $\alpha_{18O}$ and offset $\beta$ found by Kindler et al. (2014) for the
15 NGRIP ice core assuming a linear relationship of $\delta^{18}O_{ice}$ with temperature.

$$T_{spin}(t) = \frac{1}{\alpha_{18O}(t)} \cdot [\delta^{18}O_{ice}(t) + 35.2\ \text{‰}] - 31.4°C + \beta(t) \tag{13}$$

The values 35.2 ‰ and -31.4 °C are modern-time parameters for the GISP2 site (Grootes and Stuiver, 1997; Schwander et al., 1997). The spin-up is needed to bring the firn model to a well-defined starting condition that takes possible memory effects (influence of earlier conditions) of firn states into account.

**Generating synthetic target data**

In order to develop and evaluate the presented algorithm, eight temperature scenarios were constructed and used to model synthetic $\delta^{15}N$ data, which serve later as targets for the reconstruction. From these eight synthetic surface-temperature and related $\delta^{15}N$ scenarios (S1-S5 and H1-H3), three data sets (later called Holocene like scenarios H1-H3) were constructed in
such a way that the resulting $\delta^{15}N$ time-series are very close to the $\delta^{15}N$ values measured by Kobashi et al. (2008b) in terms of variability (amplitudes) and frequency (data resolution) of the GISP2 nitrogen-isotope data (Fig. 04, Fig. 05).

The synthetic surface-temperature scenarios S1-S5 are created by generating a long-term temperature time-series ($T_{syn,smooth}$) analogous to the Monte-Carlo part of the reconstruction procedure for only one iteration step (see Sect. 2.1). The values for the cut-off-period used for the filtering of the random values, and the s values (standard deviation of the random values, see
Sect. 2.1) for the first five scenarios can be found in Table 03. The long-term temperatures (Fig. 04I) are calculated on a 20 yr grid, which is nearly similar to the time resolution of the GISP2 $\delta^{15}N$ measurement values of about 17 yr (Kobashi et al., 2008b). For the Holocene-like scenarios, the smooth temperature time-series were generated from the temperature reconstruction for the GISP2 $\delta^{15}N$ data (not shown here). The final Holocene surface-temperature solution was filtered with a 100 yr cut-off to obtain the long-term temperature scenario.

Following this, high frequency information is added to the long-term temperature histories. A set of normally-distributed random numbers with a zero mean and a standard deviation (1σ) of 1 K for scenarios S1-S5 and 0.3 K for Holocene-like scenarios H1-H3 is generated on the same 20 yr grid and added up to the long-term temperature time-series. Finally, the resulting synthetic target-temperature-scenarios (Fig. 04II, Fig. 05I) are linearly interpolated to a 1 yr grid.

These synthetic temperatures are combined with the spin-up temperature and are used together with the accumulation-rate input to feed the firn model. From the model output the synthetic $\delta^{15}N$ targets are calculated according to section 2.1. The firn-model output provides ice-age as well as gas-age information. The final synthetic $\delta^{15}N$ target time-series ($\delta^{15}N_{syn}$) are set intentionally on the ice-age scale to mirror measured data, because no prior information is available for the gas-ice-age difference (Δage) for ice-core data.

## 3. Results

### 3.1 Monte Carlo type input generator

Figure 06 shows the evolution of the misfit $D_{\delta15N,mc,j}$ between the synthetic target data ($\delta^{15}N_{syn}$) versus the modelled output $\delta^{15}N_{mc,j}$ of the Monte-Carlo part (step 2) as a function of the applied iterations (j) for all synthetic scenarios. One can easily see that all scenarios show a steep decline of the mismatch during the first 50 to 200 iterations followed by a rather moderate

decrease, which finally leads to a constant value. During the Monte-Carlo part, it was possible to reduce the misfit $D_{\delta15N,mc}$ compared to the first-guess solution $D_{\delta15N,g,0}$ by about 15 % to 75 % depending on the scenario and the mismatch of the first-guess solution (see Table 02). This leads to a reduction of the temperature mismatches $D_{T,mc}$ compared to the first-guess temperature $D_{T,g,0}$ mismatch of about 51 % to 87 %.

Figure 07 provides the comparison between the first-guess (g,0; step 1) and Monte-Carlo (mc,fin; step 2) solution versus the

20 synthetic target data (syn) for the modelled $\delta^{15}N$ (a-c) and surface-temperature values (d-f) for scenario S5. Subplots (a) and (d) show the time-series of the synthetic target (black dotted line), the first-guess solution (blue line) and the Monte-Carlo solution (red line) for $\delta^{15}N$ and temperature. In subplots (b) and (e), the distribution of the pointwise mismatch $D_i$ of the first-guess (blue) and the Monte-Carlo solution (red) versus the synthetic target data for $\delta^{15}N$ ($D_{\delta15N}$) and temperature ($D_T$) can be found. Subplots (c) and (f) contain the time-series for $D_{\delta15N,i}$ and $D_{T,i}$. The $D_{\delta15N,mc,fin}(t)$ data (red) are used to calculate the

25 high-frequency signal, that is superimposed to the long-term temperature solution $T_{mc,fin}$ according to eq. (3) and eq. (5) (see Sect. 2.1, step 3). From Fig. 07 it can be concluded that the Monte-Carlo part of the reconstruction algorithm (step 2) leads to two major improvements of the first-guess solution. First, it is obvious that the Monte-Carlo approach corrects the offsets of the first-guess input (g,0), which shifts the midpoint of the distributions of $D_{\delta15N,mc,i}$ and $D_{T,mc,i}$ to zero (see blue against red in Fig. 07b,e). The second improvement is that the distributions become more symmetric and the misfit is overall reduced

(the distributions become narrower) compared to the first-guess, due to the middle passage through the $\delta^{15}N_{syn}$ targets. These improvements can be observed for all eight synthetic scenarios, showing the robustness of the Monte-Carlo part (see Table 02, Fig. 07).

## 3.2 High frequency step and final correction

Figure 08 provides the comparison between the Monte-Carlo (mc,fin; step 2), the high-frequency (hf; step 3) and the correction (corr; step 4) parts of the reconstruction procedure for the scenarios S5. Additional data for all other scenarios can be found in Table 04. The upper four plots (a-d) illustrate each reconstruction step and their effect on the modelled $\delta^{15}$N; the bottom four plots (e-h) show the corresponding results on the temperature. Plots (a) and (e) contain the time-series of the synthetic $\delta^{15}N_{syn}$ or $T_{syn}$ target (syn; black dotted line), the high-frequency solution (hf; blue line), and the final solution after the correction part (corr; red line). For visibility reasons, subplots (b) and (f) display a zoom-in for a randomly chosen time-window of about 500 yr for the same quantities, which shows the excellent agreement in timing and amplitudes of the modelled $\delta^{15}$N and temperature compared to the synthetic target data. Histograms (c) and (g) and subplots (d) and (h) show the distribution and the time-series of the pointwise mismatches ($D_{\delta15N,i}$ for $\delta^{15}$N; $D_{T,i}$ for temperature) between the modelled and the synthetic target data in $\delta^{15}$N and temperature for each reconstruction step.

Compared to the Monte-Carlo solution, the high-frequency part leads to a large refinement of the reconstructions. For the mean $\delta^{15}$N misfits $D_{\delta15N}$, the improvement between the Monte-Carlo and the high-frequency parts is in the range of 64 % to 76 % (see Table 04). This leads to a reduction of the temperature mismatches $D_T$ of 43 % to 67 %. The standard deviations (1$\sigma$) of the pointwise mismatches (Fig. 08c,d,g,h) in $\delta^{15}$N and temperature after the high-frequency parts are in the range of about 2.7 permeg to 5.4 permeg (one permeg equals $10^{-6}$) for $\delta^{15}$N and 0.22 K to 0.40 K for the reconstructed temperatures depending on the scenario, which is clearly visible in the decreasing width of the histograms (subplots (c) and (g) of Fig. 08, blue against grey).

The mismatches after the correction part of the reconstruction approach show clearly a further decrease of the misfits. This means that the width of the distributions of the pointwise mismatches $D_{\delta15N,i}$ as well as $D_{T,i}$ is further reduced, and the distributions become more symmetric (long tales disappear; see histogram (c) and (g); red against blue of Fig. 08). The time series of the mismatches (subplots (d) and (h) of Fig.08) clearly illustrate that the correction approach mainly tackles the extreme deviations (sharp reduction of extreme values occurrence in the red distribution compared to the blue distribution) leading to a further improvement of about 40 % in $\delta^{15}$N and temperature. Finally, the 95 % quantiles ($2\sigma_{\delta15N,corr,95}$, $2\sigma_{T,corr,95}$) of the remaining pointwise mismatches of $\delta^{15}$N and temperature ($D_{\delta15N,i}$ or $D_{T,i}$) were calculated for the final solutions for all scenarios and are used as an estimate for the 2$\sigma$ uncertainty of the reconstruction algorithm (see Fig. 08c,g and Table 04). The final uncertainties (2$\sigma$) are in the order of 3.0 permeg to 6.3 permeg for $\delta^{15}$N and 0.23 K to 0.51 K for the surface temperature misfits. It is noteworthy that the measurement uncertainties (per point) of state of the art $\delta^{15}$N measurements are in the same order of magnitude, i.e. 3 permeg to 5 permeg (Kobashi et al., 2008b), highlighting the effectiveness of the presented fitting approach. Table 05 contains the final mismatches (2$\sigma$) in $\Delta$age between the synthetic target and the final modelled data after the correction step for all scenarios and shows that with a known accumulation rate and assumed perfect firn physics, it is possible to fit the $\Delta$age history in the Holocene with mean uncertainties better than 2 yr. In other words, the uncertainty in $\Delta$age reconstruction due to the inversion algorithm alone is in the order of 2 yr.

## 4. Discussion

### 4.1 Monte Carlo type input generator

Figure 09 shows the distribution of the cut-off-periods (COP) (I) and s values (II) used to create the improvements (Sect. 2.1, step 2) for all scenarios. The cut-off-periods are more or less evenly distributed, which shows that nearly the whole of the allowed frequency range (500 yr to 2000 yr) was used to create the improvements during the iterations. In contrast, the distributions of the s values show clearly that mostly small s values are used to create the improvements, which implies that iterations with small perturbations more likely lead to an improvement than larger ones.

Figure 06 reveals a weak point of the Monte-Carlo part, namely the absence of a suitable termination criterion for the optimization. The implementation until now is conducted such that the maximum number of iterations is given by the user or the iterations are terminated after a certain time (e.g. 15 h). Figure 06 shows that for nearly all scenarios it would be possible to stop the optimization after about 400 iterations, due to rather small additional improvements later on. This would decrease the time needed for the Monte-Carlo part to about 10 h (a single iteration needs about 90 s). Since the goal of the Monte-Carlo part is to find a temperature realisation that leads to an optimal middle passage through the $\delta^{15}$N target data, it would be possible to use the mean difference between the $\delta^{15}$N target and spline-filtered $\delta^{15}$N data using a certain cut-off-period as a termination criterion. This issue is under investigation at the moment. Another possibility to decrease the time needed for the Monte-Carlo part could be an increase in the numbers of CPUs used for the parallelization of the model runs. For this study an eight-core parallelization was used. A further increase in numbers of workers would improve the speed of the optimization.

### 4.2 High frequency step and final correction

To investigate the timing and contributions of the remaining mismatches in $\delta^{15}$N and temperature for scenario S1 after the high-frequency (step 3) and correction part (step 4), different cross-correlation experiments were conducted (see supplement S3). The experiments lead to equal results. The major fraction of the final mismatches of $\delta^{15}$N emerges from mismatches in the thermal-diffusion component $D_{\delta15Ntherm}$. Also a cancelation effect between the gravitational component $D_{\delta15Ngrav}$ and $D_{\delta15Ntherm}$ of the total mismatch in $\delta^{15}$N became obviously, affecting the calculation of $lag_{max,\delta15N}$ and $lag_{min,\delta15N}$ and most likely leading to a fundamental residual uncertainty in the low-permeg level for the corrected $\delta^{15}$N data. The same analyses were conducted for all synthetic scenarios, leading to similar results.

Additionally, the influence of the window length, used for the calculation of IF(t), on the correction was analysed, showing that for all investigated window lengths the correction reduces the mismatches of $\delta^{15}$N and temperature, whatever correction mode was used (calculated with $xcf_{max}$, $xcf_{min}$, or both quantities). Moreover, the correction is most efficient for window lengths in the range of 100 yr to 300 yr with an optimum at 200 yr for all cases.

### 4.3 Key points to be considered for the application to real data

*Benefits of the novel gas isotope fitting approach*

In addition to the fitting of $\delta^{15}N$ data, the algorithm is able to fit $\delta^{40}Ar$ and $\delta^{15}N_{excess}$ data as well using the same basic concepts (Fig. 10). Here the $\delta^{40}Ar$ and $\delta^{15}N_{excess}$ data from Kobashi et al. (2008b) were used as the fitting targets. We reach

final mismatches ($2\sigma$) of 4.0 permeg for $\delta^{40}Ar/4$ and 3.7 permeg for $\delta^{15}N_{excess}$, which are for both quantities below the analytical measurement uncertainty of 4.0 permeg to 9.0 permeg for $\delta^{40}Ar/4$ and 5.0 permeg to 9.8 permeg for $\delta^{15}N_{excess}$ measured data (Kobashi et al., 2008b).

The automated inversion of different gas-isotope quantities ($\delta^{15}N$, $\delta^{40}Ar$, $\delta^{15}N_{excess}$) provides a unique opportunity to study the differences in the gained solutions using different targets and to improve our knowledge about the uncertainties of gas-

isotope-based temperature reconstructions using a single firn model. Next, the presented algorithm is not dependent on the firn model, which leads to the implication that the algorithm can be coupled to different firn models describing firn physics in different ways. Furthermore, an automated reconstruction algorithm avoiding manual manipulation and leading to reproducible solutions makes it possible for the first time, to study and learn from the differences between solutions matching different targets. Finally, differences obtained by applying different firn physics (densification equations,

convective zone, etc.) but the very same inversion algorithm may help to assess firn model shortcomings, resulting in more robust uncertainty estimates than it was ever possible before.

In this publication we show the functionality and the basic concepts of the automated inversion algorithm using well known synthetic $\delta^{15}N$ fitting targets. In this "perfect world scenario" the forward problem, converting surface temperature to $\delta^{15}N$, as well as the inverse problem, converting $\delta^{15}N$ to surface temperature, is completely described by the used firn model.

Consequently all sources of signal noise are ignored. For the later use of the algorithm on $\delta^{15}N$, $\delta^{40}Ar$ or $\delta^{15}N_{excess}$ measured data this will not be the case anymore due to different sources of signal noise in the used measured data. As a result, differences between temperature solutions obtained from individual targets ($\delta^{15}N$, $\delta^{40}Ar$, $\delta^{15}N_{excess}$) will become obvious. These differences will allow to quantify the uncertainties associated with different unconstrained processes. Next, we will list and discuss potential sources of uncertainties and try to provide suggestions for their handling and quantification in our

approach.

*Measurement uncertainty and firn heterogeneity (cm-scale variability):*

Many studies have investigated the influence of firn heterogeneity (or density fluctuations) on measurements of air compounds and quantities (e.g. $\delta^{15}N$, $\delta^{40}Ar$, $CH_4$, $CO_2$, $O_2/N_2$ ratio, air content) extracted from ice cores resulting in cm-scale

variability and leading to additional noise on the measured data (e.g. Capron et al., 2010; Etheridge et al., 1992; Fourteau et al., 2017; Fujita et al., 2009; Hörhold et al., 2011; Huber and Leuenberger, 2004; Rhodes et al., 2013, 2016). Using discrete measurement technique instead of continuous sampling methods makes it difficult to quantify these effects. However, during discrete analyses of ice-core air-data it is common to measure replicates for given depths, from which the measurement uncertainties of the gas-isotope data are calculated using pooled-standard-deviation (Hedges L. V., 1985). Often it is not

possible to take real replicates (same depth) and instead the replicates are taken from nearby depths. Hence, any potential cm-scale variability is to some degree already included in the measurement uncertainty, because each measurement point represents the average over a few centimetres of ice. This is especially the case for low-accumulation sites or glacial ice samples for which the vertical length of a sample (e.g., 10-25 cm long for the glacial part of the NGRIP ice core, Kindler et al., 2014) covers the equivalent of 20 yr to 50 yr of ice at approximately 35 kyr b2k. Increasing the depth resolution of the samples would increase our knowledge of cm-scale variability, for e.g. identifying anomalous entrapped gas-layers that could have been rapidly isolated from the surface due to an overlying high-density layer (e.g., Rosen et al., 2014). As this variability is likely due to heterogeneity in the density profile, modelling such heterogeneities (if possible at all) may not help to better reconstruct a meaningful temperature history, but rather to reproduce the source of noise. This means that the potential cm-scale variability, in many cases, is already incorporated in the analytical noise obtained from gas-isotope measurements, due to analytical techniques themselves. Assuming the measurement uncertainty as Gaussian distributed, it is easy to incorporate this source of uncertainty in the inverse-modelling approach presented here. This will increase the uncertainty of the temperature according to eq. (3). The same equation can also be used for the calculation of the uncertainty in temperature related to measurement uncertainty in general.

To answer the pertinent question of how to better extract a meaningful temperature history from a noisy ice-core record, an excellent – but costly – solution is of course to use multiple ice cores. For example, a $\delta^{15}$N-based temperature reconstruction from the combination of data from the GISP2 ice core with the "sister ice core" GRIP drilled 30 kilometres apart is likely one of the best ways to overcome potential cm-scale variability. A comparison of ice cores that were drilled even closer might be even more advantageous.

*Smoothing effects due to gas diffusion and trapping:*

It is known that gas-diffusion and trapping processes in the firn can smooth out fast signals and result in a damping of the amplitudes of gas-isotope signals (e.g. Grachev and Severinghaus, 2005; Spahni, 2003). The duration of gas diffusion from the top of the diffusive column to the bottom where the air is closed off in bubbles is for Holocene conditions in Greenland approximately in the order of 10 yr (Schwander et al. 1997), whereas the data resolution of the synthetic targets was set to 20 yr to mimic the measurement data from Kobashi et al. (2008b) with a mean data resolution of about 17 yr (see Sect. 2.3: "Generating synthetic target data"). In the study of Kindler et al. (2014) it was shown that a glacial Greenland lock-in-depth leads to a damping of the $\delta^{15}$N signal of about 30 % for a 10 K temperature rise in 20 yr. We further assume that the smoothing according to the lock-in process is negligible for Greenland Holocene conditions according to the much smaller amplitude signals and shallower LID. Yet, for glacial conditions it requires attention.

*Accumulation rate uncertainties:*

For the synthetic data study presented in this paper it is assumed that the used accumulation-rate data are well known with zero uncertainty. This simplification is used to show the functionality and basic concepts of the presented fitting algorithm in

every detail on well-known $\delta^{15}$N and temperature targets and to focus on the final uncertainties originating from the presented fitting algorithm itself. For the later reconstruction using measured gas-isotope data together with the published accumulation-rate scenarios shown in supplement S1 this will not be the case anymore. Uncertainties in layer-counting and corrections for ice thinning lead to a fundamental uncertainty. Especially in the early Holocene, this can easily exceed 10 %.

As the accumulation-rate data are used to run the firn model, all potential accumulation uncertainties are in part incorporated into the temperature reconstruction. On the other hand, as we discussed in supplement S2, the accumulation rate variability has a minor impact compared to the input temperature on the variability of $\delta^{15}$N data in the Holocene. The influence of these quantities, accumulation rate or temperature, on the temperature reconstruction is not equal; during the Holocene, accumulation-rate variability explains about 12 % to 30 % of $\delta^{15}$N variability. 30 % corresponds to the 8.2 kyr event and

12 % for the mean of the whole Holocene period including the 8.2 kyr event. Hence the influence of accumulation changes, excluding the extreme 8.2 kyr event, is generally below 10 % during the Holocene. If the accumulation is assumed to be completely correct then the missing part will be assigned to temperature variations. Nevertheless for the fitting of the Holocene measurement-data we will use all three accumulation-rate scenarios as shown in S1. The difference in the reconstructed temperatures arising from the differences of these three scenarios will be used for the uncertainty calculation as

well and is most likely higher than the uncertainty arising from uncertainties due to the process of producing the accumulation-rate data and from the conversion of the accumulation-rate data to the GICC05 timescale.

*Convective zone variability:*

Many studies have shown the existence of a well-mixed zone at the top of the diffusive-firn-column, called convective zone

(CZ). The CZ is formed by strong katabatic winds and pressure gradients between the surface and the firn (e.g. Kawamura et al., 2006, 2013; Severinghaus et al., 2010). The existence of a CZ changes the gravitational background signal in $\delta^{15}$N and $\delta^{40}$Ar as it reduces the diffusive-column-height. The presented fitting algorithm was used together with the two most frequently used firn models for temperature reconstructions based on stable isotopes of air, the Schwander et al. (1997) model which has no CZ build in (or better a constant CZ of 0 m) and the Goujon firn model (Goujon et al., 2003) (which

assumes a constant convective zone over time, that can easily be set in the code). This difference between the two firn models only changes significantly the absolute temperature rather than the temperature anomalies as it was shown by other studies (e.g., Guillevic et al., 2013, Fig. 3). In the presented work, we show the results using the model from Schwander et al. (1997), because the differences between the obtained solutions using the two models are negligible besides a constant temperature offset. Also, noteworthy is that there is no firn model at the moment which uses a dynamically changing CZ.

Indeed, this should be investigated but requires additional intense work. Additionally, the knowledge of the time-evolution of CZ changes for time periods of millennia to several hundreds of millennia (in frequency and magnitude) is too poor to estimate the influence of this quantity on the reconstruction. In principle it is possible to cancel-out the influence of a potentially changing CZ by using $\delta^{15}$N$_{excess}$ data for temperature reconstruction, as due to the subtraction of $\delta^{40}$Ar/4 from $\delta^{15}$N the gravitational term of the signals is eliminated. From that point of view it will be interesting to compare temperature

solutions gained from $\delta^{15}N_{excess}$ fitting with the solutions based on $\delta^{15}N$ or $\delta^{40}Ar$ alone. This can offer a useful tool for quantifying the magnitude and frequency of CZ changes in the time interval of interest.

It is known that for some very low accumulation-rate sites in areas with strong katabatic winds (e.g. "Megadunes", Antarctica) extremely deep CZs can occur, which are potentially able to smooth-out even decadal-scale temperature

variations (Severinghaus et al., 2010). For this its deepness would need to be of several dozens of meters, which is highly unrealistic even for glacial Summit conditions (Guillevic et al., 2013, see discussion in Annex A4, p. 1042) as well as for the rather stable Holocene period in Greenland for which no low accumulation and strong katabatic-wind situations are to be expected.

## 4.4 Proof of concept for glacial data

For glacial conditions the task of reconstructing temperature (with correct frequency and magnitude) without using $\delta^{18}O_{ice}$ information is much more challenging due to the highly variable gas-age - ice-age differences ($\Delta$age) between stadial and interstadial conditions. Here, contrary to the rather stable Holocene period, the $\Delta$age can vary by several hundreds of years. Also the accumulation-rate data are more uncertain than for the Holocene. To prove that the presented fitting algorithm also works for glacial conditions we inverted the $\delta^{15}N$ data measured for the NGRIP ice core by Kindler et al. (2014) for two

Dansgaard-Oeschger events, namely DO6 and DO7. Since the magnitudes of those events are higher and the signals are smoother than in the Holocene we only had to use the Monte-Carlo-type input-generator (see Sect. 2.3.2) for changing the temperature inputs. To compare our results to the $\delta^{18}O_{ice}$ based and manually calibrated values from Kindler et al. (2014) we use the ss09sea06bm time scale (NGRIP members: Andersen et al., 2004; Johnsen et al., 2001) as it was done in the Kindler et al. publication. For the model spin-up we use the accumulation-rate and temperature data from Kindler et al. (2014) for the

time span 36.2 kyr to 60 kyr. The reconstruction window (containing DO6 and DO7) is set to 32 kyr to 36.2 kyr. As the first-guess (starting point) of the reconstruction we use the accumulation-rate data ($Acc_{g,0}$) for NGRIP from the ss09sea06bm time-scale together with a constant temperature of about -49 °C for this time window. As minimization-criterion $D_g$ for the reconstruction we simply use the sum of the root-mean-squared-errors of the $\delta^{15}N$ and $\Delta$age mismatches weighted with their uncertainties (wRMSE) according to the following equation, instead of the mean $\delta^{15}N$ misfit alone as used for the Holocene

(eq. (1)).

$$D_{gl} = wRMSE(\delta^{15}N) + wRMSE(\Delta age) \qquad (14)$$

$$= \sqrt{\frac{1}{N} \sum_i \left[ \frac{\delta^{15}N_{meas,i} - \delta^{15}N_{mod,i}}{\varepsilon_{\delta^{15}N,i}} \right]^2} + \sqrt{\frac{1}{M} \sum_k \left[ \frac{\Delta age_{meas,k} - \Delta age_{mod,k}}{\varepsilon_{\Delta age,k}} \right]^2}$$

Here $\varepsilon_{\delta^{15}N,i}$ and $\varepsilon_{\Delta age,k}$ are the uncertainties in $\delta^{15}N$ and $\Delta$age for the measured values i or k ($\Delta$age match points: Guillevic, M. (2013), p.65, Tab. 3.2) and N, M the number of measurement values. We set $\varepsilon_{\delta^{15}N,i} = 20$ permeg for all i (Kindler et al., 2014) and $\varepsilon_{\Delta age,k} = 50$ yr for all k. The relative uncertainties in $\Delta$age can easily reach up to 50 % and more in the Glacial

using the ss09sea06bm time-scale which results in a pre-eminence of the $\delta^{15}N$ misfits over the $\Delta$age misfits (10 % to 20 %

when using GICC05 time-scale; Guillevic, M., 2013, p. 65 Tab. 3.2). Due to this issue we have to set $\Delta$age uncertainties to 50 yr to make both terms equally important for the fitting algorithm. In Fig. 11 we show preliminary results. The $\delta^{15}N$ and $\Delta$age fitting (a, b) and the resulting gained temperature and accumulation-rate solutions (c, d) using the presented algorithm are completely independent from $\delta^{18}O_{ice}$ which provides the opportunity to evaluate the $\delta^{18}O_{ice}$-based reconstructions. In this study the algorithm was used in three steps. First, starting with the first-guess (constant temperature), the temperature was changed as explained before. The accumulation rate was changed in parallel to the temperature allowing a random offset shift (up and down) together with a stretching or compressing (in y direction) of the accumulation-rate signal over the whole time-window (32 kyr to 36.2 kyr). This first step leads to the "Monte-Carlo Solution 0" (MCS0) which provides a first approximation and is the base for the next step. For the next step, we fixed the accumulation rate and let the algorithm only change the temperature to improve the $\delta^{15}N$-fit (MSC1). Finally, we allow the algorithm to change the temperature together with the accumulation rate using the Monte-Carlo-type input-generator for both quantities. This allows to change the shape of the accumulation-rate data. This final step can be seen as a fine tuning of the gained solutions from the steps before. The obtained mismatches in $\delta^{15}N$ and $\Delta$age of all steps are at least of the same quality or better than the $\delta^{18}O_{ice}$-based manual method from Kindler et al. (2014) (see Table 06). The gained temperature solutions show a very good agreement in timing and magnitude compared to the reconstruction of Kindler et al. (2014). Also the accumulation-rate solutions show that the accumulation has to be reduced significantly compared to the ss09sea06bm data to allow a suitable fit of the $\delta^{15}N$ and $\Delta$age target data, a result highly similar to Guillevic et al. (2013) and Kindler et al. (2014). The mismatches in $\delta^{15}N$ and $\Delta$age of the final MCS FIN solution show a 15 % smaller misfit of $\delta^{15}N$ ($2\sigma$) and an about 31 % smaller misfit of $\Delta$age ($2\sigma$) compared to the Kindler et al. (2014) solution. Keeping in mind that the used approach is completely independent from $\delta^{18}O_{ice}$ strengthens the functionality and quality of the presented gas-isotope fitting approach also for glacial reconstructions. As this section contains a proof-of-concept of the presented automated gas-isotope fitting algorithm on glacial data, preliminary results and ongoing work were shown here. Furthermore as the presented fitting algorithm was developed and tested in first order for Holocene-like data, it is highly probable that the functionality of the algorithm using glacial data will be further extended and adjusted in future studies.

## 5. Conclusion

A novel approach is introduced and described for inverting a firn-densification and heat-diffusion model to fit small gas-isotope-data variations as observed throughout the Holocene. From this new fitting method, it is possible to extract the surface-temperature history that drives the firn status which in turn leads to the gas-isotope time-series. The approach is a combination of a Monte-Carlo-based iterative method and the analysis of remaining mismatches between modelled and target data. The procedure works fully automated and provides a high potential for parallel computing for time consumption optimization. Additional sensitivity experiments have shown that accumulation-rate changes have only a minor influence on short-term variations of $\delta^{15}N$, which themselves are mainly driven by high-frequency temperature variations. To evaluate the

performances of the presented approach, eight different synthetic $\delta^{15}N$ time-series were created from eight known temperature histories. The fitting approach leads to an excellent agreement in timing and amplitudes between the modelled and synthetic $\delta^{15}N$ and temperature data. The obtained final mismatches follow a symmetric standard-distribution-function. 95 % of the mismatches compared to the synthetic data are in an envelope between 3.0 permeg to 6.3 permeg for $\delta^{15}N$ and

0.23 K to 0.51 K for temperature, depending on the synthetic temperature scenarios. These values can therefore be used as a $2\sigma$ estimate for the reconstruction uncertainty arising from the presented fitting algorithm itself. For $\delta^{15}N$ the obtained final uncertainties are in the same order of magnitude as state of the art measurement uncertainty. The presented reconstruction approach was also successfully applied to $\delta^{40}Ar$ and $\delta^{15}N_{excess}$ measured data. Moreover, we have shown that the presented fitting approach can also be applied to glacial temperature reconstructions with minor algorithm modifications. Based on the

demonstrated flexibility of our inversion methodology, it is reasonable to adapt this approach for reconstructions of other non-linear physical processes.

**Data availability**

The synthetic $\delta^{15}N$ and temperature targets, the reconstructed $\delta^{15}N$ and temperature data, and the used accumulation rates can be found in the data supplement of this paper. The GISP2 $\delta^{18}O_{ice}$ data used in this study for calculating the temperature spin-

up can be found in Grootes and Stuiver (1999). The source code for the inversion algorithm and additional auxiliary data are available upon request.

**Competing interests**

The authors declare that they have no competing financial interests.

**Acknowledgements**

This work was supported by the SNF grants (SNF-159563 and SNF-172550). We would like to thank Drs. Takuro Kobashi, Philippe Kindler and Myriam Guillevic for helpful discussions about the ice core data and model peculiarities.

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

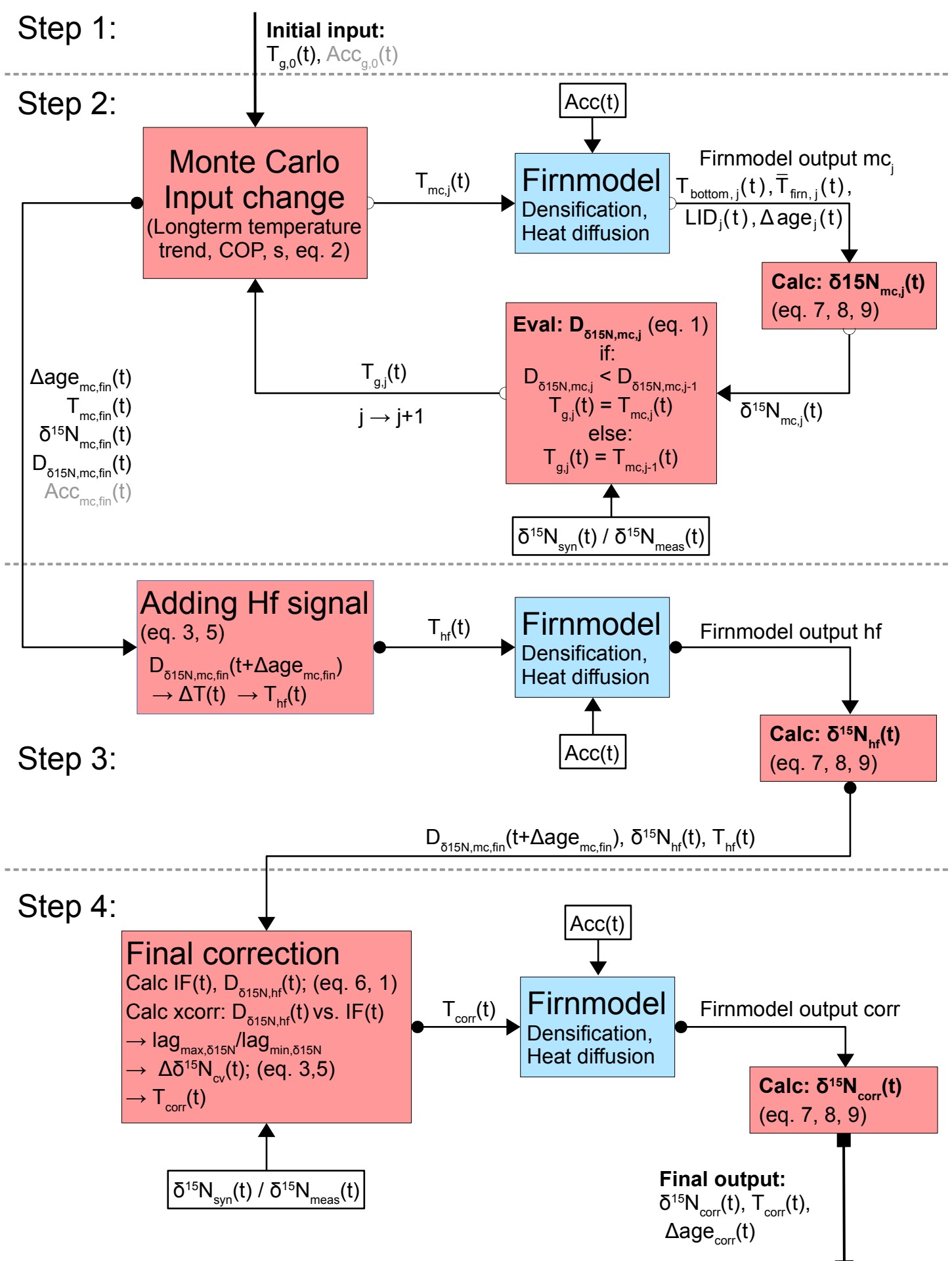

**Figure 01: Schematic illustration of the presented gas-isotope fitting algorithm. The algorithm is implemented in four steps: Step 1: first-guess input calculation; Step 2: iteratively Monte-Carlo-based input change (indicated by the open half-cycles); Step 3: signal complementation with high-frequency information; Step 4: final correction. In contrast to the synthetic data study on Holocene-like data where the accumulation input Acc(t) was fixed, for the proof-of-concept on glacial data the acccumulation and temperature input was iteratively changed in parallel indicated by the grey variables $Acc_{g,0}$ and $Acc_{mc,fin}$. For the glacial study only step 1 and 2 were used.**

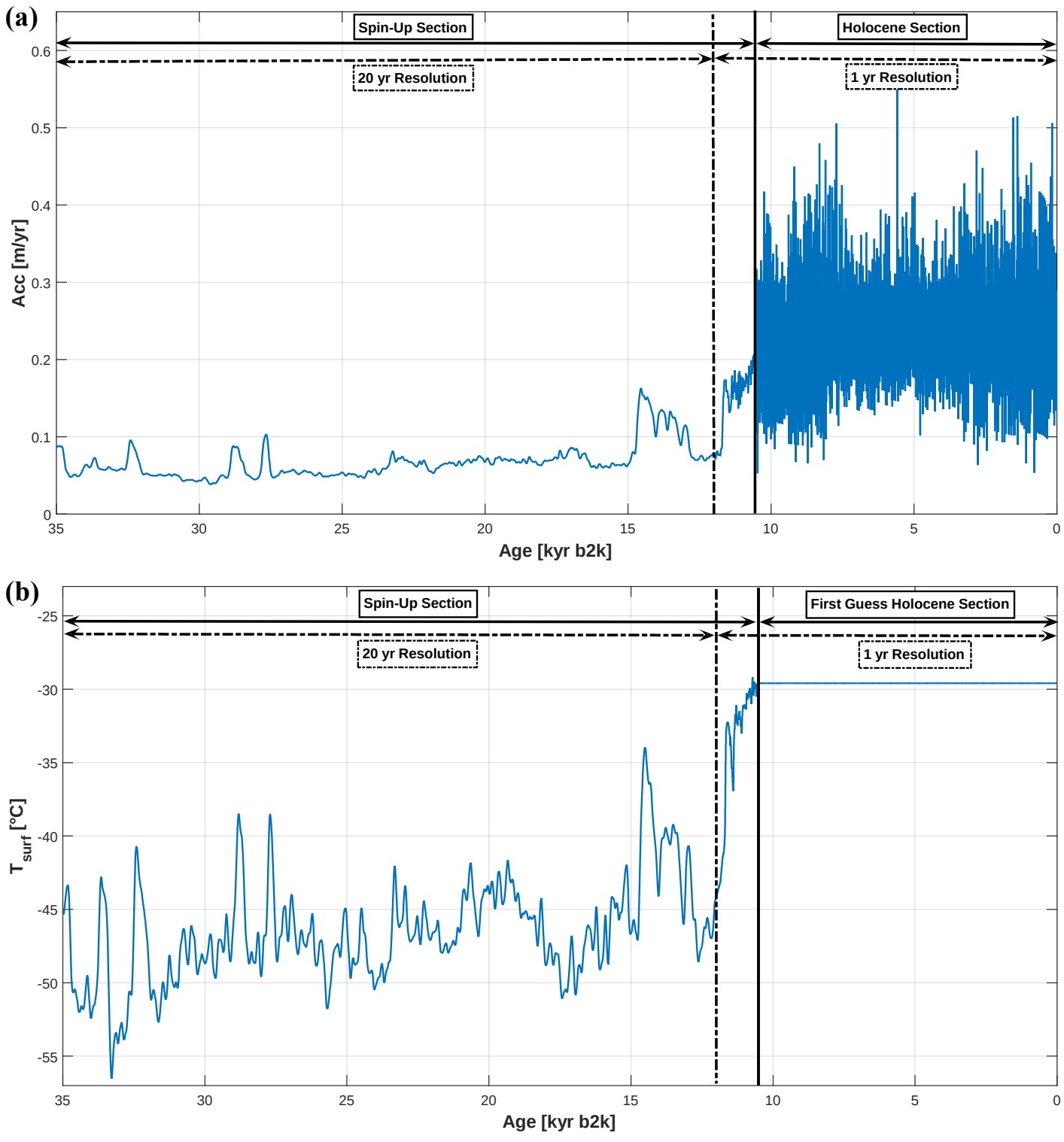

**Figure 02: (a)** Used accumulation-rate input time-series divided in a Holocene and a spin-up section, with time resolution in the Holocene section (20 yr to 10520 yr b2k) of 1 yr. The time resolution for the transition between the Holocene and the spin-up section (10520 yr to 12000 yr b2k) is 1 yr as well. This is in opposition to the rest of the spin-up section which has a time resolution of 20 yr. **(b)** Surface-temperature spin-up calculated from $\delta^{18}O_{ice}$ calibration. Time resolution equals the accumulation-rate spin-up section. First-guess surface temperature input is simply a constant value.

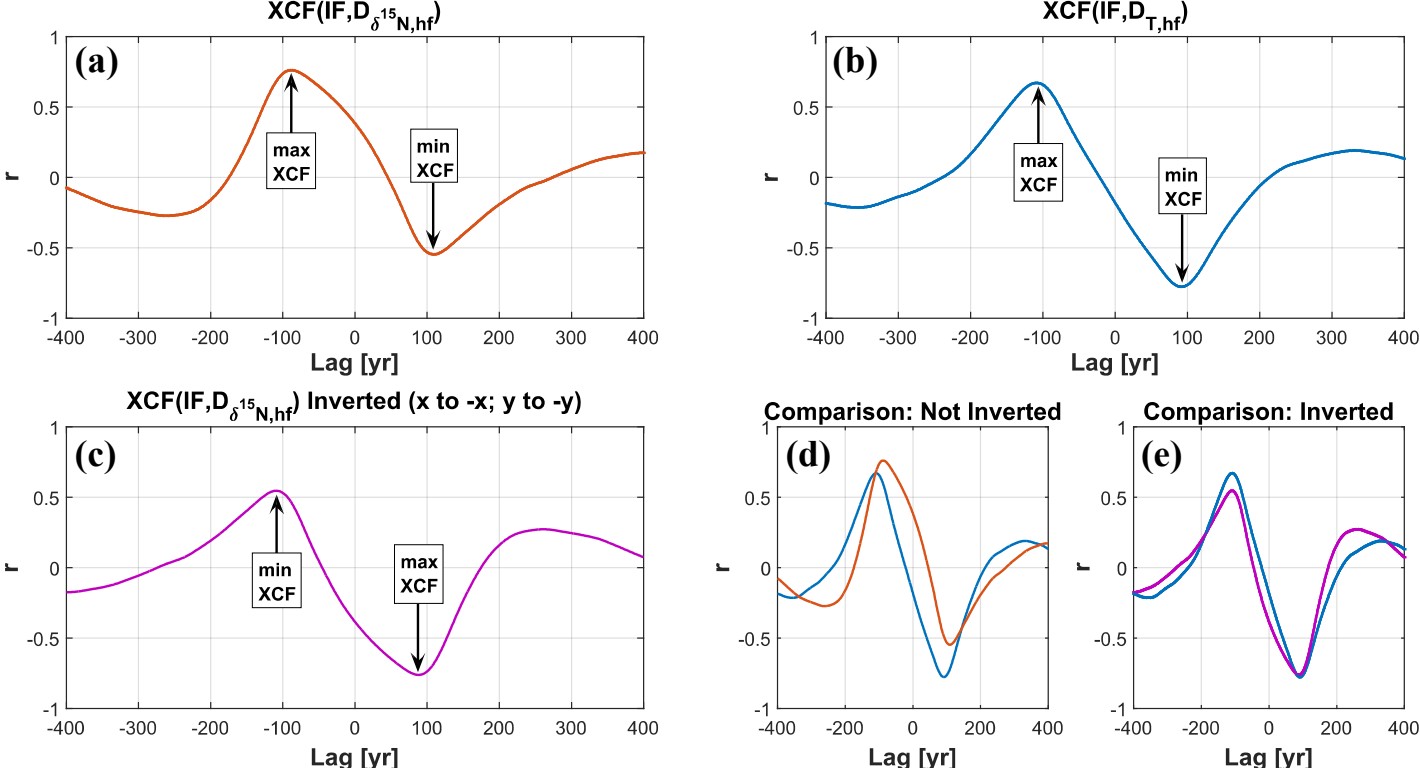

**Figure 03: Scenario S1:** (a) Cross-correlation-function (xcf) between IF(t) and the remaining mismatch in $\delta^{15}N$ ($D_{\delta15N,hf}(t)$) after the high-frequency part, shows two extrema: the maximum correlation (max xcf) and the minimum correlation (min xcf). (b) Cross-correlation-function (xcf) between IF(t) and the remaining mismatch in temperature ($D_{T,hf}(t)$) after the high-frequency part shows two extrema: the maximum correlation (max xcf) and the minimum correlation (min xcf). (c) Inverting of (a) in x (lag) and y (correlation coefficient) direction. (d) Comparison between (a) and (b). (e) Comparison between (a) and (c). The temperature-correction values are calculated from the linear dependency between IF(t) and $D_{\delta15N,hf}(t)$. After shifting IF(t) to max xcf (lag max) and to min xcf (lag min), $\Delta\delta^{15}N_{max}(t)$ and $\Delta\delta^{15}N_{min}(t)$ are calculated. Next, $\Delta\delta^{15}N_{max}(t)$ and $\Delta\delta^{15}N_{min}(t)$ are inverted. That means for $\Delta\delta^{15}N_{max}(t)$ that the values are shifted back (-lag max) and shifted further to lag min. After inverting, $\Delta\delta^{15}N_{max}(t)$ and $\Delta\delta^{15}N_{min}(t)$ are summed-up componentwise to calculate $\Delta\delta^{15}N_{cv}(t)$. Using $\Delta\delta^{15}N_{cv}(t)$ in eq. (3) leads to the temperature-correction values which are added to the temperature $T_{hf}$.

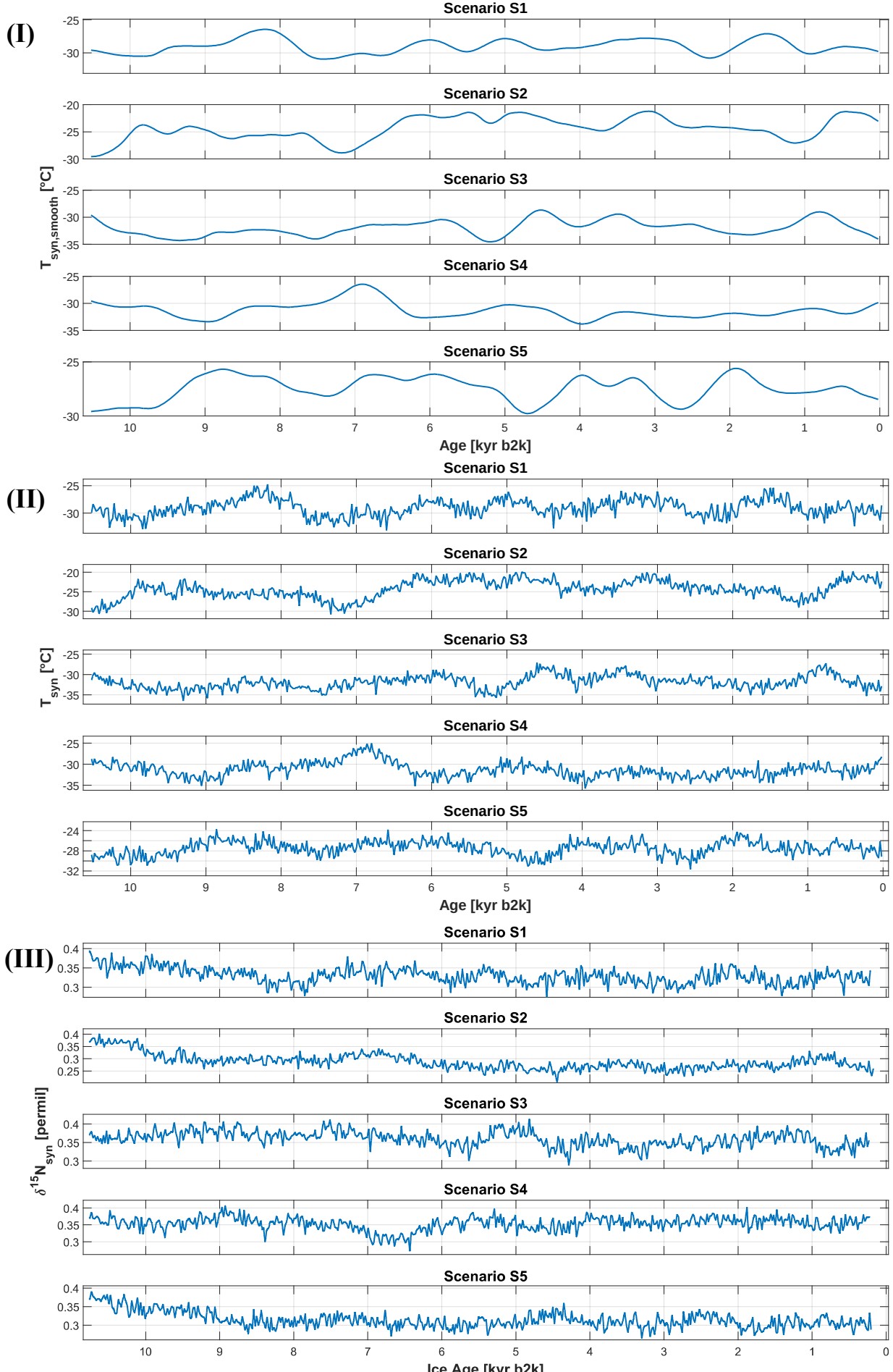

**Figure 04: (I)** S1-S5: Synthetic smooth temperature-scenarios for the construction of the target-temperature data. **(II)** S1-S5: Synthetic target surface-temperature scenarios. **(III)** S1-S5: Corresponding synthetic δ¹⁵N target time-series.

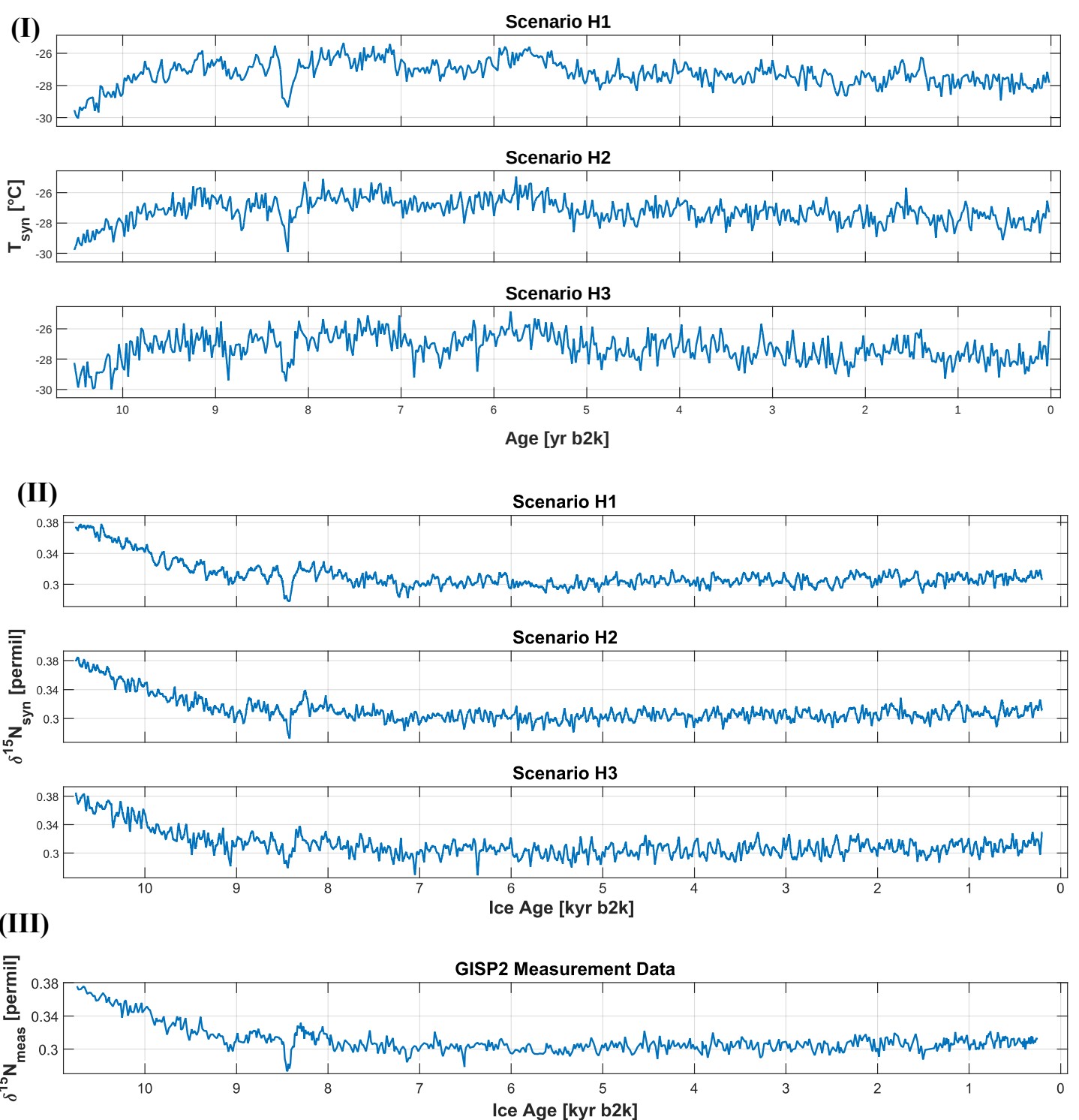

**Figure 05: (I)** Synthetic target surface-temperature scenarios H1-H3. **(II)** Corresponding synthetic δ¹⁵N target time-series H1-H3. **(III)** GISP2 δ¹⁵N measured data (Kobashi et al., 2008).

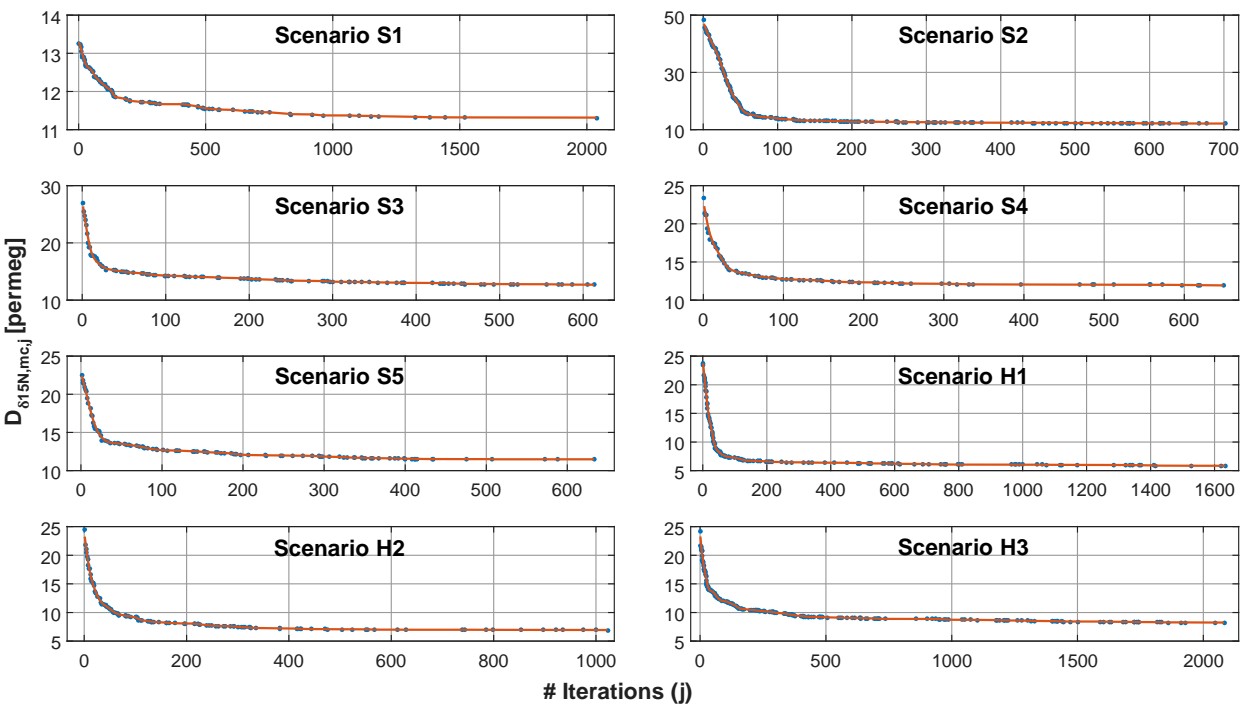

**Figure 06: Evolution of the mean misfit ($D_{\delta15N,mc}$) of the modelled $\delta^{15}N_{mc}$ vs. synthetic $\delta^{15}N_{syn}$ target as function of the number of iterations (j) for the Monte-Carlo approach for all synthetic target scenarios.**

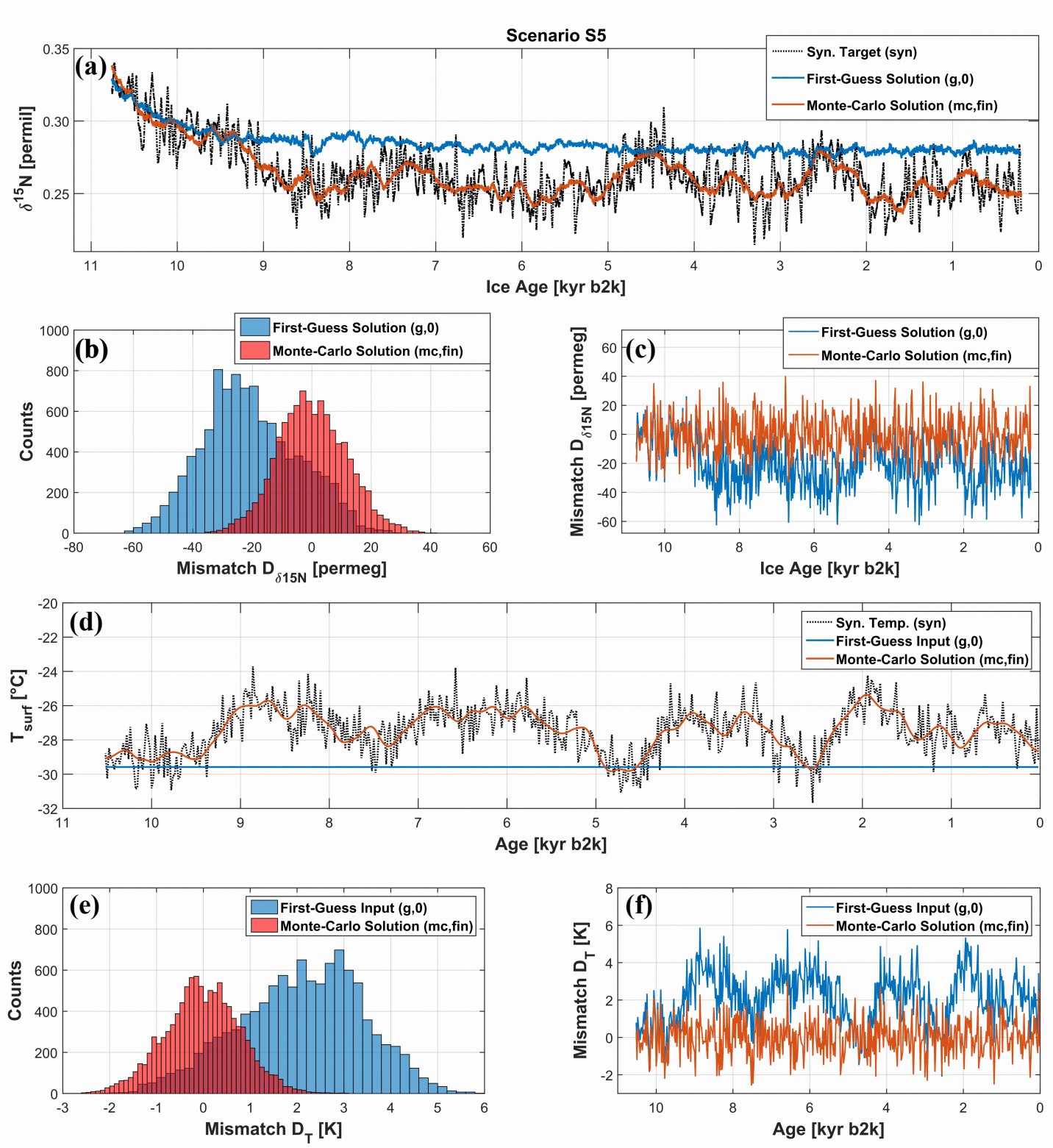

**Figure 07: (a-c)** First-guess (g,0) vs. Monte-Carlo (mc,fin) δ¹⁵N-solution for the scenario S5: **(a)** Synthetic $\delta^{15}N_{syn}$ target (black dotted line), modelled δ¹⁵N time series for the first-guess input (blue line) and Monte-Carlo solution (red line). **(b)** Histogram shows the pointwise mismatches of $D_{\delta15N}$ for the first guess solution (blue) and the Monte Carlo solution (red) versus the synthetic target. **(c)** Time-series for the pointwise mismatches of $D_{\delta15N}$ for the first-guess solution (blue) and the Monte-Carlo solution (red) versus the synthetic target. **(d-f)** First-guess vs. Mont-Carlo surface-temperature solution $T_{surf}$ for the scenario S5: **(d)** Synthetic surface-temperature target $T_{syn}$ (black dotted line), first-guess-temperature input (blue line) and Monte-Carlo solution (red line). **(e)** Histogram shows the pointwise temperature mismatches for $D_T$ for the first-guess solution (blue) and the Monte-Carlo solution (red) versus the synthetic surface-temperature target. **(f)** Time-series for the pointwise temperature mismatches for $D_T$ for the first-guess solution (blue) and the Monte-Carlo solution (red) versus the synthetic surface-temperature target.

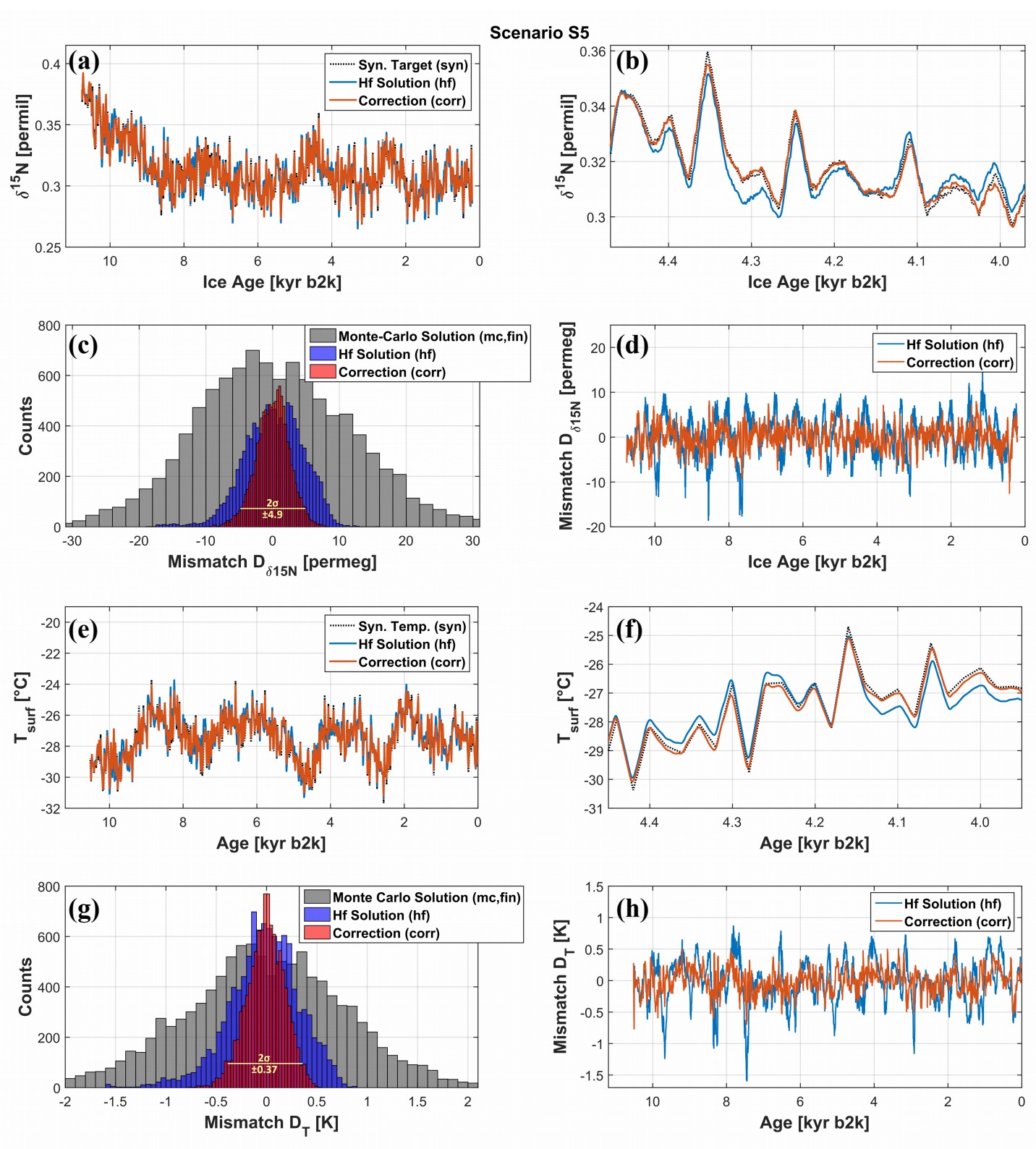

**Figure 08:** **(a-d) δ¹⁵N:** **(a)** Synthetic $\delta^{15}N_{syn}$ target (black dotted line), modelled $\delta^{15}N$ time-series after adding high-frequency information (hf, blue line) and correction (corr, red line) for the scenario S5. **(b)** Zoom-in for a randomly chosen 5 yr interval shows the decrease of the mismatch after the correction compared to the high-frequency solution. **(c)** Histogram shows the pointwise mismatches for $D_{\delta15N}$ of the synthetic $\delta^{15}N_{syn}$ target vs. the Monte-Carlo solution (mc,fin; grey), the high-frequency solution (hf; blue) and the correction (corr; red). The 95 % quantile is 4.9 permeg (yellow line) and used as an estimate for 2σ uncertainty of the final solution. **(d)** Time-series for the pointwise mismatches for $D_{\delta15N}$ of the synthetic $\delta^{15}N_{syn}$ target vs. the high-frequency solution (hf; blue) and the correction (corr; red). **(e-h) temperature:** **(e)** Synthetic temperature target $T_{surf}$ (black dotted line), modelled temperature time-series after adding high-frequency information (hf; blue line) and correction (corr; red line). **(f)** Zoom-in for a randomly chosen 5 yr interval shows the decrease of the mismatch after the correction compared to the high-frequency solution. **(g)** Histogram shows the pointwise mismatches for $D_T$ of the synthetic temperature target $T_{syn}$ vs. the Monte-Carlo solution (mc,fin; grey), the high-frequency solution (hf; blue) and the correction (corr; red). The 95 % quantile is 0.37 K (yellow line) and used as an estimate for 2σ uncertainty of the final solution. **(h)** Time-series for the pointwise mismatches for $D_T$ of the synthetic temperature target $T_{syn}$ vs. the high-frequency solution (hf; blue) and the correction (corr; red).

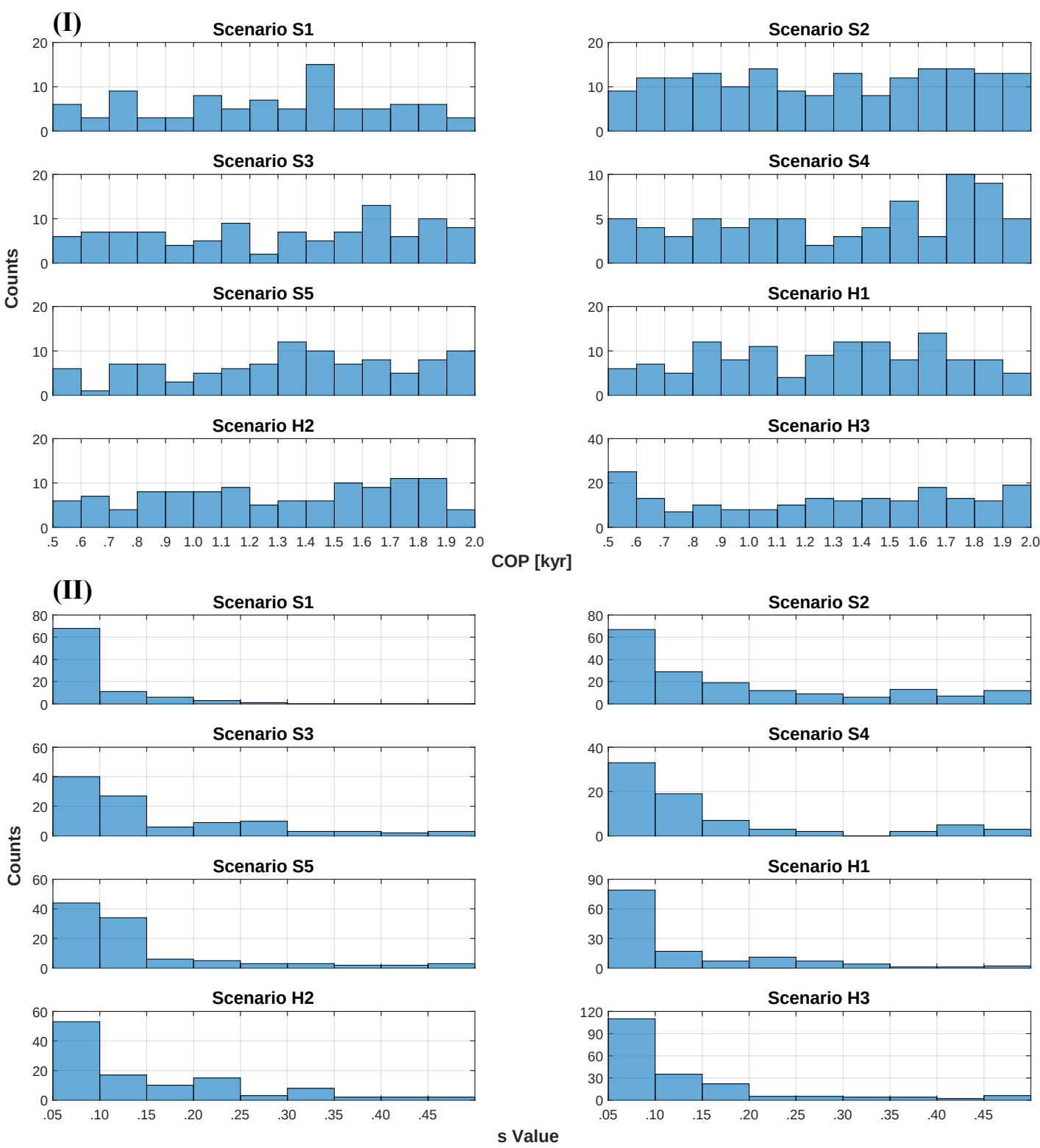

**Figure 09: (I) Counts of the cut-off-periods (COP) and (II) counts of the s values used to create the improvements for the smooth temperature solutions of the Monte-Carlo input generator for all synthetic scenarios (S1-S5 and H1-H3). A s value of 0.1 for example means that the maximum allowed perturbation of one temperature value T(t₀) is ±10 %.**

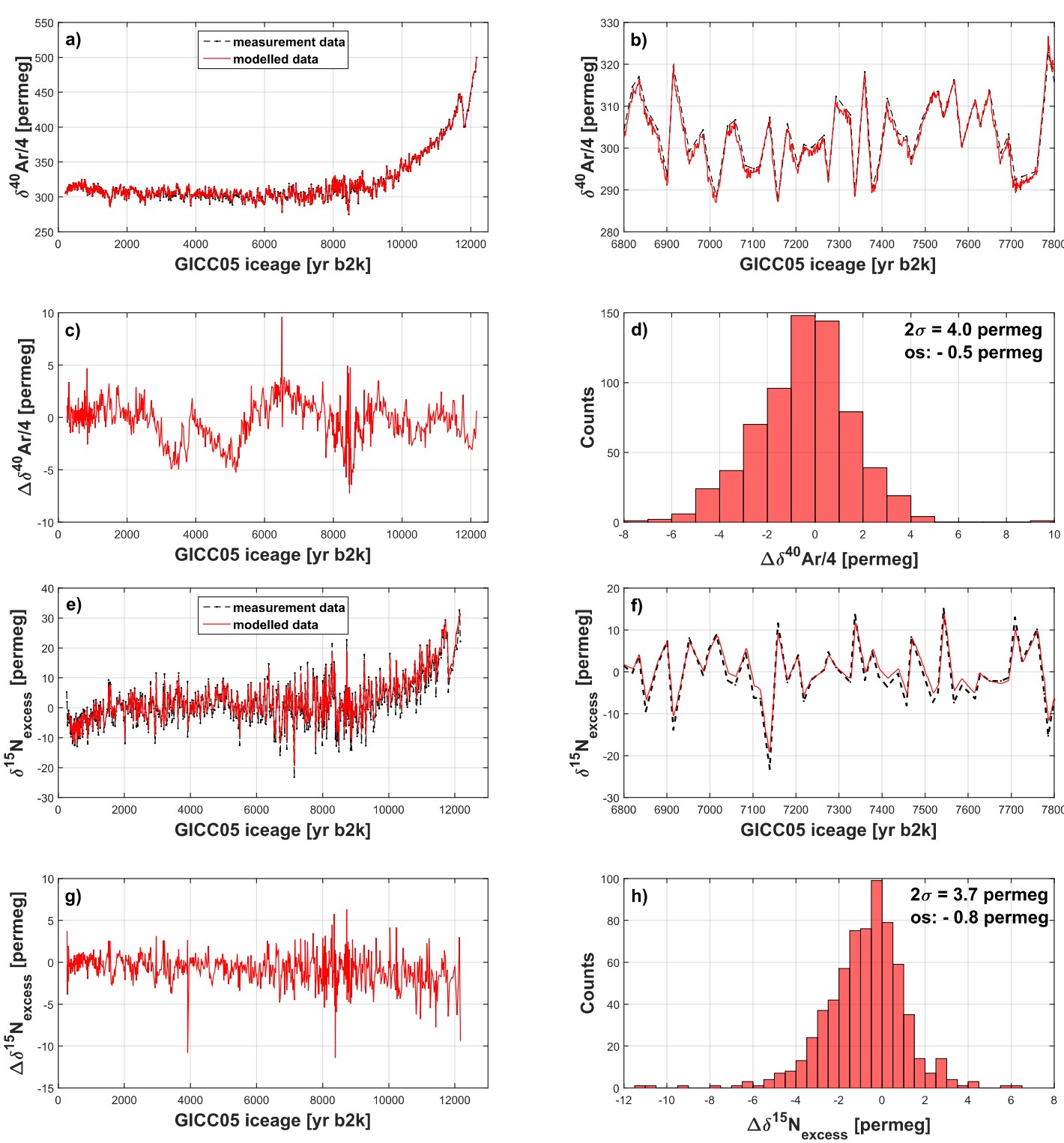

**Figure 10: Fitting of GISP2 Holocene δ⁴⁰Ar (a-d) and δ¹⁵N_excess (e-h) data (measurement data from Kobashi et al., 2008):** (a) measured vs. modelled δ⁴⁰Ar/4 time-series. (b) Zoom-in for the same quantity as in (a). (c) Time-series of the final mismatches Δδ⁴⁰Ar/4 of the measured minus the modelled δ⁴⁰Ar/4 data. (d) Histogram for the same quantity as in (c) showing an overall final mismatch (2σ) of 4.0 permeg and offset (os) of -0.5 permeg. (e) Measured vs. modelled δ¹⁵N_excess time-series. (f) Zoom-in for the same quantity as in (e). (g) Time-series of the final mismatches Δδ¹⁵N_excess of the measured minus the modelled δ¹⁵N_excess data. (h) Histogram for the same quantity as in (g) showing an overall final mismatch (2σ) of 3.7 permeg and offset (os) of -0.8 permeg.

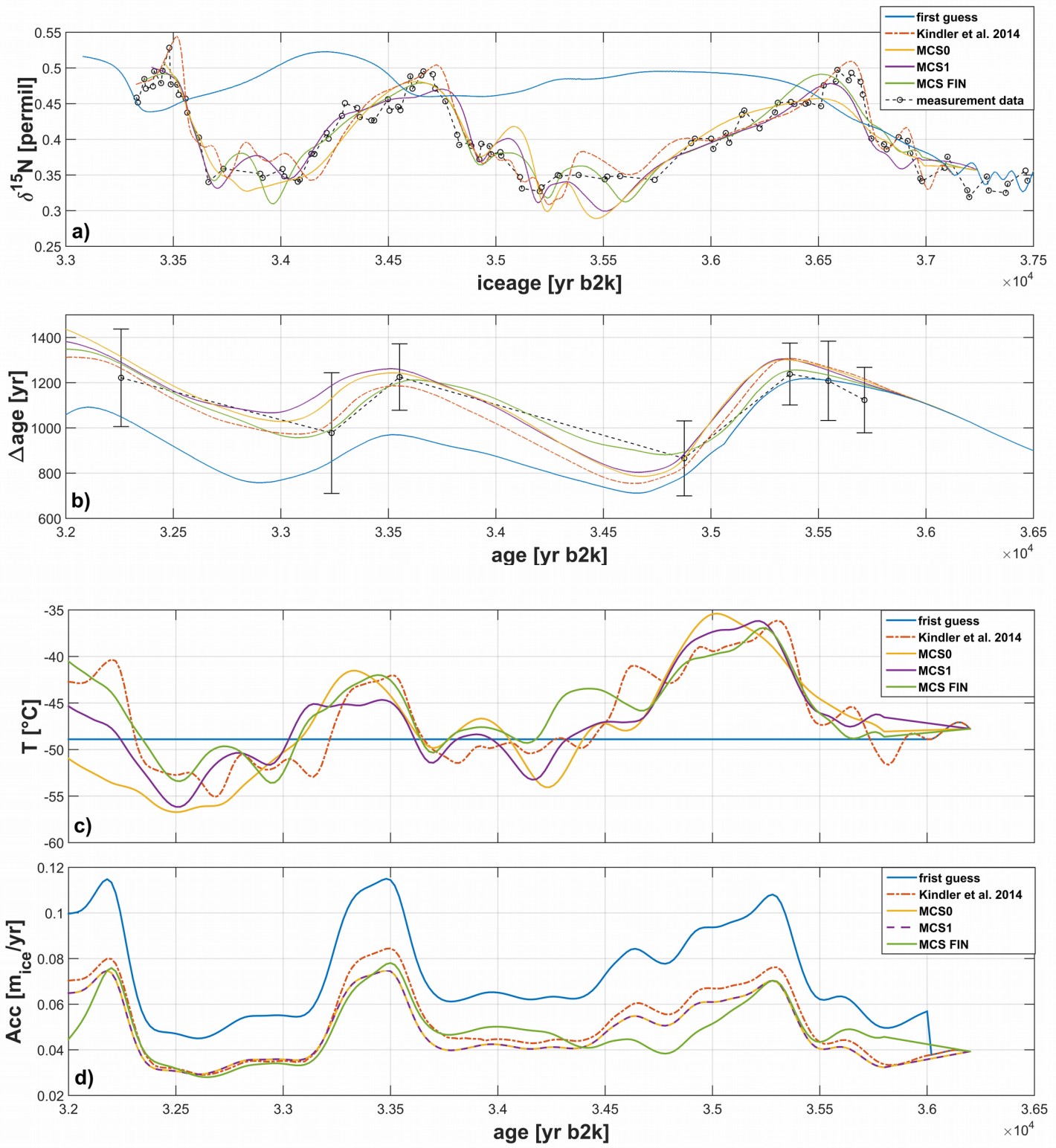

**Figure 11: Proof-of-concept for glacial reconstructions (NGRIP DO6 and DO7): (a) δ¹⁵N target plot: δ¹⁵N model output for the first-guess input (blue line), Kindler et al. (2014) fit (orange dotted line), Monte-Carlo solution 0 (yellow line, unpublished data), Monte-Carlo solution 1 (purple line, unpublished data), final Monte-Carlo solution (green line, unpublished data), δ¹⁵N measurement target (black dotted line, measurement points are black cycles, data from Kindler et al., 2014). (b) Δage target plot: Δage model output for the first-guess input (blue line), Kindler et al. (2014) fit (orange dotted line), Monte-Carlo solution 0 (yellow line, unpublished data), Monte-Carlo solution 1 (purple line, unpublished data), final Monte-Carlo solution (green line, unpublished data), Δage measurement target (black dotted line, measurement points are black cycles, data from Guillevic, M., 2013). (c) Temperature solution plot: first-guess input (blue line), Kindler et al. (2014) solution (orange dotted line), Monte-Carlo solution 0 (yellow line, unpublished data), Monte-Carlo solution 1 (purple line, unpublished data), final Monte-Carlo solution (green line, unpublished data); (d) Accumulation-rate solution plot: first-guess input (blue line), Kindler et al. (2014) solution (orange dotted line), Monte-Carlo solution 0 (yellow line, unpublished data), Monte-Carlo solution 1 (purple line, unpublished data), final Monte-Carlo solution (green line, unpublished data).**

| Variable | Explanation |
|---|---|
| $\alpha_T$ | thermal-diffusion constant calculated from eq. (12) |
| $\alpha_{18O}$ | slope for $\delta^{18}O_{ice}$ calibration (surface-temperature spin-up), eq. (13) |
| Acc | accumulation-rate data |
| $Acc_{g,0}$ | first-guess (prior) input accumulation-rate data |
| $Acc_{mc,fin}$ | modelled accumulation-rate data from the final Monte-Carlo output |
| $\beta$ | intercept for $\delta^{18}O_{ice}$ calibration (surface-temperature spin-up), eq. (13) |
| COP | cut-off-period for cubic-spline-filtering |
| corr | index related to the final correction step (step 4) |
| CZ | convective zone |
| D | mean mismatch (general) calculated from eq. (1) |
| $D(t), D_i$ | pointwise mismatches (general) |
| $D_{\delta15N}$ | mean mismatch of $\delta^{15}N$ |
| $D_{\delta15N,g,0}$ | mean mismatch of $\delta^{15}N$ ($\delta^{15}N_{syn}$ vs. $\delta^{15}N_{g,0}$) calculated from the output of the first-guess data |
| $D_{\delta15N,hf}$ | mean mismatch of $\delta^{15}N$ ($\delta^{15}N_{syn}$ vs. $\delta^{15}N_{hf}$) calculated from the output of the high-frequency step (step 3) |
| $D_{\delta15N,mc,fin}$ | mean mismatch of $\delta^{15}N$ ($\delta^{15}N_{syn}$ vs. $\delta^{15}N_{mc,fin}$) calculated from the final Monte-Carlo output (step 2) |
| $D_{gl}$ | minimization-criterion for the prove-of-concept on glacial data as used in eq. (14) |
| $D_T$ | mean mismatch of temperature |
| $D_{T,corr}$ | mean mismatch of temperature ($T_{syn}$ vs. $T_{corr}$) calculated from the output of the final correction (step 4) |
| $D_{T,g,0}$ | mean mismatch of temperature calculated from $T_{syn}$ vs. $T_{g,0}$ |
| $D_{T,hf}$ | mean mismatch of temperature ($T_{syn}$ vs. $T_{hf}$) calculated from the output of the high-frequency step (step 3) |
| $D_{T,mc}$ | mean mismatch of temperature calculated from the final output of the Monte-Carlo step (step 2) |
| $\delta^{15}N_{grav}$ | gravitational component of the $\delta^{15}N$ signal |
| $\delta^{15}N_{g,0}$ | modelled $\delta^{15}N$ signal from the output of the first-guess data (step 1) |
| $\delta^{15}N_{hf}$ | modelled $\delta^{15}N$ signal from the output of the high-frequency step (step 3) |
| $\delta^{15}N_{mc,fin}$ | modelled (smooth) $\delta^{15}N$ signal from the final Monte-Carlo output (step 2) |
| $\delta^{15}N_{mod}$ | modelled $\delta^{15}N$ signal (general) |
| $\delta^{15}N_{therm}$ | thermal-fractionation / thermal-diffusion component of the $\delta^{15}N$ signal |
| $\delta^{15}N_{syn}$ | synthetic $\delta^{15}N$ target (fitting target) |
| $\Delta age$ | gas-ice-age difference |
| $\Delta age_{mc,fin}$ | final gas-ice-age output from the Monte-Carlo step (step2) |
| $\Delta\delta^{15}N_{cv}$ | $\delta^{15}N$ correction values calculated from $\Delta\delta^{15}N_{max}$ and $\Delta\delta^{15}N_{min}$ |
| $\Delta\delta^{15}N_{max}$ | $\delta^{15}N$ correction values calculated from the linear dependency of $xcf_{max,\delta15N}$ |
| $\Delta\delta^{15}N_{min}$ | $\delta^{15}N$ correction values calculated from the linear dependency of $xcf_{min,\delta15N}$ |
| $\Delta m$ | molar mass-difference between the heavy and light isotopes |
| $\Delta T$ | high-frequency temperature signal obtained from eq. (3) (step 3) |

| Variable | Explanation |
| --- | --- |
| $\varepsilon_{\delta^{15}N}$ | uncertainty of the $\delta^{15}N$ data as used in eq. (14) |
| $\varepsilon_{\Delta age}$ | uncertainty of the $\Delta age$ data as used in eq. (14) |
| $g$ | gravitational acceleration |
| $g,0$ | index related to the first-guess (prior) data (step 1) |
| $hf$ | index related to the high-frequency step (step 3) |
| $i$ | time index |
| $IF$ | "integrated factor" calculated from eq. (6), needed for the final correction step (step 4) |
| $j$ | running index for the Monte-Carlo iterations (step 2) |
| $lag_{max}$ | time-lag attributed to the maximum of the sample-cross-correlation-function (xcf), (general) |
| $lag_{max,\delta 15N}$ | time-lag attributed to the maximum of the sample-cross-correlation-function (xcf) of $IF(t)$ vs. $D_{\delta 15N,hf}(t)$ |
| $lag_{max,T}$ | time-lag attributed to the maximum of the sample-cross-correlation-function (xcf) of $IF(t)$ vs. $D_{T,hf}(t)$ |
| $lag_{min}$ | time-lag attributed to the minimum of the sample-cross-correlation-function (general) |
| $lag_{min,\delta 15N}$ | time-lag attributed to the minimum of the sample-cross-correlation-function of $IF(t)$ vs. $D_{\delta 15N,hf}(t)$ |
| $lag_{min,T}$ | time-lag attributed to the minimum of the sample-cross-correlation-function of $IF(t)$ vs. $D_{T,hf}(t)$ |
| $mc$ | index related to the Monte-Carlo step (step 2) |
| $mc,fin$ | index related to the final Monte-Carlo output (step 2) |
| $n$ | number of data points of the target |
| $n_{mc}$ | length of the Holocene temperature vectors (w/o spin-off) |
| $\Omega_{N_2,i}$ | thermal-diffusion sensitivity calculated from eq. (4) |
| $\vec{P}_j$ | spline-filtered $\vec{P}_{r,j}$ |
| $\vec{P}_{r,j}$ | vector containing $n_{mc}$ uniformly-distributed random numbers |
| $R$ | ideal gas-constant |
| $\rho_{ice}$ | ice density |
| $\rho_{LID}$ | lock-in-density, density threshold for calculating $z_{LID}$ |
| $s$ | standard deviation of the random numbers for $\vec{P}_{r,j}$ |
| $\sigma_{\delta 15N,corr}$ | standard deviation of $D_{\delta 15N,corr}(t)$ $(= D_{\delta 15N,corr,i})$ |
| $2\sigma_{\delta 15N,corr,95}$ | 95 % quantile of $D_{\delta 15N,corr}(t)$ $(= D_{\delta 15N,corr,i})$ |
| $\sigma_{\delta 15N,hf}$ | standard deviation of $D_{\delta 15N,hf}(t)$ $(= D_{\delta 15N,hf,i})$ |
| $\sigma_{T,corr}$ | standard deviation of $D_{T,corr}(t)$ $(= D_{T,corr,i})$ |
| $2\sigma_{T,corr,95}$ | 95 % quantile of $D_{T,corr}(t)$ $(= D_{T,corr,i})$ |
| $\sigma_{T,hf}$ | standard deviation of $D_{T,hf}(t)$ $(= D_{T,hf,i})$ |
| $\overline{T}, \overline{T}_{firn}$ | mean firn temperature |
| $T_{bottom}$ | temperatures at the bottom of the diffusive firn-layer |
| $T_{corr}$ | temperature signal calculated from the final correction step (step 4) |
| $T_{g,0}$ | first-guess (prior) temperature input |
| $T_{hf}$ | temperature signal calculated from the high-frequency step (step 3) |
| $T_{mc,j}$ | Monte-Carlo temperature guess for iteration $j$ |

| Variable | Explanation |
|---|---|
| $T_{mc,fin}$ | (smooth) temperature modelled from the final Monte-Carlo output (step 2) |
| $T_{spin}$ | surface-temperature spin-up |
| $T_{surf}$ | temperatures at the top of the diffusive firn-layer |
| wRMSE | mean-squared-errors weighted with data uncertainty as used in eq. (14) |
| xcf / XCF | sample-cross-correlation-function, needed for the final correction step (step 4) |
| $xcf_{max}$ | maximum of the sample-cross-correlation-function (general) |
| $xcf_{max,\delta15N}$ | maximum of the sample-cross-correlation-function of IF(t) vs. $D_{\delta15N,hf}(t)$ |
| $xcf_{max,T}$ | maximum of the sample-cross-correlation-function of IF(t) vs. $D_{T,hf}(t)$ |
| $xcf_{min}$ | minimum of the sample-cross-correlation-function |
| $xcf_{min,\delta15N}$ | minimum of the sample-cross-correlation-function of IF(t) vs. $D_{\delta15N,hf}(t)$ |
| $xcf_{min,T}$ | minimum of the sample-cross-correlation-function of IF(t) vs. $D_{T,hf}(t)$ |
| $X_{mod}$ | modelled data (general), can be $\delta^{15}N$, T or measured data ($\delta^{40}Ar$, $\delta^{15}N_{exc}$) |
| $X_{target}$ | fitting target (general), can be synthetic $\delta^{15}N_{syn}$, $T_{syn}$, or measured data ($\delta^{40}Ar$, $\delta^{15}N_{exc}$) |
| $z_{LID}$, LID | lock-in-depth |

**Table 01: Used variables and acronyms with their explanations.**

| Scenario | S1 | S2 | S3 | S4 | S5 | H1 | H2 | H3 |
|---|---|---|---|---|---|---|---|---|
| $D_{\delta15N,g,0}$ [permeg] | 13.3 | 48.4 | 27.0 | 23.3 | 22.4 | 23.8 | 24.1 | 23.8 |
| $D_{\delta15N,mc,fin}$ [permeg] | 11.3 | 12.4 | 12.7 | 11.9 | 11.5 | 5.8 | 6.9 | 8.2 |
| $\delta^{15}N$ Improvement | 2.0 | 36.0 | 14.3 | 11.4 | 10.9 | 18.0 | 17.2 | 15.6 |
| [permeg \| %] | 15.0 | 74.4 | 53.0 | 48.9 | 48.7 | 75.6 | 71.4 | 65.5 |
| # improvements | 119 | 351 | 152 | 108 | 174 | 223 | 173 | 325 |
| # used improvements | 89 | 174 | 103 | 74 | 102 | 129 | 112 | 193 |
| # iterations | 2103 | 706 | 620 | 656 | 637 | 1636 | 1027 | 2086 |
| # tried solutions | 16824 | 5648 | 4960 | 5248 | 5096 | 13088 | 8216 | 16688 |
| Execution time [h] | 52.6 | 17.7 | 15.5 | 16.4 | 15.9 | 40.9 | 25.7 | 52.2 |
| $D_{T,g,0}$ [K] | 1.24 | 5.24 | 2.45 | 2.09 | 2.17 | 2.34 | 2.38 | 2.32 |
| $D_{T,mc}$ [K] | 0.61 | 0.69 | 0.70 | 0.64 | 0.64 | 0.32 | 0.39 | 0.46 |
| Temp. Improvement | 0.63 | 4.55 | 1.75 | 1.45 | 1.53 | 2.02 | 1.99 | 1.86 |
| [K \| %] | 50.8 | 86.8 | 71.4 | 69.4 | 70.5 | 86.3 | 83.6 | 80.2 |

Table 02: Summary for the Monte-Carlo approach: Mismatch $D_{g,0}$ between the modelled $\delta^{15}N$ (or temperature) values using the first-guess input and the synthetic $\delta^{15}N$ (or temperature) target for each scenario. $D_{mc}$ is the mismatch between the modelled $\delta^{15}N$ (or temperature) using the final Monte-Carlo temperature solution and the synthetic $\delta^{15}N$ (or temperature) target for each scenario.

| Scenario: | COP [yr] | s |
|---|---|---|
| S1 | 1135 | 0.2065 |
| S2 | 1007 | 0.3967 |
| S3 | 1177 | 0.4002 |
| S4 | 1315 | 0.2952 |
| S5 | 1244 | 0.2388 |

Table 03: Cut-off-periods (COP) and s values used for creating the smooth synthetic temperature scenarios according to the Monte-Carlo approach.

| Scenario | S1 | S2 | S3 | S4 | S5 | H1 | H2 | H3 |
|---|---|---|---|---|---|---|---|---|
| $D_{\delta15N,hf}$ [permeg] | 2.7 | 3.6 | 4.3 | 3.2 | 3.5 | 2.1 | 2.5 | 2.6 |
| Improvement (hf vs. MC) [%] | 76.1 | 71.0 | 66.1 | 73.1 | 69.6 | 63.8 | 63.8 | 68.3 |
| $\sigma_{\delta15N,hf}$ [permeg] | 3.5 | 4.6 | 5.4 | 4.0 | 4.3 | 2.7 | 3.1 | 3.3 |
| $D_{\delta15N,corr}$ [permeg] | 1.7 | 2.1 | 2.6 | 1.9 | 2.0 | 1.2 | 1.3 | 1.6 |
| Improvement (corr vs. hf) [%] | 37.0 | 41.7 | 39.5 | 40.6 | 42.9 | 42.9 | 48.0 | 38.5 |
| $\sigma_{\delta15N,corr}$ [permeg] | 2.2 | 2.7 | 3.3 | 2.4 | 2.5 | 1.5 | 1.7 | 1.9 |
| $2\sigma_{\delta15N,corr,95}$ [permeg] | **4.4** | **5.3** | **6.3** | **4.7** | **4.9** | **3.0** | **3.4** | **3.7** |
| $D_{T,hf}$ [K] | 0.20 | 0.32 | 0.33 | 0.25 | 0.27 | 0.18 | 0.21 | 0.22 |
| $\sigma_{T,hf}$ [K] | 0.26 | 0.40 | 0.43 | 0.32 | 0.35 | 0.22 | 0.26 | 0.27 |
| $D_{T,corr}$ [K] | 0.12 | 0.18 | 0.20 | 0.14 | 0.15 | 0.10 | 0.11 | 0.12 |
| $\sigma_{T,corr}$ [K] | 0.15 | 0.24 | 0.25 | 0.19 | 0.19 | 0.12 | 0.14 | 0.15 |
| $2\sigma_{T,corr,95}$ [K] | **0.31** | **0.48** | **0.51** | **0.38** | **0.37** | **0.23** | **0.27** | **0.30** |

**Table 04: Summary for the high-frequency (hf) and correction part (corr) of the reconstruction approach. D is the mean mismatch between the modelled $\delta^{15}$N (or temperature) data versus the synthetic $\delta^{15}$N (or temperature) target. σ is the standard deviation of the pointwise mismatches $D_i$. The 95 % quantiles ($2\sigma_{\delta15N,corr,95}$ or $2\sigma_{T,corr,95}$) of the pointwise $\delta^{15}$N (or temperature) mismatches are used as an estimate for the 2σ uncertainty for the final solution.**

| Scenario: | 2σ Δ(Δage) [yr] | Scenario: | 2σ Δ(Δage) [yr] |
|---|---|---|---|
| **S1** | 1.14 | **S5** | 1.24 |
| **S2** | 1.60 | **H1** | 1.23 |
| **S3** | 1.98 | **H2** | 1.18 |
| **S4** | 1.41 | **H3** | 1.30 |

**Table 05: Final mismatches Δ(Δage) (2σ) of Δage between the corrected solution and the synthetic targets for all scenarios.**

| Solution | $D_{gl}$ | Mismatch $\delta^{15}$N (2σ) [permeg] | Mean mismatch $\delta^{15}$N* [permeg] | Mismatch Δage (2σ) [yr] | Mean mismatch Δage* [yr] |
|---|---|---|---|---|---|
| **Kindler 2014** | 3.6 | 44.5 | 17.9 | 256 | 101 |
| | | | | | |
| **first guess** | 7.8 | 128.7 | 63.8 | 328 | 138 |
| **MCS0** | 3.1 | 50.0 | 19.3 | 199 | 82 |
| **MCS1** | 2.9 | 44.3 | 17.6 | 200 | 84 |
| **MCS FIN** | 2.6 | 37.8 | 15.6 | 175 | 63 |

**Table 06: Proof-of-concept for Glacial reconstruction: $D_{gl}$ is the used minimization criterion (see Sect. 4.4).**
**\*The mean mismatches for $\delta^{15}$N and Δage were calculated according to eq. (1).**