# Peer review of "Novel automated inversion algorithm for temperature reconstruction using gas isotopes from ice cores"

_Climate of the Past, 2017_

## Referee Comment (RC1) · Anonymous Referee #1 · 17 Jul 2017

This work describes a technical and mathematical variation on previously published work by Kobashi et al. (2010; 2011; 2012). This earlier work used nitrogen isotopes of N2 trapped in air bubbles in Greenland ice during the Holocene to reconstruct surface temperature, which gives complementary information to the classical oxygen-isotope paleothermometer. The added value of the gas isotope work is that the physics underlying this proxy is simpler than the classical oxygen-isotope method, being based on the fundamental physics of gaseous thermal diffusion fractionation. This well-known phenomenon causes heavy isotopes to migrate to colder regions in the firn layer on top of polar ice sheets, leaving a measurable signal in the air being trapped in bubbles at the base of the firn. Kobashi also used argon isotopes, measured in the same ice

samples as the nitrogen isotopes. Because argon and nitrogen have constant atmospheric isotope ratios over the time scales considered here, the only processes that affect their isotopic composition in trapped air are those processes such as thermal diffusion that occur in the firn column.

One of the challenges that Kobashi et al. faced was translating the observed gas isotope data into a surface temperature history. Kobashi et al. innovated by creating a novel hybrid firn densification-thermal diffusion model, much the same as is done here. The new work is different in that it uses an inverse Monte Carlo sampling and cubic spline filtering instead of Kobashi et al.'s iterative forward-model step-and-adjust method. In that sense it is likely an improvement mathematically over Kobashi et al.'s work.

The scientific advance represented by this work is useful but is very incremental, almost to the point of not standing alone as a publishable scientific paper. It is not clear to me that this work suffices as a "Least Publishable Unit", and reads more like an Appendix to a publication.

It would improve the paper if Kobashi's work could be compared to the results found here, and placed in a larger context. Additionally, it would be helpful if the present work were actually used to reconstruct Greenland temperature over the Holocene, much as Kobashi et al. did. I am somewhat surprised that Kobashi is not a co-author, considering how heavily this work relies on Kobashi et al.'s prior work. The synthetic data looks a lot like Kobashi et al.'s actual data. Why not show the actual data?

If this change is made, then the reader could judge for herself/himself how much of an impact the new inversion methodology has made. Since this journal concerns climate, it would be much more satisfying for the reader if the actual improvement in climate reconstruction is shown.

Minor comments:

[Figure]

page 1 line 11 "The presented approach is completely automated..."

page 1 line 23 "since it represents a time of moderate natural variations prior to anthropogenic disturbance, often referred to as a baseline...."

page 2 line 6 "The studies of Dahl-Jensen et al. (1998) and Cuffey et al. (1995; 1997) demonstrate the usefulness of inverting the measured borehole temperature profile for surface temperature history....."

page 2 line 9 " unable to resolve..."

page 3 line 24. It is not clear from the wording here which thermal diffusion sensitivity value was used here. Is it the Grachev and Severinghaus (2003), or the Leuenberge et al. (1999)? This must be clarified.

A separate issue is that the Leuenberger et al. value is based on measurements that were made in pure nitrogen, not in air. It is well known, and indeed predicted from theory, that the thermal diffusion sensitivity (and thermal diffusion factor) is larger in pure gases than in air. For example, Grachev and Severinghaus measured these parameters in both pure N2 and in air, and found a substantial difference between the two (Figure 1). As can be seen in Figure 1, the thermal diffusion factor in pure N2 is 0.0037 whereas in air it is less than 0.0036. Even more troubling is the fact that the 1960s-era measurements made in pure N2 by the sources that Leuenberger et al. use disagree well outside the analytical error (0.0035) with the pure-N2 value of Grachev and Severinghaus, which was made with a modern mass spectrometer. This suggests that the 1960s era measurements by Boersma-Klein and De Vries (1966) were badly in error. Given the primitive technology of that time, this is not a criticism of these workers, but it is clear that their values should not be used for the present study.

page 3 line 26 "The firn model used here behaves purely as a forward model,....."

page 4 line 3 You must say which ice core was used here. Is it GISP2?

[Figure]

**Thermal diffusion factor of $^{15}$N of N$_2$ in air**

Fig. 1. Comparison of thermal diffusion factor measurements

---

## Short Comment (SC1) · 27 Jul 2017

The submitted paper by Döring and Leuenberger presents an new fitting approach to model $\delta^{15}$N time series as measured in air extracted from ice cores, together with the corresponding temperature history. This fitting approach is combined with an existing firn model (Schwander et al., 1997) run in a forward mode. First, a synthetic dataset mimicking Holocene $\delta^{15}$N and temperature is constructed. Then, the three steps of the fitting method are described. At each step of the procedure the accuracy compared to the synthetic dataset, used as reference value, is estimated. The final distribution of all distances to the synthetic dataset is used to estimate a 95% confidence interval (or 2

$\sigma$ uncertainty estimate). It is worth mentioning that this method can also be adapted to invert other parameters such as $\delta^{40}$Ar or $\delta^{15}$N$_{excess}$.

In my view this paper presents clear and outstanding improvements compared to existing methods. Temperature reconstructions obtained with such a method applied to new or existing $\delta^{15}$N datasets will be very valuable in the future for the scientific community.

Four specific advantages of the new approach are discussed hereafter and compared to existing published work in this field:

1. The fitting approach requires as input measured $\delta^{15}$N data only (as well as time scale and accumulation rate, as any firn model). The reconstructed temperature is therefore entirely independent from any other temperature proxy records (in particular, water stable isotopes), allowing future unbiased intercomparison in between different reconstructions in the future;

2. An uncertainty of the fitting approach itself is provided (excluding uncertainty due to $\delta^{15}$N measurement, firn physics and firn dynamics modeling). Again this will allow for future valuable intercomparison of temperature proxies;

3. The method is entirely automated, i.e. it is completely user-independent and allows to save significant working hours;

4. The fitting approach, uncertainty estimation and automation are described in an entirely transparent and reproducible manner.

**1  Independence of the produced temperature history from any other temperature proxy records**

Döring and Leuenberger present a temperature reconstruction method requiring $\delta^{15}$N data only as input. The fitting approach is based on a randomly generated first guess

(i.e., a constant temperature) combined with randomly generated noise to progressively improve the initial guess. The only section requiring water isotope variations as input, scaled with a linear relationship, is the spin-off section of the dataset, and should therefore have only an insignificant influence on the final obtained temperature history. Therefore this method requires no prior input temperature signal nor any other assumption on the expected shape of the temperature signal. The obtained temperature history is consequently fully independent from other proxy records. Hence, this approach permits for the first time a future intercomparison of results in an unbiased manner. This is a clear improvement compared to existing work.

Many previous work relied indeed on other proxy records to generate a first guess temperature scenario, also because the fitting approach was (partly) manual and it would have been a huge effort to start from a flat temperature scenario. In many studies, water stable isotope variations measured in the same ice core were used as first input, with a linear function as scaling factor. This is likely one of the best option, as both proxies are sensitive to local, surface of the ice sheet temperature variations, with second order effects for water isotopes (such as seasonality of precipitation, temperature gradient in between surface and cloud condensation temperature, source temperature variations). This approach has been applied to fitting $\delta^{15}$N data only (e.g., Lang et al., 1999; Huber et al., 2006; Kindler et al., 2014; Guillevic et al., 2013; Buizert et al., 2014) or combined with $\delta^{40}$Ar data (e.g., Landais et al., 2004a; Capron et al., 2010), leading to the second order parameter $\delta^{15}$N$_{excess}$.

To model abrupt events, a step function as probable shape for the temperature increase/decrease has also been used instead of a linear function to water isotopes (e.g., Severinghaus et al., 1998; Severinghaus and Brook, 1999; Rosen et al., 2014), which is also an approximation, however valid for large and sharp temperature increase/decrease. Indeed changing the steepness of the increase has a non negligible effect on the $\delta^{15}$N and temperature reconstruction, a sharper increase leading to a smaller estimated amplitude for the temperature increase (e.g., Landais et al.

(2004b), Fig.3, Rosen et al. (2014), Fig. 3). Another alternative to using water isotopes, without supposing any prior shape for the temperature increase, was developed by Kobashi et al. (2008a). They used both nitrogen and argon isotope data to calculate a first firn $\Delta T$, combined with the Goujon firn model (Goujon et al., 2003) to reconstruct lock-in-depth temperature, leading finally to a surface temperature reconstruction. Orsi et al. (2014) also developed a new fitting approach using $\delta^{15}$N and $\delta^{40}$Ar, using a linearized firn model. However both approaches work only when argon isotope data are measured in addition to nitrogen, with a sufficient precision. This requires dedicated ice samples and a highly precise measurement procedure (e.g., Severinghaus et al., 2003). Such argon data are (still) not available for most of the NGRIP or NEEM ice core for example (Huber et al., 2006; Kindler et al., 2014; Guillevic et al., 2013; Rosen et al., 2014), making the approach from Döring and Leuenberger highly relevant.

Using the approach presented in Döring and Leuenberger, it would be valuable (in a future work) to try to reproduce previous results on the relationship between $\delta^{15}$N-based temperature reconstruction and water isotopes, that used water isotope variations as first guess for the temperature scenario (e.g., Landais et al., 2004a; Kindler et al., 2014; Guillevic et al., 2013; Buizert et al., 2014).

To clearly show the usefulness of this new approach, it could be helpful for the reader to highlight in the method section of this manuscript the independence of this approach from other temperature proxies, in particular from water isotope records.

**2 Uncertainty estimate of the fitting approach**

A soundly estimated uncertainty with a 95% confidence interval is provided. This is made possible by first generating a synthetic temperature scenario mimicking Holocene temperature variations, compatible with the precise, high resolution GIPS2 $\delta^{15}$N dataset published in Kobashi et al. (2008b). The constructed synthetic dataset

is used as 'true value' and the difference in between the fitting approach result and this true value is quantified for each time point of the generation. The aggregation of all distances produces a distribution of distances around the 'true value'; the limits corresponding to 95% of the distances are used as 2 $\sigma$ uncertainty estimate. To my knowledge, this is the first time the accuracy of a fitting approach for $\delta^{15}$N is tested in this way by first creating an arbitrary 'true value'. It seems an excellent method to test and quantify the uncertainty introduced by the fitting approach itself, independently of any other uncertainty (introduced by $\delta^{15}$N data resolution and precision, firn processes, accumulation rate, time scale, etc, which are not considered in this paper).

In the literature, uncertainty estimate are usually reported for the entire reconstruction as a whole, including all considered types of uncertainty. Moreover until now the uncertainty of the reconstructed temperature was usually estimated using a sensitivity study of the reconstructed temperature to various changes in input values (e.g., Landais et al., 2004b; Huber et al., 2006; Guillevic, 2013). The Monte Carlo simulation used in Buizert et al. (2014) (n.b. also modeling – part of – the GISP2 $\delta^{15}$N dataset published in Kobashi et al. (2008b), among other records) was already a highly valuable alternative method for uncertainty estimation. However, until this study from Döring and Leuenberger, using a synthetic dataset as reference value to estimate fitting approach accuracy, no uncertainty was really possible to estimate for the fitting approach itself.

In a future study applied to the glacial period (as hinted by Döring and Leuenberger), it would be valuable to test the fitting accuracy for the reconstructed delta age as well, by comparison to a synthetic delta-age obtained at the same time as synthetic $\delta^{15}$N. This is likely not highly relevant for the Holocene period where $\Delta$age remains relatively constant and small, but should be an interesting aspect to test for the glacial.

**3 Automation**

The automation presented in Döring et al. is actually certainly a prerequisite for the generation of thousands of temperature scenarii progressively modified to match $\delta^{15}$N data. Moreover, this automated fitting method is applied over a relatively long time period of ~10 kyr, and as stated by the authors could be applied to the entire last glacial period (120 kyr). The automation supersedes the manual or semi-automated fitting process, and therefore decouples the result from the potential influence of the user. This automated method is working for the reconstruction of a thousand year long temperature record where few clear and sharp temperature variations exist (except perhaps the 8.2 kyr event). So I expect that this method should work as well for the glacial, where extremely abrupt events occurred, giving a better constrain to the reconstructed temperature scenario. The authors demonstrate moreover that the successive steps of the approach, starting first by matching low frequency $\delta^{15}$N variance (expected to be caused mostly by firn thickness changes), then high frequency variations (caused by surface temperature oscillations), and finally fine tuning the timing of each temperature oscillation, is a robust method well suited to match firn dynamics history.

To my knowledge, Buizert et al. (2014) also used an automated approach applied to long dataseries, to reconstruct the temperature history of the last deglaciation, as well as gas age histories (Buizert et al., 2015; Rasmussen et al., 2013; Seierstad et al., 2014). However as stated on Sect. 1, the Buizert automated method requires input water isotope data and user-chosen points where the linear fit in between water isotopes and temperature is allowed to change.

The approach presented in Döring and Leuenberger, combining automation and independence from any other temperature proxy records, run over a long time period, can therefore be considered as a remarkable improvement.

**4 Transparency**

The approach developed in Döring and Leuenberger is described in full details together with the required input data, the different steps of the procedure and the parameters needed to be chosen/defined at the beginning of the procedure. Also the target of the modeling procedure is clearly stated, i.e. the synthetic $\delta^{15}$N dataset, as well as the testing criterion to estimate if a newly generated $\delta^{15}$N scenario better fits the synthetic target. I find excellent that effort is given to making the fitting approach completely transparent for the reader. This has not always been the case, likely because (i) at the very beginning fitting approaches were manual and therefore difficult to precisely describe, (ii) the uncertainty of the fitting approach was likely very small compared to $\delta^{15}$N analytical uncertainty, (iii) finding appropriate physical description of firn densification processes and dynamics was the most challenging scientific goal and (iv) only small datasets, covering limited time slices (usually one to two Dansgaard-Oeschger events), were considered.

I would actually recommend to add a schematic figure sketching each step of the procedure, to provide a method overview for the reader. This would help to follow the text description.

**Final comment**

In short, I consider the method presented in Döring and Leuenberger an outstanding contribution to the field of $\delta^{15}$N modeling.

**References**

Buizert, C., Cuffey, K. M., Severinghaus, J. P., Baggenstos, D., Fudge, T. J., Steig, E. J., Markle, B. R., Winstrup, M., Rhodes, R. H., Brook, E. J., Sowers, T. A., Clow, G. D., Cheng, H., Edwards, R. L., Sigl, M., McConnell, J. R., and Taylor, K. C. (2015). The WAIS Divide deep ice core WD2014 chronology – Part 1: Methane synchronization (68–31 ka BP) and the gas age–ice age difference. *Climate of the Past*, 11(2):153–173.

Buizert, C., Gkinis, V., Severinghaus, J. P., He, F., Lecavalier, B. S., Kindler, P., Leuenberger, M., Carlson, A. E., Vinther, B., Masson-Delmotte, V., White, J. W. C., Liu, Z., Otto-Bliesner, B., and Brook, E. J. (2014). Greenland temperature response to climate forcing during the last deglaciation. *Science*, 345:1177–1180.

Capron, E., Landais, A., Chappellaz, J., Schilt, A., Buiron, D., Dahl-Jensen, D., Johnsen, S. J., Jouzel, J., Lemieux-Dudon, B., Loulergue, L., Leuenberger, M., Masson-Delmotte, V., Meyer, H., Oerter, H., and Stenni, B. (2010). Millennial and sub-millennial scale climatic variations recorded in polar ice cores over the last glacial period. *Clim. Past*, 6:345–365.

Goujon, C., Barnola, J.-M., and Ritz, C. (2003). Modeling the densification of polar firn including heat diffusion: application to close-off characteristics and gas isotopic fractionation for Antarctica and Greenland sites. *J. Geophys. Res.*, 108(D24):4792.

Guillevic, M. (2013). *Characterisation of rapid climate changes through isotope analyses of ice and entrapped air in the NEEM ice core*. PhD thesis, Niels Bohr Institute, Faculty of Science, University of Copenhagen, Denmark and Université de Versailles Saint Quentin en Yvelines, France.

Guillevic, M., Bazin, L., Landais, A., Kindler, P., Orsi, A., Masson-Delmotte, V., Blunier, T., Buchardt, S. L., Capron, E., Leuenberger, M., Martinerie, P., Prié, F., and Vinther, B. M. (2013). Spatial gradients of temperature, accumulation and $\delta^{18}$O-ice in Greenland over a series of Dansgaard–Oeschger events. *Clim. Past*, 9:1029–1051.

Huber, C., Leuenberger, M., Spahni, R., Flückiger, J., Schwander, J., Stocker, T. F., Johnsen, S. J., Landais, A., and Jouzel, J. (2006). Isotope calibrated Greenland temperature record over Marine Isotope Stage 3 and its relation to CH$_4$. *Earth Planet. Sc. Lett.*, 243:504–519.

Kindler, P., Guillevic, M., Baumgartner, M., Schwander, J., Landais, A., and Leuenberger, M. (2014). Temperature reconstruction from 10 to 120 kyr b2k from the NGRIP ice core. *Clim. Past*, 10:887 – 902.

Kobashi, T., Severinghaus, J. P., and Barnola, J. M. (2008a). 4±1.5 °c abrupt warming 11,270

yr ago identified from trapped air in Greenland ice. *Earth Planet. Sc. Lett.*, 268:397–407.

Kobashi, T., Severinghaus, J. P., and Kawamura, K. (2008b). Argon and nitrogen isotopes of trapped air in the GISP2 ice core during the Holocene epoch (0–11,500 B.P.): Methodology and implications for gas loss processes. *Geochim. Cosmochim. Ac.*, 72:4675–4686.

Landais, A., Barnola, J.-M., Masson-Delmotte, V., Jouzel, J., Chappellaz, J., Caillon, N., Huber, C., Leuenberger, M., and Johnsen, S. J. (2004a). A continuous record of temperature evolution over a sequence of Dansgaard-Oeschger events during Marine Isotopic Stage 4 (76 to 62 kyr BP). *Geophys. Res. Lett.*, 31:L22211.

Landais, A., Caillon, N., Goujon, C., Grachev, A. M., Barnola, J. M., Chappellaz, J., Jouzel, J., Masson-Delmotte, V., and Leuenberger, M. (2004b). Quantification of rapid temperature change during DO event 12 and phasing with methane inferred from air isotopic measurements. *Earth Planet. Sc. Lett.*, 225:221–232.

Lang, C., Leuenberger, M., Schwander, J., and Johnsen, S. (1999). $16-30\,°C$ rapid temperature variation in central Greenland 70,000 years ago. *Science*, 286(5441):934–937.

Orsi, A. J., Cornuelle, B. D., and Severinghaus, J. P. (2014). Magnitude and temporal evolution of dansgaard–oeschger event 8 abrupt temperature change inferred from nitrogen and argon isotopes in {GISP2} ice using a new least-squares inversion. *Earth and Planetary Science Letters*, 395(0):81–90.

Rasmussen, S. O., Abbott, P. M., Blunier, T., Bourne, A. J., Brook, E., Buchardt, S. L., Buizert, C., Chappellaz, J., Clausen, H. B., Cook, E., Dahl-Jensen, D., Davies, S. M., Guillevic, M., Kipfstuhl, S., Laepple, T., Seierstad, I. K., Severinghaus, J. P., Steffensen, J. P., Stowasser, C., Svensson, A., Vallelonga, P., Vinther, B. M., Wilhelms, F., and Winstrup, M. (2013). A first chronology for the North Greenland Eemian Ice Drilling (NEEM) ice core. *Clim. Past*, 9(6):2713–2730.

Rosen, J. L., Brook, E. J., Severinghaus, J. P., Blunier, T., Mitchell, L. E., Lee, J. E., Edwards, J. S., and Gkinis, V. (2014). An ice core record of near-synchronous global climate changes at the Bølling transition. *Nat. Geosci.*, 7:459–463.

Schwander, J., Sowers, T., Barnola, J.-M., Blunier, T., Fuchs, A., and Malaizé, B. (1997). Age scale of the air in the Summit ice: Implication for glacial-interglacial temperature change. *J. Geophys. Res.*, 102(D16):19483–19493.

Seierstad, I. K., Abbott, P. M., Bigler, M., Blunier, T., Bourne, A. J., Brook, E., Buchardt, S. L., Buizert, C., Clausen, H. B., Cook, E., Dahl-Jensen, D., Davies, S. M., Guillevic, M., Johnsen, S. J., Pedersen, D. S., Popp, T. J., Rasmussen, S. O., Severinghaus, J. P., Svensson, A.,

and Vinther, B. M. (2014). Consistently dated records from the Greenland GRIP, GISP2 and NGRIP ice cores for the past 104 ka reveal regional millennial-scale $\delta^{18}$O gradients with possible Heinrich event imprint. *Quaternary Science Reviews*, 106(0):29 – 46. Dating, Synthesis, and Interpretation of Palaeoclimatic Records and Model-data Integration: Advances of the INTIMATE project(INTegration of Ice core, Marine and TErrestrial records, COST Action ES0907).

Severinghaus, J. and Brook, E. (1999). Abrupt climate change at the end of the Last Glacial Period inferred from trapped air in polar ice. *Science*, 286:930–934.

Severinghaus, J., Grachev, A., Luz, B., and Caillon, N. (2003). A method for precise measurement of argon 40/36 and krypton/argon ratios in trapped air in polar ice with applications to past firn thickness and abrupt climate change in Greenland and at Siple Dome, Antarctica. *Geochim. Cosmochim. Ac.*, 67:325–343.

Severinghaus, J., Sowers, T., Brook, E. J., Alley, R. B., and Bender, M. L. (1998). Timing of abrupt climate change at the end of the Younger Dryas interval from thermally fractionated gases in polar ice. *Nature*, 391:141–146.

---

## Referee Comment (RC2) · Anonymous Referee #2 · 7 Aug 2017

Döring and Leuenberger present a new approach aimed at reconstructing temperature variations from Greenland d15N-N2 data, which is based on inverting an existing firn densification model. They test the model using synthetic data. The method optimizes the fit to the data in several steps.

Unfortunately, both the method and the paper have severe shortcomings, that I list below.

(1) The method assumes that the forward problem (converting surface temperature to d15N) is completely described by the firn model, and that all variations in d15N can be linked 1-to-1 to past surface temperature. It is thus no surprise that they can reconstruct the original temperature very accurately, because they know the exact accumulation rates and physics of the forward model. Unfortunately, that is not at all true in the real world. The d15N is influenced by variations in convective zone thickness (the CZ is ignored here altogether), firn layering that influences the lock-in process, melt layers and wind crusts, etc. Real data (as opposed to the synthetic data used) further suffer from analytical noise in the laboratory. All these things will reduce the ability to reconstruct temperature from d15N. Furthermore, our understanding of firn densification is incomplete, with several physical models giving different results, microstructure effects not included in models, and hypothesized influences of dust softening. All these effects remain unaccounted for, which further reduces the ability to use d15N. The authors use identical firn physics in the forward and inverse models, which is an idealization that is untenable.

Several studies have shown that on the cm-scale there is much variation in parameters like d15N and CH4, reflecting a staggered trapping of gas bubbles within the firn-ice transition zone. See e.g. Etheridge et al. (1992), Rhodes et al. (2016), and Mitchell et al. (2015). This may be relevant as the sample size is typically smaller than the average layer thickness.

Gas diffusion and trapping smooths out the d15N signal, which provides a fundamental limit on the time resolution at which surface temperature is recorded and could potentially be reconstructed.

From the above it is clear to me that the precision that the authors state for their method is a meaningless number, that teaches us nothing about how well d15N can reconstruct temperature. A more interesting approach would be to include these fundamental uncertainties in a stochastic way, and see how well the method works under realistic settings. The synthetic data could e.g. be generated with a different firn physics description, and should be subject to CZ fluctuations, LIZ thickness variations and analytical noise.

[Figure]

(2) There are 2 fundamental inputs into the model, namely temperature and accumulation rate. The authors assume the latter is known with zero uncertainty (both in values and age model). This is a very unrealistic assumption. Even if the layer-count were perfect (which it is not), correcting for ice thinning has a fundamental uncertainty. Especially in the early Holocene, this can easily exceed 10%. As the method fits the d15N data, all accumulation errors are mapped into the temperature reconstruction. This is not accounted for.

Exactly for this reason, the method by Kobashi et al. uses a combination of 40Ar and 15N data to isolate the thermal component. The authors do not give any justification why that approach is abandoned. [As an aside, the authors convert the accumulation record from Cuffey et al. onto the GICC05 scale, which makes it internally inconsistent because the accumulation rate is the derivative of the age scale, so changing the age scale should change the accumulation values. Since the method is sensitive to the decadal-scale accumulation variability, it may be insufficient to use this crude approach.]

(3) The authors have no way of validating that their Delta-age is correct, which is critical to constrain the timing of climate change. In all d15N modeling studies I'm aware of, the use of d18O as a temperature template ensures that Delta-age is correct. In particular during abrupt events, the timing of gas and ice signals gives you Delta-age. This information is lost in their method, which is completely independent of d18O.

If the modeled Dage is off by 50 years (which is easy to do in Greenland, particularly during the glacial), the timing of the temperature solution is also off by 50 years. It would be interesting to run their algorithm on data from the last deglaciation, and see whether it reproduces the timing of abrupt change as seen in d18O. Because Delta-age is underconstrained, the timing of all reconstructed high-frequency temperature variations is uncertain.

(4) I am surprised the authors don't event attempt to invert the existing GISP2 data

(which are even plotted). This seems like a missed opportunity; especially given that it would allow comparison to existing reconstructions to estimate the accuracy of the method.

(5) The paper is overly long. I recommend section 2.3 be removed entirely, and other sections be shortened considerably. There are also 32 (!) figures in the manuscript, which is too many. Dividing the figures into main, appendix and supplement figures is annoying, as it requires a lot of going back and forth.

While the topic is of interest, and the method potentially interesting, I unfortunately cannot recommend publication of the work due to the severity of the flaws in the methodology, and the very limited scope of the presented work. Below are a few detailed comments should the authors decide to resubmit the manuscript elsewhere. I would recommend they first address the major comments listed above.

Page 1 Line 26: Give references for Holocene temperature reconstructions (there are many!). Page 3 Eq. (1): what about the convective zone? You should correct for that Eq. (2): The surface temperature should really be the temperature at the bottom of the convective zone where diffusion starts to dominate. This may smooth out some of the abrupt decadal-scale temperature variations. Line 17: Martinerie et al. (1994) gives the depth of the bubble close-off, whereas d15N is set at the lock-in depth instead. The LID is shallower than the COD. Is this difference accounted for, and how? Page 4 Section 2.2: what are the model parameters? What are the time and spatial step? How deep does the domain extend? What geothermal heat flux is used, etc. Section 2.3: I recommend this is removed completely. I don't see the point, especially the dynamic case which we know doesn't behave linearly due to memory effects. Page 6 Line 13: Not too robust. It'd be easy to have a 10% uncertainty in the thinning function.

---

## Short Comment (SC2) · 27 Aug 2017

The PAGES Data Stewardship Integrative Activity seeks to advance best practices for sharing data generated and assembled as part of all PAGES-related activities. A team of reviewers has been constituted for the "PAGES Young Scientists Meeting 2017" Special Issue. The data team, including the editors of the Special Issue, is reviewing the data handling within each of the CP-Discussion papers in relation to the CP data policy (https://www.climate-of-the-past.net/about/data_policy.html) and current best practices. The team is making recommendations for each paper, with the goal of achieving a high and consistent level of data stewardship across the Special Issue. We recognize

that an additional effort will likely be required to meet the high level of data steward-ship envisaged, and we appreciate the dedication and contribution of the authors. This includes the use of Data Citations (see example below). Authors are also strongly en-couraged to deposit significant code into a suitable repository and to cite it using a Data Citation.

We ask authors to respond to our comments as part of the regular open interactive dis-cussion. If you have any questions about PAGES Data Stewardship principles, please contact any of us directly. Best wishes for the success of your paper.

YSM Special Issue Data Review Team D.S. Kaufman, M.F. Loutre, M.N. Evans, S.C. Fritz, C. Tabor, H. Plumpton, R. Barnett, Y. Zhang, E. Razanatsoa, and E. Dearing Crampton Flood —

Essential additions for this paper: (1) Add a separate "Data Availability" section as required by the publisher. Specify where all of the essential input and output data are archived, including formal Data Citations for each of the datasets (see below). This includes the ice accumulation and oxygen isotope data.

(2) For essential datasets used in the study but not already in a public repository, submit the data and related metadata to an established public data repository and cite the persistent identifier in "Data Availability".

(3) Prior to publication of this study, submit the primary original data or results of nu-merical modeling to a public repository and cite the corresponding persistent identifier in "Data Availability". This includes the final time series of d15N and surface temper-atures and any other data that might be useful for future users to replicate the study outcomes and to readily compare the results with future studies. We also strongly en-courage the authors to deposit their significant code into a suitable repository and to cite it using a Data Citation. —

What is a "Data Citation"? Data Citations track the provenance of a dataset giving credit

to the data generator; this is in addition to any references to publications where the data are described. Data Citations are used in the text (or tables) alongside and in the same way as publication citations. In the Reference list, they include: Creators, Title, Repository, Identifier, Submission Year. More information about Data Citations is here: <https://www.datacite.org/mission.html> Here is an example of text and corresponding citations (using CP punctuation style):

The PAGES2k Consortium (2017a) assembled a large global dataset of temperature-sensitive proxy records (PAGES2k Consortium, 2017b). Among the records is the paleo-temperature reconstruction from Laguna Chepical (de Jong et al., 2016), which was described by de Jong et al. (2013).

References de Jong, R., von Gunten, I., Maldonado, A., and Grosjean, M.: Late Holocene summer temperatures in the central Andes reconstructed from the sediments of high-elevation Laguna Chepical, Chile (32° S), Climate of the Past, 9, 1921-1932, 2013.

de Jong, R., von Gunten, I., Maldonado, A., and Grosjean, M.: Laguna Chepical summer temperature reconstruction, World Data Center for Paleoclimatology, https://www.ncdc.noaa.gov/paleo/study/20366, 2016.

PAGES 2k Consortium: A global multiproxy database for temperature reconstructions of the Common Era, Scientific Data, 4,170088, 2017a.

PAGES 2k Consortium: A global multiproxy database for temperature reconstructions of the Common Era, version 2.0.0, figshare, https://figshare.com/s/d327a0367bb908a4c4f2, 2017b.

---

## Author Comment (AC1) · 8 Sep 2017

We have attached our replies to the review of Anonymous Referee #1 and Anonymous Referee #2 as a single supplement pdf document. The reply to the review of Anonymous Referee #2 can be found on pages 1-11, the reply to the review of Anonymous Referee #1 on pages 12-15.

Please also note the supplement to this comment:
https://www.clim-past-discuss.net/cp-2017-92/cp-2017-92-AC1-supplement.pdf
* * *
[Figure]

**Supplement:**

**Reply to reviewer 2:**

We thank reviewer 2 for the detailed examination of the presented work. This allows us to clarify some issues potentially not emphasized enough within our discussion manuscript. Therefore we will use this opportunity to addresses major issues together with detailed answers to the key points mentioned by the reviewers. Reviewer comments are given in italic letters whereas our replies are given in normal letters.

**Point* (1):**

The method assumes that the forward problem (converting surface temperature to d15N) is completely described by the firn model, and that all variations in d15N can be linked 1-to-1 to past surface temperature. It is thus no surprise that they can reconstruct the original temperature very accurately, because they know the exact accumulation rates and physics of the forward model. Unfortunately, that is not at all true in the real world.

We are well aware of the fact that this assumption does not hold true. Due to uncertainties and simplifications in firn densification and gas diffusion physics, uncertainties in common firn models and measurement data our assumption is only an approximation of the real world as mentioned by the reviewer. Therefore, we will discuss several issues in this reply to reviewer 2. Nevertheless we used this assumption here to show the functionality of the automated fitting algorithm. Detailed uncertainty estimations for a "real world" scenario as demanded by the reviewer are behind the scope of this work and will follow for the reconstructions using measurement data ( $\delta^{15}N$ ,  $\delta^{40}Ar$ ,  $\delta^{15}N_{excess}$ ) in next publications. Again, the aim of this work is to present the automated gas isotope fitting algorithm applied to synthetic Holocene  $\delta^{15}N$  data and to study the uncertainties emerging from the algorithm itself. Furthermore the focal question in this study is: what is the minimum final mismatch in  $\delta^{15}N$  for Holocene data we can reach and what does this mean for the final temperature mismatches. Studying and moreover answering these questions makes it mandatory to create well defined  $\delta^{15}N$  targets and related temperature histories, as we did here. It is impossible to answer these questions without using synthetic data in a methodology study. The aim is to evaluate the accuracy and associated uncertainty of the inverse method itself to then later (in a future study) apply this method to a real  $\delta^{15}N$  dataset, for which of course the original driving temperature history in unknown.

The d15N is influenced by variations in convective zone thickness (the CZ is ignored here altogether), firn layering that influences the lock-in process, melt layers and wind crusts, etc. Real data (as opposed to the synthetic data used) further suffer from analytical noise in the laboratory. All these things will reduce the ability to reconstruct temperature from d15N. Furthermore, our understanding of firn densification is incomplete, with several physical models giving different results, microstructure effects not included in models, and hypothesized influences of dust softening. All these effects remain unaccounted for, which further reduces the ability to use d15N. The authors use identical firn physics in the forward and inverse models, which is an idealization that is untenable.

Regarding the convective zone (CZ): The presented fitting algorithm was used together with the two most frequently used firn models for temperature reconstructions based on stable isotopes of air, the Schwander et al. (1997) model which has no CZ build in (or better a constant CZ of 0 m) and with the Goujon firn model (Goujon et al., 2003) (which assumes constant convective zone over time, that can easily be set in the code). This difference between the two firn models only changes significantly the absolute temperature rather than the temperature anomalies as it was shown by other studies (e.g., Guillevic et al. (2013), fig. 3). In the presented work, we show the results using the model from Schwander et al. (1997), because the differences between the obtained solutions using the two models are negligible besides a constant temperate offset of about 2.3K. Also, noteworthy is that there is no firn model at the moment which uses a dynamically changing CZ. Indeed, this should be investigated but requires additional intense work. Additionally, the knowledge of the time evolution of CZ changes for the time periods of millennia to several hundreds of millennia (in frequency and magnitude) is too poor to estimate the influence of this quantity on the reconstruction.

In addition the algorithm is able to fit  $\delta^{15}N$ ,  $\delta^{40}Ar$  and  $\delta^{15}N_{excess}$  data as mentioned in the paper (e.g. in the abstract at line 17). In fig.1 we show unpublished data to clarify that the algorithm is usable for  $\delta^{15}N_{excess}$  besides  $\delta^{15}N$  data. Here the  $\delta^{15}N_{excess}$  data from Kobashi et al. (2008) was used as the fitting target using the same approach. We reach a final mismatch (2 $\sigma$ ) of 3.7 permeg, which is below the analytic measurement uncertainty of 5.0 to 9.8 permeg of the measurement data. We hope that this is convincing enough to show the functionality of our algorithm also for this quantity. The automated inversion of different gas isotope quantities ( $\delta^{15}N$ ,  $\delta^{40}Ar$ ,

 $\delta^{15}N_{excess}$ ) provides a unique opportunity to study the difference of the gained solutions for the different targets and to improve our knowledge about the uncertainties of gas isotope based temperature reconstructions using a single firn model. Because of the "perfect physics scenario" as mentioned above it is not necessary to show the synthetic  $\delta^{40}Ar$  and  $\delta^{15}N_{excess}$  fits here, because the gained solutions are the same. This is will be different when using measurement data. Here differences between the temperature solutions gained from the single targets ( $\delta^{15}N, \delta^{40}Ar, \delta^{15}N_{excess}$ ) will become obvious due to several sources of signal noise. These differences will allow to quantify the uncertainties associated with processes mentioned by the reviewer.

Next, the presented algorithm is not dependent on the firn model, which leads to the implication that the algorithm can be coupled to different firn models describing firn physics in different ways. An automated reconstruction algorithm avoiding manual manipulation and leading to reproducible solutions makes it possible for the first time, to study and learn from the differences between the solutions. Differences that then can be assigned to different firn models and their shortcomings, resulting in more robust uncertainty estimates as was possible before. Thus, the algorithm provides the possibility to test firn models by fitting different targets and as mentioned before to learn from the differences between the solutions obtained by matching single targets. This is exactly the reason the algorithm was developed for.

Several studies have shown that on the cm-scale there is much variation in parameters like d15N and CH4, reflecting a staggered trapping of gas bubbles within the firn-ice transition zone. See e.g. Etheridge et al. (1992), Rhodes et al. (2016), and Mitchell et al. (2015). This may be relevant as the sample size is typically smaller than the average layer thickness.

Also that point is not related to the scope this paper. Within the scope of paleoclimate reconstruction, the pertinent focus is more on how to extract signals from gas isotope data rather than how to represent potential sources of signal noise. Of course signal noise (such as firn heterogeneity) should be included in the uncertainty estimation, which is planned in a future study dealing with modelling (among others) real  $\delta^{15}$ N data. However, we will try to account for this question here. We fully agree, for the reconstruction using measurement data, it is necessary to keep cm scale variability in mind. Our view on this point can be summarized as follows: During the analytical analyses of ice core air data it is common to measured replicates for given depths, from which the measurement uncertainties of the gas isotope data is calculated using pooled-standard-deviation (Hedges L. V., 1985). Often it is not possible to take real replicates (same depth) and instead the replicates are taken from nearby depths. So, the cm scale variability is to some degree already included in the measurement uncertainty, because each measurement point represents the average over a few centimetres of ice. This is especially the case for low accumulation sites or glacial ice samples for which the vertical length of a sample (e.g., 10-25 cm long for the glacial part of the NGRIP ice core, Kindler et al., 2014) covers the equivalent of 20-50 yrs of ice at approx. 35 kyrs b2k. Increasing the depth resolution of the samples would increase our knowledge of cm scale variability, for e.g. identifying anomalous layers that could have been rapidly isolated from the surface due to a high density layer (e.g., Rosen et al. (2014)). As this variability is likely due to heterogeneity in the density profile, this may not help to better reconstruct a meaningful temperature history, rather to observe the source of signal noise. To sum up: The cm scale variability, in many cases, is already incorporated in the analytical noise obtained from gas isotope measurements, due to analytical techniques themselves. Assuming the measurement uncertainty as Gaussian distributed, it is very easy to incorporate this source of uncertainty in the inverse modelling approach. This will increase the uncertainty of the temperature according to Eq. (9) in our manuscript using the presented approach. The same equation can also be used for the calculation of the uncertainty in temperature related to measurement uncertainty in general.

To answer the pertinent question of how to better extract a meaningful temperature history from a noisy ice core record, an excellent – but costly – solution is of course to use multiple ice cores. The GISP2 ice core has actually the chance to have a "sister ice core" drilled only a few kilometres apart (the GRIP ice core) and combining  $\delta^{15}$ N-based temperature reconstructed from both ice cores is likely one of the best ways to overcome potential cm scale variability. A comparison of ice cores that were drilled even closer might be even more advantageous.

Gas diffusion and trapping smooths out the d15N signal, which provides a fundamental limit on the time resolution at which surface temperature is recorded and could potentially be reconstructed.

The duration of gas diffusion from the top of the diffusive column to the bottom where the air is closed off in bubbles is for Holocene conditions in Greenland approximately in the order of 10 yr (Schwander et al. 1997), whereas the data resolution of the synthetic targets was set to 20 yr to mimic the measurement data from Kobashi et al. (2008) with a mean data resolution of about 17 yr (see section 2.4: "*Generating synthetic target data*"). In the study of Kindler et al. (2014) it was shown that a Glacial Greenland lock-in depth leads to a damping of the  $\delta^{15}$ N signal of about 30% for a 10 K temperature rise in 20 yr. We further assume that the smoothing according to the lock-in process is negligible for Greenland Holocene conditions according to the much smaller amplitude signals and shallower lock-in depth for Holocene conditions.

From the above it is clear to me that the precision that the authors state for their method is a meaningless number, that teaches us nothing about how well d15N can reconstruct temperature. A more interesting approach would be to include these fundamental uncertainties in a stochastic way, and see how well the method works under realistic settings. The synthetic data could e.g. be generated with a different firn physics description, and should be subject to CZ fluctuations, LIZ thickness variations and analytical noise.

This is obviously a misunderstanding. Indeed we did not mention that the mismatch in  $\delta^{15}N$  and the therefrom calculated temperature range would correspond to an uncertainty of temperature reconstructions. This range is only the uncertainty part that directly relates to the inversion model approach and does not include any other uncertainties that exist. In the "perfect world" scenario it should be theoretically possible to reach a final mismatch of zero for  $\delta^{15}N$  as well as for temperature. The reason why this was not reached in our study is related to the memory effects in the ice sheet model which leads to a rising computational effort for reaching very low mismatches. An improvement of one section of the time series will be paid by degradation in another part. To circumvent the computational demand we developed the correction step (step 4, section 2.4.4), which accounted for this memory effects. This means that in finite time there has to be a limit the algorithm can reach, which is exactly characterized by the final mismatches presented here.

Additionally, in the perspective of making a complete uncertainty budget for a temperature history reconstructed based on  $\delta^{15}N$  data (again, as will be done in a future publication), this uncertainty value for the inverse modelling method, being not zero, cannot be neglected and should therefore be taken into account.

**Point* (2):**

There are 2 fundamental inputs into the model, namely temperature and accumulation rate. The authors assume the latter is known with zero uncertainty (both in values and age model). This is a very unrealistic assumption. Even if the layer-count were perfect (which it is not), correcting for ice thinning has a fundamental uncertainty. Especially in the early Holocene, this can easily exceed 10%. As the method fits the d15N data, all accumulation errors are mapped into the temperature reconstruction. This is not accounted for...As an aside, the authors convert the accumulation record from Cuffey et al. onto the GICC05 scale, which makes it internally inconsistent because the accumulation rate is the derivative of the age scale, so changing the age scale should change the accumulation values. Since the method is sensitive to the decadal-scale accumulation variability, it may be insufficient to use this crude approach.

Answer for the point on the conversion of the Cuffey accumulation record to the GICC05 time scale: We think this comment arose from a lack of details given in the paper. Therefore we describe in the following in more detail the procedure we used to produce the finally used accumulation rate data for our modelling work. The original accumulation rate for the GISP2 ice core is the one published in Cuffey and Clow (1997), produced using an ice flow model adapted to the GISP2 location. The accumulation rate used to feed the ice flow model was optimised in order to match the time scale from Meese et al. (1994) for the Holocene, based on annual layer counting. Seierstad et al. (2014) transferred the GISP2 chronology to the GICC05 reference timeframe using multiple match points to the NGRIP and GRIP ice cores, both already on GICC05. We used these match points and modified the GISP2 duration in between match points linearly in order for the considered interval to match exactly the GICC05 duration. This way, the detailed GISP2 annual layer counting information is kept, but is only stretched/compressed in time. This was done for all intervals in between two match points. The accumulation

data were then re-calculated accordingly, as obviously (as stated by the reviewer) this is needed in order to keep the same total amount of ice accumulated at the GISP2 site. Actually, to obtain an even better consistency, the best would be to re-run the Cuffey and Clow ice sheet model, using the GICC05 timescale as target timescale, and use the resulting accumulation rate data (but this is beyond the scope if this study).

Furthermore, as we have shown in the paper in section 2.4 "Accumulation rate input", the accumulation rate variability has a minor impact compared to the temperature on the variability of the  $\delta^{15}N$  data in the Holocene (see also fig.A02). The influence of the quantities, accumulation rate or temperature, into the temperature reconstruction is not equal, the accumulation rate variability during the Holocene explains about 12 to 30% of the  $\delta^{15}N$  variability. 30% corresponds to the 8.2 kyr event and 12% for the mean of the whole Holocene period including the 8.2 kyr event. Hence the influence of accumulation changes is generally below 10% during the Holocene. If the accumulation is assumed to be completely correct then the missing part will be assigned to temperature variations. Also in section 2.3.1 we show that the polynomial degree in temperature is more important than for the accumulation for the calculation of a polynomial transfer function (see line 3-6 at page 5 and fig.S02). Nevertheless for the fitting of the Holocene measurement data we will use all three accumulation rate scenarios as shown in fig.S01. The difference in the reconstructed temperature arising from the differences of these three scenarios will be used for the uncertainty calculation as well and is most likely higher than the uncertainty arising from the conversion of the accumulation rate data to the GICC05 timescale.

**Point (3):**

The authors have no way of validating that their Delta-age is correct, which is critical to constrain the timing of climate change. In all d15N modeling studies I'm aware of, the use of d18O as a temperature template ensures that Delta-age is correct. In particular during abrupt events, the timing of gas and ice signals gives you Delta-age. This information is lost in their method, which is completely independent of d18O. If the modeled Dage is off by 50 years (which is easy to do in Greenland, particularly during the glacial), the timing of the temperature solution is also off by 50 years. It would be interesting to run their algorithm on data from the last deglaciation, and see whether it reproduces the timing of abrupt change as seen in d18O. Because Deltaage is underconstrained, the timing of all reconstructed high-frequency temperature variations is uncertain.

We thank the reviewer for mentioning that point, since we have not explicitly discussed the behaviour of the  $\Delta$ age agreement in the paper and we will catch up on this. The  $\Delta$ age adjustment in the Holocene case is related to the smooth temperature solution calculated by the Monte Carlo part of the algorithm. If a smooth temperature solution is found which creates a robust long term signal in  $\delta^{15}$ N the gas age - ice age difference from that model output is used to calculate the high frequency information and to find the right timing for adding the high frequency signal to the smooth temperature solution as it was explained in section 2.4.2 (page 9, lines 11-13) and section 2.4.3 (page 9, lines 18-23). As the measurement target data is set on the ice age scale (like all gas isotope data after measurement) and the accumulation rate is known, the high frequency temperature signal has to have the right timing when the final calculated  $\delta^{15}$ N signal matches the target (or measurement) data. Table 1 contains the final mismatches ( $2\sigma$ ) in  $\Delta$ age for all scenarios and shows very well that with a known accumulation rate and firm physics it is possible to fit the  $\Delta$ age history in the Holocene with mean uncertainties better than 2 yr. This table together with a similar statement will be added to the paper in the results section. Figure 2 shows the time series of the mismatches in  $\Delta$ age for all scenarios and is used to clarify the functionality of the algorithm itself.

More interesting for the reviewer is probably the "real world" scenario. Due to the large uncertainties in measured or modelled  $\Delta$ age it is a challenging task to validate the correctness of the  $\Delta$ age regime anyway. But to give an idea to that issue we show here ongoing work. Figure 3 shows the comparison of the  $\Delta$ age regime modelled using our algorithm together with the GISP2  $\delta^{15}N$  data from Kobashi et al. (2008) and the  $\Delta$ age regime published with the GICC05 GISP2 gas age scale from Seierstad et al. (2014) and Rasmussen et al. (2014). Besides a nearly constant offset of about 20 yr in the early Holocene the agreement is amazing with a standard deviation ( $2\sigma$ ) of the mismatches of 7.8 yr over the whole time series and 3.5 yr for the last 8.2 kyr.

For Glacial conditions the task of reconstructing the temperature (with the right frequency and magnitude) without  $\delta^{18}O_{ice}$  information is much more challenging as mentioned by the reviewer due to the highly variable gas age - ice age differences between stadial and interstadial conditions. Here the  $\Delta$ age can vary several hundreds of years. Also the accumulation rate data is more uncertain than in the Holocene. To prove that the presented fitting algorithm also works for Glacial conditions we inverted the  $\delta^{15}N$  data measured for the NGRIP

ice core by Kindler et al. (2014) for two Dangsgard-Oeschger events, namely DO6 and DO7. Since the magnitudes of those events are higher and the signals are smoother than in the Holocene we only had to use the Monte Carlo type input generator (section 2.4.2) for changing the temperature inputs. To compare our results to the  $\delta^{18}O_{ice}$  based manually calibration method from Kindler et al. (2014) we used the ss09sea06bm time scale (NGRIP members (2004), Johnsen et al. (2001)) as it was done in the Kindler et al. publication. For the model spin-up we use the accumulation rate and temperature data from Kindler et al. (2014) for the time span 36.2 to 60 kyr. The reconstruction window (containing DO6 and DO7) was set to 32 to 36.2 kyr. As the first guess (starting point) of the reconstruction we used the accumulation rate data for NGRIP from the ss09sea06bm time scale together with a constant temperature of about -49 °C for this time window. As minimization criterion D for the reconstruction we simply use the sum of the mean squared errors (wRMSE) of the  $\delta^{15}N$  and  $\Delta$ age mismatches weighted with their uncertainties according to the following equation instead of the mean  $\delta^{15}N$  misfit alone as used for the Holocene.

$$D = \sqrt{wRMSE(\delta^{15}N)} + \sqrt{wRMSE(\Delta age)}$$
(1)
$$= \sqrt{\frac{1}{N}\sum_{i} \left[\frac{\delta^{15}N_{meas,i} - \delta^{15}N_{mod,i}}{\varepsilon_{\delta^{15}N,i}}\right]^{2}} + \sqrt{\frac{1}{M}\sum_{j} \left[\frac{\Delta age_{meas,j} - \Delta age_{mod,j}}{\varepsilon_{\Delta age,j}}\right]^{2}}$$

Here  $\varepsilon_{\delta^{15}N,i}$  and  $\varepsilon_{\Delta age,i}$  are the uncertainties in  $\delta^{15}N$  and  $\Delta age$  for the measured values i or j ( $\Delta age$  match points: Guillevic, M. (2013), p.65, Tab. 3.2) and N, M the number of measurement values. We set  $\varepsilon_{\delta^{15}N,i} = 20$  permeg for all i (Kindler et al. 2014) and  $\varepsilon_{\Delta age,i} = 50$  yr for all j. The values of 50 yr for the  $\Delta age$  uncertainties were chosen according to reach the same mean relative errors for both terms. The relative uncertainties in  $\Delta$ age can easily reach up to 50% and more in the Glacial using the ss09sea06bm time scale which results in a domination of the  $\delta^{15}$ N misfits over the  $\Delta$ age misfits (10-20% when using GICC05 time scale, pers. communication M. Guillevic). Because of that issue we had to set the  $\Delta$ age uncertainties to 50 yr to make both terms equally important for the fitting algorithm. To sum up: The temperature variations were exactly done in the same way as described in section 2.4.2 within the paper without any further adjustments. We only had to add one target more ( $\Delta$ age) to the minimization criterion to account for a second unknown, i.e. the also uncertain accumulation rates. In fig.4 we show preliminary results. The  $\delta^{15}$ N and  $\Delta$ age fitting (a,b) and the resulting gained temperature and accumulation rate solutions (c,d) using the presented algorithm are completely independent from  $\delta^{18}$ O which provides a great opportunity to evaluate the  $\delta^{18}$ O based reconstruction. In this study the algorithm was used in three steps (MCS0, MCS1, MCS FIN). First, starting with the first guess (constant temperature), the temperature was changed as explained before. The accumulation rate was changed parallel to the temperature allowing a random offset shift (up and down) together with a stretching or compressing (in y direction) of the accumulation rate signal over the whole time window (32 to 36.2 kyr). This first step leads to the "Monte Carlo Solution 0" (MCS0) which provides a first approximation and is the base for the next step. For the next step, we fixed the accumulation rate and let the algorithm only changes the temperature to improve the  $\delta^{15}$ N fit (MSC1). Finally, we allow the algorithm to change the temperature together with the accumulation rate using the Monte Carlo type input generator for both quantities. This also allows the change of the shape of the accumulation rate data. This final step can be seen as a fine tuning of the gained solutions from the steps before. The reached mismatches in  $\delta^{15}$ N and  $\Delta$ age of all steps are at least of the same quality or better than the  $\delta^{18}$ O based manual method from Kindler et al. (2014) (see Tab.2). The gained temperature solutions show a very good agreement in timing and magnitude compared to the reconstruction of Kindler et al. (2014). Also the accumulation rate solutions show that the accumulation has to be reduced significantly compared to the ss09sea06bm data to allow a high quality fit of the  $\delta^{15}$ N and  $\Delta$ age target data, a result highly similar to Kindler et al. (2014) and Guillevic et al. (2013). Regarding the mismatches in  $\delta^{15}$ N and  $\Delta$ age of the final MCS FIN solution show a 15% smaller misfit in  $\delta^{15}N(2\sigma)$  and an about 31% smaller misfit for  $\Delta age(2\sigma)$ . Keeping in mind that the used approach is completely independent from  $\delta^{18}$ O should clarify the functionality and quality of the presented gas isotope fitting approach also for Glacial reconstructions.

**Point (4):**

I am surprised the authors don't event attempt to invert the existing GISP2 data (which are even plotted). This seems like a missed opportunity; especially given that it would allow comparison to existing reconstructions to estimate the accuracy of the method.

We understand the surprise of the reviewer of the missing application on existing data but the focus on this paper is indeed the inversion model, its mathematics as well as a proper analysis on the capabilities of the algorithm itself based on a synthetic data set. Yet, we will provide a limited projection on future publications hereafter. However, we underline once more that the accuracy of the inverse modelling algorithm can only be estimated using a synthetic dataset, as shown in our paper. The GISP2 data for  $\delta^{15}N$ ,  $\delta^{40}Ar$  and  $\delta^{15}N_{excess}$  are already inverted using the presented algorithm and will be presented in a following publication, since there are a couple of items to be addressed in detail which would overload the scope of the present methodological manuscript. But we want to discuss the algorithm itself to examine what are the possibilities and the limits of the presented fitting method in a well-known modelling frame work. The main focus of the present manuscript is to present the algorithm in a single publication rather than in the supplementary to bring the attention on the functionality and fundamental ideas of the algorithm rather than on the gained solutions. We think that is important to give the interested reader the chance to understand the basic concepts behind the algorithm and to show the functionality on a well-known example (here synthetic  $\delta^{15}$ N). We hope that we can simplify gas isotope based reconstructions for a broad spectrum of researchers using our or maybe a related approach later on. As we have shown, the approach works for all relevant gas isotope quantities ( $\delta^{15}N$ ,  $\delta^{40}Ar$ ,  $\delta^{15}N_{excess}$ ) and for Holocene and Glacial data as well. The approach is a completely new method which enables the automatized fitting of gas isotope data without manual tuning of parameters minimizes the "subjective" impact of a single researcher. All together we are sure that this is the best way to present our new elegant fitting method in the framework we have chosen.

**Point* (5):**

The paper is overly long. I recommend section 2.3 be removed entirely, and other sections be shortened considerably. There are also 32 (!) figures in the manuscript, which is too many. Dividing the figures into main, appendix and supplement figures is annoying, as it requires a lot of going back and forth.

We agree that the paper is long and that the amount of figures is possibly too much. This said, we tried to explain and discuss the algorithm in every detail to clarify the functionality of all parts of the fitting method. Our aim was also to present this new method in a totally transparent manner. To shorten the paper we will remove section 2.3 as suggested by the reviewer. This section was thought as a motivation for the presented fitting algorithm. We agree that it is not necessary for the paper itself. Also, we will reduce the numbers of figures by removing the following figures:

Main part: fig01, fig02, fig03 Supplementary: fig.S02 to fig.S16

Next we will shift all the appendix figures in a new supplementary. This means we will have now 4 figures in the manuscript and 11 figures in the supplementary. To keep the paper understandable a further reduction of figures and pages is not possible.

**Detailed comments:**

**Page 1 Line 26: Give references for Holocene temperature reconstructions (there are many!).**

Since we developed a novel algorithm for ice core based temperature reconstructions and explained the functionality based on synthetic data of Holocene like behaviour, we gave references to other ice core based reconstruction methods. (borehole inversion, page 1 line 6ff; calibration of water isotopes from the ice core water samples, page 1 line 9ff;  $\delta^{15}N$ ,  $\delta^{40}Ar$ ,  $\delta^{15}N_{excess}$  based methods, page 1 line 14ff). Because no reconstruction for measurement data is shown here, we think it is not necessary to refer for other (or non-ice-core-based) reconstructions.

Page 3 Eq. (1): what about the convective zone? You should correct for that Eq. (2): The surface temperature should really be the temperature at the bottom of the convective zone where diffusion starts to dominate. This may smooth out some of the abrupt decadal-scale temperature variations.

The Schwander model does not use a convective zone at this stage but such a CZ could be implemented in the calculation of  $\delta^{15}N_{grav}$ , by subtracting the gravitational signal formed over the length of the CZ. Has the reviewer examples of a convective zone deep enough to smooth out decadal scale signals except "Megadune" sites (Severinghaus et al., 2010)? We are a bit surprised with this sentence, as for example J. Severinghaus, using measurements from South Pole, shows that already the seasonal signal in gas diffusion affect the first 10 to 12 m of firm (Severinghaus, 1998) pointing to a shallow or even non-existing CZ. Furthermore, we have to remember that we are discussing the rather stable Holocene period in Greenland for which no low accumulation and strong katabatic wind situations are to be expected minimizing the effect of deep CZ. For a CZ to have an effect as strong as to smooth out decadal scale variation, its deepness would need to be of several dozens of meters, which is highly unrealistic even for Glacial Summit conditions. On the contrary the process definitely affecting the damping of the signal is gas diffusion occurring in the firn, producing i) an increase in the mean gas age of the gas at the LID and ii) a damping of the signal whose amplitude is positively correlated with the LID (see for example, Buizert et al. (2012), Fig. 7; Buizert et al. (2013), Fig. 2; and Kindler et al. (2014), Fig. 2).

**Line 17: Martinerie et al. (1994) gives the depth of the bubble close-off, whereas d15N is set at the lock-in depth instead. The LID is shallower than the COD. Is this difference accounted for, and how?**

This is explained in details in the description of the Schwander firn model (Schwander, 1997). We did not report details in this paper because we thought this model is a) already quite well known and b) well described in its original paper. However we report this information here: Indeed it is well known that the LID is shallower than the COD, due to the presence of a non-diffusive zone. Originally the COD is defined by a density threshold, calculated as a function of temperature (Martinerie et al., 1994). In the Schwander model, to account for the presence of the non-diffusive zone, this COD definition is modified by subtracting 14 kg/m3 to the COD density definition, in order to match the observed depth where gas diffusion stops. This offset was optimised using firn data from Summit (GRIP) collected in the 90', Greenland, and we therefore believe this definition in highly appropriate for the GISP2 site over the Holocene.

Page 4 Section 2.2: what are the model parameters? What are the time and spatial step? How deep does the domain extend? What geothermal heat flux is used, etc.

The model parameters are described in detail in Schwander et al. (1997).

Section 2.3: I recommend this is removed completely. I don't see the point, especially the dynamic case which we know doesn't behave linearly due to memory effects.

To shorten the paper we will remove this section as suggested by the reviewer. This section was thought as a motivation for the presented fitting algorithm. We agree that it is not necessary for the paper itself.

Page 6 Line 13: Not too robust. It'd be easy to have a 10% uncertainty in the thinning function.

We reformulate these sentences from:

"Except for these technical adjustments, the accumulation rate input data remains unmodified, assuming high reliability of this data during the Holocene. This is due to the fact that the data was gained by annual layer counting, and the use of a thinning model which should be rather robust for the first 1500 m of the 3000 m ice core (Cuffey and Clow, 1997)."

The new text now reads:

"Except for these technical adjustments, the accumulation rate input data remains unmodified, assuming high reliability of this data during the Holocene. The data was gained by annual layer counting, and the use of a thinning model which should lead to maximum relative uncertainty of 10% for the first 1500 m of the 3000 m ice core (Cuffey and Clow, 1997)."

Fig.1: Fitting of GISP2  $\delta^{15}N_{excess}$  data (measurement data from Kobashi et al. 2008): a) measured versus modelled  $\delta^{15}N_{excess}$  time series; b) zoom-in for a randomly chosen 1000 yr interval; c) time series of final mismatches  $\Delta\delta^{15}N_{excess}$  for the measured minus the modelled  $\delta^{15}N_{excess}$  data; d) histogram for the same quantity as in c) with values for the final mismatch (2 $\sigma$ ) and offset;

---

## Author Comment (AC2) · 8 Sep 2017

Reply to reviewer 2:

We thank reviewer 2 for the detailed examination of the presented work. This allows us to clarify some issues potentially not emphasized enough within our discussion manuscript. Therefore we will use this opportunity to addresses major issues together with detailed answers to the key points mentioned by the reviewers. Reviewer comments are given in italic letters whereas our replies are given in normal letters.

*Point (1):*
*The method assumes that the forward problem (converting surface temperature to d15N) is completely described by the firn model, and that all variations in d15N can be linked 1-to-1 to past surface temperature. It is thus no surprise that they can reconstruct the original temperature very accurately, because they know the exact accumulation rates and physics of the forward model. Unfortunately, that is not at all true in the real world.*

We are well aware of the fact that this assumption does not hold true. Due to uncertainties and simplifications in firn densification and gas diffusion physics, uncertainties in common firn models and measurement data our assumption is only an approximation of the real world as mentioned by the reviewer. Therefore, we will discuss several issues in this reply to reviewer 2. Nevertheless we used this assumption here to show the functionality of the automated fitting algorithm. Detailed uncertainty estimations for a "real world" scenario as demanded by the reviewer are behind the scope of this work and will follow for the reconstructions using measurement data ($\delta^{15}N$, $\delta^{40}Ar$, $\delta^{15}N_{excess}$) in next publications. Again, the aim of this work is to present the automated gas isotope fitting algorithm applied to synthetic Holocene $\delta^{15}N$ data and to study the uncertainties emerging from the algorithm itself. Furthermore the focal question in this study is: what is the minimum final mismatch in $\delta^{15}N$ for Holocene data we can reach and what does this mean for the final temperature mismatches. Studying and moreover answering these questions makes it mandatory to create well defined $\delta^{15}N$ targets and related temperature histories, as we did here. It is impossible to answer these questions without using synthetic data in a methodology study. The aim is to evaluate the accuracy and associated uncertainty of the inverse method itself to then later (in a future study) apply this method to a real $\delta^{15}N$ dataset, for which of course the original driving temperature history in unknown.

*The d15N is influenced by variations in convective zone thickness (the CZ is ignored here altogether), firn layering that influences the lock-in process, melt layers and wind crusts, etc. Real data (as opposed to the synthetic data used) further suffer from analytical noise in the laboratory. All these things will reduce the ability to reconstruct temperature from d15N. Furthermore, our understanding of firn densification is incomplete, with several physical models giving different results, microstructure effects not included in models, and hypothesized influences of dust softening. All these effects remain unaccounted for, which further reduces the ability to use d15N. The authors use identical firn physics in the forward and inverse models, which is an idealization that is untenable.*

Regarding the convective zone (CZ): The presented fitting algorithm was used together with the two most frequently used firn models for temperature reconstructions based on stable isotopes of air, the Schwander et al. (1997) model which has no CZ build in (or better a constant CZ of 0 m) and with the Goujon firn model (Goujon et al., 2003) (which assumes constant convective zone over time, that can easily be set in the code). This difference between the two firn models only changes significantly the absolute temperature rather than the temperature anomalies as it was shown by other studies (e.g., Guillevic et al. (2013), fig. 3). In the presented work, we show the results using the model from Schwander et al. (1997), because the differences between the obtained solutions using the two models are negligible besides a constant temperate offset of about 2.3K. Also, noteworthy is that there is no firn model at the moment which uses a dynamically changing CZ. Indeed, this should be investigated but requires additional intense work. Additionally, the knowledge of the time evolution of CZ changes for the time periods of millennia to several hundreds of millennia (in frequency and magnitude) is too poor to estimate the influence of this quantity on the reconstruction.

In addition the algorithm is able to fit $\delta^{15}N$, $\delta^{40}Ar$ and $\delta^{15}N_{excess}$ data as mentioned in the paper (e.g. in the abstract at line 17). In fig.1 we show unpublished data to clarify that the algorithm is usable for $\delta^{15}N_{excess}$ besides $\delta^{15}N$ data. Here the $\delta^{15}N_{excess}$ data from Kobashi et al. (2008) was used as the fitting target using the same approach. We reach a final mismatch ($2\sigma$) of 3.7 permeg, which is below the analytic measurement uncertainty of 5.0 to 9.8 permeg of the measurement data. We hope that this is convincing enough to show the functionality of our algorithm also for this quantity. The automated inversion of different gas isotope quantities ($\delta^{15}N$, $\delta^{40}Ar$,

$\delta^{15}N_{excess}$) provides a unique opportunity to study the difference of the gained solutions for the different targets and to improve our knowledge about the uncertainties of gas isotope based temperature reconstructions using a single firn model. Because of the "perfect physics scenario" as mentioned above it is not necessary to show the synthetic $\delta^{40}Ar$ and $\delta^{15}N_{excess}$ fits here, because the gained solutions are the same. This is will be different when using measurement data. Here differences between the temperature solutions gained from the single targets ($\delta^{15}N$, $\delta^{40}Ar$, $\delta^{15}N_{excess}$) will become obvious due to several sources of signal noise. These differences will allow to quantify the uncertainties associated with processes mentioned by the reviewer.

Next, the presented algorithm is not dependent on the firn model, which leads to the implication that the algorithm can be coupled to different firn models describing firn physics in different ways. An automated reconstruction algorithm avoiding manual manipulation and leading to reproducible solutions makes it possible for the first time, to study and learn from the differences between the solutions. Differences that then can be assigned to different firn models and their shortcomings, resulting in more robust uncertainty estimates as was possible before. Thus, the algorithm provides the possibility to test firn models by fitting different targets and as mentioned before to learn from the differences between the solutions obtained by matching single targets. This is exactly the reason the algorithm was developed for.

*Several studies have shown that on the cm-scale there is much variation in parameters like d15N and CH4, reflecting a staggered trapping of gas bubbles within the firn-ice transition zone. See e.g. Etheridge et al. (1992), Rhodes et al. (2016), and Mitchell et al. (2015). This may be relevant as the sample size is typically smaller than the average layer thickness.*

Also that point is not related to the scope this paper. Within the scope of paleoclimate reconstruction, the pertinent focus is more on how to extract signals from gas isotope data rather than how to represent potential sources of signal noise. Of course signal noise (such as firn heterogeneity) should be included in the uncertainty estimation, which is planned in a future study dealing with modelling (among others) real $\delta^{15}N$ data. However, we will try to account for this question here. We fully agree, for the reconstruction using measurement data, it is necessary to keep cm scale variability in mind. Our view on this point can be summarized as follows: During the analytical analyses of ice core air data it is common to measured replicates for given depths, from which the measurement uncertainties of the gas isotope data is calculated using pooled-standard-deviation (Hedges L. V., 1985). Often it is not possible to take real replicates (same depth) and instead the replicates are taken from nearby depths. So, the cm scale variability is to some degree already included in the measurement uncertainty, because each measurement point represents the average over a few centimetres of ice. This is especially the case for low accumulation sites or glacial ice samples for which the vertical length of a sample (e.g., 10-25 cm long for the glacial part of the NGRIP ice core, Kindler et al., 2014) covers the equivalent of 20-50 yrs of ice at approx. 35 kyrs b2k. Increasing the depth resolution of the samples would increase our knowledge of cm scale variability, for e.g. identifying anomalous layers that could have been rapidly isolated from the surface due to a high density layer (e.g., Rosen et al. (2014)). As this variability is likely due to heterogeneity in the density profile, this may not help to better reconstruct a meaningful temperature history, rather to observe the source of signal noise. To sum up: The cm scale variability, in many cases, is already incorporated in the analytical noise obtained from gas isotope measurements, due to analytical techniques themselves. Assuming the measurement uncertainty as Gaussian distributed, it is very easy to incorporate this source of uncertainty in the inverse modelling approach. This will increase the uncertainty of the temperature according to Eq. (9) in our manuscript using the presented approach. The same equation can also be used for the calculation of the uncertainty in temperature related to measurement uncertainty in general.

To answer the pertinent question of how to better extract a meaningful temperature history from a noisy ice core record, an excellent – but costly – solution is of course to use multiple ice cores. The GISP2 ice core has actually the chance to have a "sister ice core" drilled only a few kilometres apart (the GRIP ice core) and combining $\delta^{15}N$-based temperature reconstructed from both ice cores is likely one of the best ways to overcome potential cm scale variability. A comparison of ice cores that were drilled even closer might be even more advantageous.

*Gas diffusion and trapping smooths out the d15N signal, which provides a fundamental limit on the time resolution at which surface temperature is recorded and could potentially be reconstructed.*

The duration of gas diffusion from the top of the diffusive column to the bottom where the air is closed off in bubbles is for Holocene conditions in Greenland approximately in the order of 10 yr (Schwander et al. 1997), whereas the data resolution of the synthetic targets was set to 20 yr to mimic the measurement data from Kobashi et al. (2008) with a mean data resolution of about 17 yr (see section 2.4: *"Generating synthetic target data"*). In the study of Kindler et al. (2014) it was shown that a Glacial Greenland lock-in depth leads to a damping of the $\delta^{15}N$ signal of about 30% for a 10 K temperature rise in 20 yr. We further assume that the smoothing according to the lock-in process is negligible for Greenland Holocene conditions according to the much smaller amplitude signals and shallower lock-in depth for Holocene conditions.

*From the above it is clear to me that the precision that the authors state for their method is a meaningless number, that teaches us nothing about how well d15N can reconstruct temperature. A more interesting approach would be to include these fundamental uncertainties in a stochastic way, and see how well the method works under realistic settings. The synthetic data could e.g. be generated with a different firn physics description, and should be subject to CZ fluctuations, LIZ thickness variations and analytical noise.*

This is obviously a misunderstanding. Indeed we did not mention that the mismatch in $\delta^{15}N$ and the therefrom calculated temperature range would correspond to an uncertainty of temperature reconstructions. This range is only the uncertainty part that directly relates to the inversion model approach and does not include any other uncertainties that exist. In the "perfect world" scenario it should be theoretically possible to reach a final mismatch of zero for $\delta^{15}N$ as well as for temperature. The reason why this was not reached in our study is related to the memory effects in the ice sheet model which leads to a rising computational effort for reaching very low mismatches. An improvement of one section of the time series will be paid by degradation in another part. To circumvent the computational demand we developed the correction step (step 4, section 2.4.4), which accounted for this memory effects. This means that in finite time there has to be a limit the algorithm can reach, which is exactly characterized by the final mismatches presented here.

Additionally, in the perspective of making a complete uncertainty budget for a temperature history reconstructed based on $\delta^{15}N$ data (again, as will be done in a future publication), this uncertainty value for the inverse modelling method, being not zero, cannot be neglected and should therefore be taken into account.

*Point (2):*

*There are 2 fundamental inputs into the model, namely temperature and accumulation rate. The authors assume the latter is known with zero uncertainty (both in values and age model). This is a very unrealistic assumption. Even if the layer-count were perfect (which it is not), correcting for ice thinning has a fundamental uncertainty. Especially in the early Holocene, this can easily exceed 10%. As the method fits the d15N data, all accumulation errors are mapped into the temperature reconstruction. This is not accounted for...As an aside, the authors convert the accumulation record from Cuffey et al. onto the GICC05 scale, which makes it internally inconsistent because the accumulation rate is the derivative of the age scale, so changing the age scale should change the accumulation values. Since the method is sensitive to the decadal-scale accumulation variability, it may be insufficient to use this crude approach.*

Answer for the point on the conversion of the Cuffey accumulation record to the GICC05 time scale: We think this comment arose from a lack of details given in the paper. Therefore we describe in the following in more detail the procedure we used to produce the finally used accumulation rate data for our modelling work. The original accumulation rate for the GISP2 ice core is the one published in Cuffey and Clow (1997), produced using an ice flow model adapted to the GISP2 location. The accumulation rate used to feed the ice flow model was optimised in order to match the time scale from Meese et al. (1994) for the Holocene, based on annual layer counting. Seierstad et al. (2014) transferred the GISP2 chronology to the GICC05 reference timeframe using multiple match points to the NGRIP and GRIP ice cores, both already on GICC05. We used these match points and modified the GISP2 duration in between match points linearly in order for the considered interval to match exactly the GICC05 duration. This way, the detailed GISP2 annual layer counting information is kept, but is only stretched/compressed in time. This was done for all intervals in between two match points. The accumulation

data were then re-calculated accordingly, as obviously (as stated by the reviewer) this is needed in order to keep the same total amount of ice accumulated at the GISP2 site. Actually, to obtain an even better consistency, the best would be to re-run the Cuffey and Clow ice sheet model, using the GICC05 timescale as target timescale, and use the resulting accumulation rate data (but this is beyond the scope if this study).

Furthermore, as we have shown in the paper in section 2.4 "Accumulation rate input", the accumulation rate variability has a minor impact compared to the temperature on the variability of the $\delta^{15}N$ data in the Holocene (see also fig.A02). The influence of the quantities, accumulation rate or temperature, into the temperature reconstruction is not equal, the accumulation rate variability during the Holocene explains about 12 to 30% of the $\delta^{15}N$ variability. 30% corresponds to the 8.2 kyr event and 12% for the mean of the whole Holocene period including the 8.2 kyr event. Hence the influence of accumulation changes is generally below 10% during the Holocene. If the accumulation is assumed to be completely correct then the missing part will be assigned to temperature variations. Also in section 2.3.1 we show that the polynomial degree in temperature is more important than for the accumulation for the calculation of a polynomial transfer function (see line 3-6 at page 5 and fig.S02). Nevertheless for the fitting of the Holocene measurement data we will use all three accumulation rate scenarios as shown in fig.S01. The difference in the reconstructed temperature arising from the differences of these three scenarios will be used for the uncertainty calculation as well and is most likely higher than the uncertainty arising from the conversion of the accumulation rate data to the GICC05 timescale.

*Point (3):*
*The authors have no way of validating that their Delta-age is correct, which is critical to constrain the timing of climate change. In all d15N modeling studies I'm aware of, the use of d18O as a temperature template ensures that Delta-age is correct. In particular during abrupt events, the timing of gas and ice signals gives you Delta-age. This information is lost in their method, which is completely independent of d18O. If the modeled Dage is off by 50 years (which is easy to do in Greenland, particularly during the glacial), the timing of the temperature solution is also off by 50 years. It would be interesting to run their algorithm on data from the last deglaciation, and see whether it reproduces the timing of abrupt change as seen in d18O. Because Deltaage is underconstrained, the timing of all reconstructed high-frequency temperature variations is uncertain.*

We thank the reviewer for mentioning that point, since we have not explicitly discussed the behaviour of the $\Delta$age agreement in the paper and we will catch up on this. The $\Delta$age adjustment in the Holocene case is related to the smooth temperature solution calculated by the Monte Carlo part of the algorithm. If a smooth temperature solution is found which creates a robust long term signal in $\delta^{15}N$ the gas age - ice age difference from that model output is used to calculate the high frequency information and to find the right timing for adding the high frequency signal to the smooth temperature solution as it was explained in section 2.4.2 (page 9, lines 11-13) and section 2.4.3 (page 9, lines 18-23). As the measurement target data is set on the ice age scale (like all gas isotope data after measurement) and the accumulation rate is known, the high frequency temperature signal has to have the right timing when the final calculated $\delta^{15}N$ signal matches the target (or measurement) data. Table 1 contains the final mismatches ($2\sigma$) in $\Delta$age for all scenarios and shows very well that with a known accumulation rate and firn physics it is possible to fit the $\Delta$age history in the Holocene with mean uncertainties better than 2 yr. This table together with a similar statement will be added to the paper in the results section. Figure 2 shows the time series of the mismatches in $\Delta$age for all scenarios and is used to clarify the functionality of the algorithm itself.

More interesting for the reviewer is probably the "real world" scenario. Due to the large uncertainties in measured or modelled $\Delta$age it is a challenging task to validate the correctness of the $\Delta$age regime anyway. But to give an idea to that issue we show here ongoing work. Figure 3 shows the comparison of the $\Delta$age regime modelled using our algorithm together with the GISP2 $\delta^{15}N$ data from Kobashi et al. (2008) and the $\Delta$age regime published with the GICC05 GISP2 gas age scale from Seierstad et al. (2014) and Rasmussen et al. (2014). Besides a nearly constant offset of about 20 yr in the early Holocene the agreement is amazing with a standard deviation ($2\sigma$) of the mismatches of 7.8 yr over the whole time series and 3.5 yr for the last 8.2 kyr.

For Glacial conditions the task of reconstructing the temperature (with the right frequency and magnitude) without $\delta^{18}O_{ice}$ information is much more challenging as mentioned by the reviewer due to the highly variable gas age - ice age differences between stadial and interstadial conditions. Here the $\Delta$age can vary several hundreds of years. Also the accumulation rate data is more uncertain than in the Holocene. To prove that the presented fitting algorithm also works for Glacial conditions we inverted the $\delta^{15}N$ data measured for the NGRIP

ice core by Kindler et al. (2014) for two Dangsgard-Oeschger events, namely DO6 and DO7. Since the magnitudes of those events are higher and the signals are smoother than in the Holocene we only had to use the Monte Carlo type input generator (section 2.4.2) for changing the temperature inputs. To compare our results to the $\delta^{18}O_{ice}$ based manually calibration method from Kindler et al. (2014) we used the ss09sea06bm time scale (NGRIP members (2004), Johnsen et al. (2001)) as it was done in the Kindler et al. publication. For the model spin-up we use the accumulation rate and temperature data from Kindler et al. (2014) for the time span 36.2 to 60 kyr. The reconstruction window (containing DO6 and DO7) was set to 32 to 36.2 kyr. As the first guess (starting point) of the reconstruction we used the accumulation rate data for NGRIP from the ss09sea06bm time scale together with a constant temperature of about -49 °C for this time window. As minimization criterion D for the reconstruction we simply use the sum of the mean squared errors (wRMSE) of the $\delta^{15}N$ and $\Delta age$ mismatches weighted with their uncertainties according to the following equation instead of the mean $\delta^{15}N$ misfit alone as used for the Holocene.

$$D = \sqrt{wRMSE(\delta^{15}N)} + \sqrt{wRMSE(\Delta age)} \qquad (1)$$

$$= \sqrt{\frac{1}{N}\sum_i \left[\frac{\delta^{15}N_{meas,i} - \delta^{15}N_{mod,i}}{\varepsilon_{\delta^{15}N,i}}\right]^2} + \sqrt{\frac{1}{M}\sum_j \left[\frac{\Delta age_{meas,j} - \Delta age_{mod,j}}{\varepsilon_{\Delta age,j}}\right]^2}$$

Here $\varepsilon_{\delta^{15}N,i}$ and $\varepsilon_{\Delta age,j}$ are the uncertainties in $\delta^{15}N$ and $\Delta age$ for the measured values i or j ($\Delta age$ match points: Guillevic, M. (2013), p.65, Tab. 3.2) and N, M the number of measurement values. We set $\varepsilon_{\delta^{15}N,i} = 20$ permeg for all i (Kindler et al. 2014) and $\varepsilon_{\Delta age,j} = 50$ yr for all j. The values of 50 yr for the $\Delta age$ uncertainties were chosen according to reach the same mean relative errors for both terms. The relative uncertainties in $\Delta age$ can easily reach up to 50% and more in the Glacial using the ss09sea06bm time scale which results in a domination of the $\delta^{15}N$ misfits over the $\Delta age$ misfits (10-20% when using GICC05 time scale, pers. communication M. Guillevic). Because of that issue we had to set the $\Delta age$ uncertainties to 50 yr to make both terms equally important for the fitting algorithm. To sum up: The temperature variations were exactly done in the same way as described in section 2.4.2 within the paper without any further adjustments. We only had to add one target more ($\Delta age$) to the minimization criterion to account for a second unknown, i.e. the also uncertain accumulation rates. In fig.4 we show preliminary results. The $\delta^{15}N$ and $\Delta age$ fitting (a,b) and the resulting gained temperature and accumulation rate solutions (c,d) using the presented algorithm are completely independent from $\delta^{18}O$ which provides a great opportunity to evaluate the $\delta^{18}O$ based reconstruction. In this study the algorithm was used in three steps (MCS0, MCS1, MCS FIN). First, starting with the first guess (constant temperature), the temperature was changed as explained before. The accumulation rate was changed parallel to the temperature allowing a random offset shift (up and down) together with a stretching or compressing (in y direction) of the accumulation rate signal over the whole time window (32 to 36.2 kyr). This first step leads to the "Monte Carlo Solution 0" (MCS0) which provides a first approximation and is the base for the next step. For the next step, we fixed the accumulation rate and let the algorithm only changes the temperature to improve the $\delta^{15}N$ fit (MSC1). Finally, we allow the algorithm to change the temperature together with the accumulation rate using the Monte Carlo type input generator for both quantities. This also allows the change of the shape of the accumulation rate data. This final step can be seen as a fine tuning of the gained solutions from the steps before. The reached mismatches in $\delta^{15}N$ and $\Delta age$ of all steps are at least of the same quality or better than the $\delta^{18}O$ based manual method from Kindler et al. (2014) (see Tab.2). The gained temperature solutions show a very good agreement in timing and magnitude compared to the reconstruction of Kindler et al. (2014). Also the accumulation rate solutions show that the accumulation has to be reduced significantly compared to the ss09sea06bm data to allow a high quality fit of the $\delta^{15}N$ and $\Delta age$ target data, a result highly similar to Kindler et al. (2014) and Guillevic et al. (2013). Regarding the mismatches in $\delta^{15}N$ and $\Delta age$ of the final MCS FIN solution show a 15% smaller misfit in $\delta^{15}N$ ($2\sigma$) and an about 31% smaller misfit for $\Delta age$ ($2\sigma$). Keeping in mind that the used approach is completely independent from $\delta^{18}O$ should clarify the functionality and quality of the presented gas isotope fitting approach also for Glacial reconstructions.

*Point (4):*
*I am surprised the authors don't event attempt to invert the existing GISP2 data (which are even plotted). This seems like a missed opportunity; especially given that it would allow comparison to existing reconstructions to estimate the accuracy of the method.*

We understand the surprise of the reviewer of the missing application on existing data but the focus on this paper is indeed the inversion model, its mathematics as well as a proper analysis on the capabilities of the algorithm itself based on a synthetic data set. Yet, we will provide a limited projection on future publications hereafter. However, we underline once more that the accuracy of the inverse modelling algorithm can only be estimated using a synthetic dataset, as shown in our paper. The GISP2 data for $\delta^{15}N$, $\delta^{40}Ar$ and $\delta^{15}N_{excess}$ are already inverted using the presented algorithm and will be presented in a following publication, since there are a couple of items to be addressed in detail which would overload the scope of the present methodological manuscript. But we want to discuss the algorithm itself to examine what are the possibilities and the limits of the presented fitting method in a well-known modelling frame work. The main focus of the present manuscript is to present the algorithm in a single publication rather than in the supplementary to bring the attention on the functionality and fundamental ideas of the algorithm rather than on the gained solutions. We think that is important to give the interested reader the chance to understand the basic concepts behind the algorithm and to show the functionality on a well-known example (here synthetic $\delta^{15}N$). We hope that we can simplify gas isotope based reconstructions for a broad spectrum of researchers using our or maybe a related approach later on. As we have shown, the approach works for all relevant gas isotope quantities ($\delta^{15}N$, $\delta^{40}Ar$, $\delta^{15}N_{excess}$) and for Holocene and Glacial data as well. The approach is a completely new method which enables the automatized fitting of gas isotope data without manual tuning of parameters minimizes the "subjective" impact of a single researcher. All together we are sure that this is the best way to present our new elegant fitting method in the framework we have chosen.

*Point (5):*

*The paper is overly long. I recommend section 2.3 be removed entirely, and other sections be shortened considerably. There are also 32 (!) figures in the manuscript, which is too many. Dividing the figures into main, appendix and supplement figures is annoying, as it requires a lot of going back and forth.*

We agree that the paper is long and that the amount of figures is possibly too much. This said, we tried to explain and discuss the algorithm in every detail to clarify the functionality of all parts of the fitting method. Our aim was also to present this new method in a totally transparent manner. To shorten the paper we will remove section 2.3 as suggested by the reviewer. This section was thought as a motivation for the presented fitting algorithm. We agree that it is not necessary for the paper itself. Also, we will reduce the numbers of figures by removing the following figures:

Main part: fig01, fig02, fig03
Supplementary: fig.S02 to fig.S16

Next we will shift all the appendix figures in a new supplementary. This means we will have now 4 figures in the manuscript and 11 figures in the supplementary. To keep the paper understandable a further reduction of figures and pages is not possible.

*Detailed comments:*

*Page 1 Line 26: Give references for Holocene temperature reconstructions (there are many!).*

Since we developed a novel algorithm for ice core based temperature reconstructions and explained the functionality based on synthetic data of Holocene like behaviour, we gave references to other ice core based reconstruction methods. (borehole inversion, page 1 line 6ff; calibration of water isotopes from the ice core water samples, page 1 line 9ff; $\delta^{15}N$, $\delta^{40}Ar$, $\delta^{15}N_{excess}$ based methods, page 1 line 14ff). Because no reconstruction for measurement data is shown here, we think it is not necessary to refer for other (or non-ice-core-based) reconstructions.

*Page 3 Eq. (1): what about the convective zone? You should correct for that Eq. (2): The surface temperature should really be the temperature at the bottom of the convective zone where diffusion starts to dominate. This may smooth out some of the abrupt decadal-scale temperature variations.*

The Schwander model does not use a convective zone at this stage but such a CZ could be implemented in the calculation of $\delta^{15}N_{grav}$, by subtracting the gravitational signal formed over the length of the CZ. Has the reviewer examples of a convective zone deep enough to smooth out decadal scale signals except "Megadune" sites (Severinghaus et al., 2010)? We are a bit surprised with this sentence, as for example J. Severinghaus, using measurements from South Pole, shows that already the seasonal signal in gas diffusion affect the first 10 to 12 m of firn (Severinghaus, 1998) pointing to a shallow or even non-existing CZ. Furthermore, we have to remember that we are discussing the rather stable Holocene period in Greenland for which no low accumulation and strong katabatic wind situations are to be expected minimizing the effect of deep CZ. For a CZ to have an effect as strong as to smooth out decadal scale variation, its deepness would need to be of several dozens of meters, which is highly unrealistic even for Glacial Summit conditions. On the contrary the process definitely affecting the damping of the signal is gas diffusion occurring in the firn, producing i) an increase in the mean gas age of the gas at the LID and ii) a damping of the signal whose amplitude is positively correlated with the LID (see for example, Buizert et al. (2012), Fig. 7; Buizert et al. (2013), Fig. 2; and Kindler et al. (2014), Fig. 2).

*Line 17: Martinerie et al. (1994) gives the depth of the bubble close-off, whereas d15N is set at the lock-in depth instead. The LID is shallower than the COD. Is this difference accounted for, and how?*

This is explained in details in the description of the Schwander firn model (Schwander, 1997). We did not report details in this paper because we thought this model is a) already quite well known and b) well described in its original paper. However we report this information here: Indeed it is well known that the LID is shallower than the COD, due to the presence of a non-diffusive zone. Originally the COD is defined by a density threshold, calculated as a function of temperature (Martinerie et al., 1994). In the Schwander model, to account for the presence of the non-diffusive zone, this COD definition is modified by subtracting 14 kg/m3 to the COD density definition, in order to match the observed depth where gas diffusion stops. This offset was optimised using firn data from Summit (GRIP) collected in the 90', Greenland, and we therefore believe this definition in highly appropriate for the GISP2 site over the Holocene.

*Page 4 Section 2.2: what are the model parameters? What are the time and spatial step? How deep does the domain extend? What geothermal heat flux is used, etc.*

The model parameters are described in detail in Schwander et al. (1997).

*Section 2.3: I recommend this is removed completely. I don't see the point, especially the dynamic case which we know doesn't behave linearly due to memory effects.*

To shorten the paper we will remove this section as suggested by the reviewer. This section was thought as a motivation for the presented fitting algorithm. We agree that it is not necessary for the paper itself.

*Page 6 Line 13: Not too robust. It'd be easy to have a 10% uncertainty in the thinning function.*

We reformulate these sentences from:

"Except for these technical adjustments, the accumulation rate input data remains unmodified, assuming high reliability of this data during the Holocene. This is due to the fact that the data was gained by annual layer counting, and the use of a thinning model which should be rather robust for the first 1500 m of the 3000 m ice core (Cuffey and Clow, 1997)."

The new text now reads:

"Except for these technical adjustments, the accumulation rate input data remains unmodified, assuming high reliability of this data during the Holocene. The data was gained by annual layer counting, and the use of a thinning model which should lead to maximum relative uncertainty of 10% for the first 1500 m of the 3000 m ice core (Cuffey and Clow, 1997)."

[Figure]

Fig.1: Fitting of GISP2 $\delta^{15}N_{excess}$ data (measurement data from Kobashi et al. 2008): a) measured versus modelled $\delta^{15}N_{excess}$ time series; b) zoom-in for a randomly chosen 1000 yr interval; c) time series of final mismatches $\Delta\delta^{15}N_{excess}$ for the measured minus the modelled $\delta^{15}N_{excess}$ data; d) histogram for the same quantity as in c) with values for the final mismatch (2σ) and offset;

[Figure]

Fig.2: Comparison of the mismatches in Δage between the synthetic target and modelled data for all scenarios showing excellent agreement in Δage. All fits leads to a mean mismatch Δ(Δage) in Δage better than 2yr (2σ).

| Scenario: | 2σ Δ(Δage) [yr] | Scenario: | 2σ Δ(Δage) [yr] |
|---|---|---|---|
| S1 | 1.14 | S5 | 1.24 |
| S2 | 1.60 | H1 | 1.23 |
| S3 | 1.98 | H2 | 1.18 |
| S4 | 1.41 | H3 | 1.30 |

Tab.1: Final mismatches (2σ) of Δage for all scenarios.

| Solution | D | Mismatch δ15N (2σ) [permeg] | Mean mismatch δ15N* [permeg] | Mismatch Δage (2σ) [yr] | Mean mismatch Δage* [yr] |
|---|---|---|---|---|---|
| Kindler 2014 | 3.6 | 44.5 | 17.9 | 256 | 101 |
| | | | | | |
| first guess | 7.8 | 128.7 | 63.8 | 328 | 138 |
| MCS0 | 3.1 | 50.0 | 19.3 | 199 | 82 |
| MCS1 | 2.9 | 44.3 | 17.6 | 200 | 84 |
| MCS FIN | 2.6 | 37.8 | 15.6 | 175 | 63 |

Tab.2: Prove of concept for Glacial reconstruction; *The mean mismatches for δ15N and Δage were calculated according to Eq. (7) in the paper.

[Figure]

Fig.3: Top plot: Comparison of the modelled Δage (red, unpublished/this study) using the presented approach together with the Schwander model and δ15N target data (Kobashi et. all 2008) with the Δage time series for GICC05 GISP2 gasagescale (black curve) from Seierstad et al. (2014), Rasmussen et al. (2014) and related 2σ uncertainty (dotted line). Bottom plot: Time series of the mismatches Δ(Δage) in Δage.

[Figure]

Fig.4: Prove of concept for Glacial reconstructions (NGRIP DO6 and DO7): a) $\delta^{15}N$ target plot: $\delta^{15}N$ model output for the first guess input (blue line), Kindler et al. (2014) fit (orange dotted line), Monte Carlo solution 0 (yellow line, unpublished data), Monte Carlo solution 1 (purple line, unpublished data), final Monte Carlo solution (green line, unpublished data), $\delta^{15}N$ measurement target (black dotted line, measurement points are black cycles, data from Kindler et al. (2014)); b) $\Delta$age target plot: $\Delta$age model output for the first guess input (blue line), Kindler et al. (2014) fit (orange dotted line), Monte Carlo solution 0 (yellow line, unpublished data), Monte Carlo solution 1 (purple line, unpublished data), final Monte Carlo solution (green line, unpublished data), $\Delta$age measurement target (black dotted line, measurement points are black cycles, data from Guillevic (2013)); c) temperature solution plot: first guess input (blue line), Kindler et al. (2014) solution (orange dotted line), Monte Carlo solution 0 (yellow line, unpublished data), Monte Carlo solution 1 (purple line, unpublished data), final Monte Carlo solution (green line, unpublished data); d) accumulation rate solution plot: first guess input (blue line), Kindler et al. (2014) solution (orange dotted line), Monte Carlo solution 0 (yellow line, unpublished data), Monte Carlo solution 1 (purple line, unpublished data), final Monte Carlo solution (green line, unpublished data);

Reply to reviewer 1:

General Comment on reviewer 1:

It was very difficult to find in the comment from reviewer 1 a scientific and/or technical discussion on the scientific questions the presented work is dealing with. This makes it really challenging to find an appropriate way to give an answer on this review. However, we will try to address the key issues and will give an adequate answer in the best possible manner.

*This work describes a technical and mathematical variation on previously published work by Kobashi et al. (2010; 2011; 2012).*

That is not true at all, which makes us wonder if the reviewer read in detail our submitted article. The presented approach is completely different to the work of Kobashi et al. mentioned here from the reviewer. The calculation of temperature gradients from $\delta^{15}N_{excess}$ data used for a temporal integration using the Goujon model as it was done by Kobashi differs from our approach significantly. Our approach calculates in a first step a long term signal in temperature and the isotope target, which is superimposed by a high frequency signal in a next step. Finally we created a correction method for dealing with remaining misfits (permeg level) due to memory effects. Besides the methodology view, we will list 6 major differences between both methods:

(1) Our approach allows the automated high quality fitting (or inversion) of $\delta^{15}N$ or $\delta^{40}Ar$ or $\delta^{15}N_{excess}$ data as single targets (as it was mentioned in the paper and shown in the answer to reviewer 2) and provides consequently the opportunity to compare the solution of one target against the others. The method from Kobashi et al. uses all isotope quantities together, eliminating the possibility to compare the reconstruction obtained from one quantity using the other ones.

(2) Our approach is applicable to Holocene as well as Glacial data (as it was mentioned in the paper and shown in the answer to reviewer 2), whereas the approach of Kobashi et al. was only designed and tested for Holocene reconstructions.

(3) Our approach allows a parallel adjustment of the accumulation rate input data (if it is necessary) (shown in the answer to reviewer 2 for Glacial data).

(4) Our approach uses a well-defined minimization criterion which provides the possibility to adapt it to a variety of target combinations (e.g. $\delta^{15}N$ and $\Delta$age for Glacial reconstructions).

(5) Our approach is not dependent on the choice of the first guess for the reconstruction. A worse choice of this quantity will only elongate the computational time of the "Monte Carlo type input generator" step. This was shown since we used the same first guess (a constant temperature) for all different synthetic data scenarios.

(6) Our approach splits the reconstructed temperature and isotope signals in a long term and a high frequency signal (for Holocene), which provides additional information and bases for further research questions and uncertainty calculations.

*Kobashi et al. innovated by creating a novel hybrid firn densification-thermal diffusion model, much the same as is done here.*

To our knowledge, Kobashi et al. used for their reconstruction the already published model from Goujon et al. (2003). The Goujon model is indeed a firn densification model, coupled to an ice sheet flow model also calculating heat transfer from surface to bedrock. The Goujon model does not have a module to automatically optimise the temperature and accumulation scenario needed for the inversion. We agree to this comment in the sense that Kobashi et al. indeed innovated a novel technique to create an input temperature scenario to feed the Goujon model, using firn temperature gradients extracted from $\delta^{15}N_{excess}$. We also agree that our approach, similar to the method from Kobashi, needs a firn densification and heat diffusion model to provide the physical basis for the inversion of $\delta^{15}N$ data. Nevertheless both methods differ significantly from each other as discussed before. Also both methods were created independently.

*The scientific advance represented by this work is useful but is very incremental, almost to the point of not standing alone as a publishable scientific paper. It is not clear to me that this work suffices as a "Least Publishable Unit", and reads more like an Appendix to a publication.*

We think that this is a very subjective single opinion which is not underpinned by any scientific argument. We created a completely automated algorithm which is able to provide high quality fits for all relevant gas isotope quantities and works as well for Holocene as Glacial conditions. Furthermore the algorithm is not firn model dependent as it was coupled on two state of the art firn models, leading to comparable results. We also refer to the answer on reviewer 2 for point 1, explaining the achievements of this method in detail. Moreover, there are many examples of models presented and published in CP, presenting in details each step of the model, without publishing data related paleoclimate reconstruction alongside. A very well-known example is the recently published automatization method presented by Winstrup et al. (2012) in order to run automated annual layer counting in ice cores using multiple annually resolved records. This paper was very welcomed by the reviewers in CP, and the method has been successfully applied since then to reconstruct chronologies for many ice core records. We believe the focus of our submitted manuscript is very similar to the one from Winstrup et al., applied to a different problem, but always linked to paleoclimate reconstruction.

*It would improve the paper if Kobashi's work could be compared to the results found here, and placed in a larger context. Additionally, it would be helpful if the present work were actually used to reconstruct Greenland temperature over the Holocene, much as Kobashi et al. did. I am somewhat surprised that Kobashi is not a co-author, considering how heavily this work relies on Kobashi et al.'s prior work. The synthetic data looks a lot like Kobashi et al.'s actual data. Why not show the actual data?*

As explained in detail in the answer on point (4) for reviewer 2, due to the aim of the paper to describe the algorithm in every detail and in a well-known environment (i.e., using a synthetic dataset as target), we decided for not showing the results of the inversion of the GISP2 measurement data within this publication. Also the paper was criticized because of its length and the amount of figures which makes it all together impossible to show everything in a single publication without the danger of losing the scope on the major issues we want to address.

As explained above, our inversion method is entirely new and can therefore not be considered to rely at all on Kobashi's work, which by the way we very much appreciate. However we recognise that obviously our method (as Kobashi's method) works only when coupled to a firn densification model itself coupled to an ice sheet flow model equipped with heat transfer, such as the Schwander or the Goujon model.

*Minor comments:*

*page 1 line 11 "The presented approach is completely automated..."*

We correct for that. The new sentence reads:
"The presented approach is completely automated and leads to a match of the $\delta^{15}N$ target data in the low permeg level and to related temperature deviations of a few tenths of Kelvin for different data scenarios, showing the robustness of the reconstruction method."

*page 1 line 23 "since it represents a time of moderate natural variations prior to anthropogenic disturbance, often referred to as a baseline...."*

We correct for that. The new sentence reads:
"Holocene climate variability is of key interest to our society, since it represents a time of moderate natural variations prior to anthropogenic disturbance, often referred to as a baseline for today's increasing greenhouse effect driven by mankind."

*page 2 line 6 "The studies of Dahl-Jensen et al. (1998) and Cuffey et al. (1995; 1997) demonstrate the usefulness of inverting the measured borehole temperature profile for surface temperature history....."*

We correct for that and added the references. The new sentence reads:
"The studies of Dahl-Jensen et al. (1998) and Cuffey et al. (1995; 1997) demonstrate the usefulness of inverting the measured borehole temperature profile for surface temperature history estimates for the investigated drilling site using a coupled heat- and ice-flow model. "

*page 2 line 9 " unable to resolve..."*

We correct for that. The new sentence reads:
"Because of smoothing effects due to the nature of heat diffusion within an ice sheet, this method is unable to resolve fast temperature oscillations and leads to a rapid reduction of the time resolution towards the past."

*page 3 line 24. It is not clear from the wording here which thermal diffusion sensitivity value was used here. Is it the Grachev and Severinghaus (2003), or the Leuenberge et al. (1999)? This must be clarified. A separate issue is that the Leuenberger et al. value is based on measurements that were made in pure nitrogen, not in air. It is well known, and indeed predicted from theory, that the thermal diffusion sensitivity (and thermal diffusion factor) is larger in pure gases than in air. For example, Grachev and Severinghaus measured these parameters in both pure N2 and in air, and found a substantial difference between the two (Figure 1). As can be seen in Figure 1, the thermal diffusion factor in pure N2 is 0.0037 whereas in air it is less than 0.0036. Even more troubling is the fact that the 1960s-era measurements made in pure N2 by the sources that Leuenberger et al. use disagree well outside the analytical error (0.0035) with the pure-N2 value of Grachev and Severinghaus, which was made with a modern mass spectrometer. This suggests that the 1960s era measurements by Boersma-Klein and De Vries (1966) were badly in error. Given the primitive technology of that time, this is not a criticism of these workers, but it is clear that their values should not be used for the present study.*

We thank the reviewer a lot for mentioning that point. Indeed we used a wrong equation here, which was a relic from an older version of the paper. We did all calculations using the thermal diffusion sensitivity from Grachev and Severinghaus (2003). We changed Eq. (4) to:

$$\alpha_T = \left(8.656 - \frac{1323\,K}{\bar{\bar{T}}}\right) \cdot 10^{-3}$$

*page 3 line 26 "The firn model used here behaves purely as a forward model,......"*

We correct for that. The new sentence reads:
"The firn model used here behaves purely as a forward model, which means that for the given input time series the output parameters (here finally $\delta^{15}N_{mod}(t)$) can be calculated, but it is not easily possible to construct from measured isotope data the related surface temperature or accumulation rate histories."

*page 4 line 3 You must say which ice core was used here. Is it GISP2?*

We correct for that. The new sentence reads:
"In this study, accumulation rate data from Cuffey and Clow (1997) for the GISP2 ice core, adapted to the GICC05 chronology, is used (Rasmussen et al., 2008; Seierstad et al., 2014)."

---

## Referee Report (RR1)

*Climate of the Past*
2017-92
Novel automated inversion algorithm for temperature reconstruction using gas isotopes from ice cores
Michael Döring and Markus Leuenberger

I was asked to review a revised version of this manuscript; I have now read a version called *cp-2017-92-manuscript-version-3.pd*f, on which my remarks are based.

It was apparently unclear to the reviewers of the original submission that the scope of the manuscript was limited to describing a new algorithm to address an inverse problem. In the revised manuscript, the authors have attempted to clarify this limitation, and to address further concerns that would need to be considered in order to apply it to real data.

Although the manuscript appears to have been significantly improved since the original submission, I still found the manuscript to be dense and difficult to read. The point of the manuscript is to introduce a method to infer past temperatures, but the description of the method is difficult to follow. I think the material is appropriate for publication in *Climate of the Past*, but the presentation of the material can still be improved to make it more accessible to a broader range of climate scientists. Reaching readers is, after all, a primary goal of publishing.

Here are issues that I had with Section 2:

- The authors want to describe a complicated procedure. The 4-point summary overview at the top of page 6 is a good start. However, in my view, too much detail is then provided before readers have been given a clear understanding of the whole process. Section 2.3 *Reconstruction Approach* actually contains many intermediate results, numerical values, and detailed discussion points, before the process is fully explained. When I see text like this, I really don't want to read it. I think the text would be easier to read if the authors were to restrict 2.3 to describing the fundamental concepts and assumptions of the approach. Then, e.g. in a subsequent *Discussion* section, they could explain how some of those results were developed or obtained, why they were needed, the numerical values that arose, and how they were used.

- Illustrations integrated with the description of the method could help readers to "see" and understand the procedure. For example, a flow chart could be very useful to help readers as they follow Section 2.3. However, there are no figures in the main text to help here; instead, the text depends heavily on figures and tables in the Supplement. If material is needed to explain the main points of a paper, then that material should be in the main body of the paper. A Supplement should be limited to material that is nice to have in order to *supplement* the primary material. It should not contain primary material that is essential in order for readers to understand the main text.

Reviewers of manuscripts are generally expected to identify two types of problems.

- The first is scientific failings, such as failure to correctly interpret and cite relevant background work, or omissions of key steps in descriptions or analysis, or errors in logical development of ideas or conclusions. I was pleased to see that many of the minor points raised by the two initial reviewers have been addressed.
- The second type of problem that reviewers are asked to identify is communication failings. Is a manuscript organized in a way that readers who are not closely connected with the work can easily understand the approach and grasp the aspects that are novel? If points were unclear to the reviewers, they will probably also be unclear to many other readers. So, I am a bit concerned that more effort appears to have gone into explaining how the initial two reviewers misunderstood the manuscript (15 pages), than has gone into making the manuscript clearer on those points to other readers. Section 4.3 is new in response, but it is only 3 pages long, i.e. only 20% of the length of the argument to the reviewers. I am concerned that the authors may not fully appreciate the challenges that outsiders can have in attempting to read their work.
- In most inverse problems, non-uniqueness of the inferred model parameters grows rapidly with increasing uncertainties in either the data or the model physics. Section 4.3 addresses the new uncertainties (suggested by Reviewer #2) that could be associated individually with each of the imperfectly known real-world data sets, and with unknowns in the model physics. However, it might be prudent to remind potential users of the procedure why a formal inverse approach is necessary.

Page 1, Line 16 –
Results may be reproducible, but are they correct?
Quite apart from uncertainties in the input data or in the model physics, are there biases built into the automated procedure? This was a question that Reviewer #2 posed, and it is unclear to me that it has been answered satisfactorily.

Page 9, Equation (8) –
Why is an L1 norm used instead of an L2 norm?

Page 9, Line 8 –
"If the mismatch decreases compared to the prior input, the new input is saved and used as new guess."
I thought that in most Monte Carlo applications, there was also a probability that a result would *not* be accepted, even if it had a lower mismatch. Why is that not done here?

In their response to reviewers (page 13), the authors argue that their paper is appropriate for publication in *CP*, because another methods paper (Winstrup et al., 2012) was previously welcomed and published in *CP*. While the decision to publish

or not resides with the editors, I personally think that it is obvious that Winstrup et al. is a methods paper, and the Winstrup et al. paper is much clearer and more accessible to readers than this manuscript in its current form.  The editors must also consider these factors.

**Details**
    When acronyms are well established in the broad scientific literature, it is fine to use them.  However, very little space is saved when authors introduce new acronyms for phrases that are relatively short and which are used relatively infrequently.   Writing the phrase in full for clarity whenever it is used makes reading more efficient and produces happier readers, who don't need to search back through dense text to fine the meaning.  For example, is using "cop" for "cut-off period" really necessary?

    I think the manuscript would benefit from a table of variables (and acronyms).  If reader are going to need to frequently look up meanings of variables and acronyms, it would at least make it easier for them if there was one clearly identified place to go for that information.

**Hyphens**
    The manuscript is nearly devoid of hyphens, although in many instances, correctly used hyphens would eliminate minor textual stumbling blocks that can slow readers' grasp of the material.  For example –
- Page 3, line 1 –  "… argon-isotope-based temperature reconstructions."
Or better,
"… temperature reconstructions based on argon isotopes."
- Page 3, line 24 – "… Holocene-like data …."
- Page 6, line 19 – "low-pass filtered:
- Page 8, line 30 – "first-guess temperature"
- Page 6, line 21 - "… accumulation-rate data …"
- Page 16, line 14 – "gas-isotope-based temperature reconstructions …",
or better, "temperature reconstructions based on gas isotopes …",

**Data are plural**
- Page 6, line 21 - "… data are linearly interpolated …"
- Page 6, line 24 - "… data were linearly indeed reconstructed …"
- Page 8, line 2 – "… data are filtered …"
- Page 17, line 5 – "… gas-isotope data are calculated …"
- Page 18, line 11 – "… data are used to run …"

Page 1, line 26 –
"This is shown by high quality fitting of NGRIP $\delta^{15}$N data for two Dansgaard-Oeschger events using the presented approach, leading to results comparable to other studies."

"High quality" is wordy and can appear to be an attempt to prejudice how readers will view the results. It can often be more convincing to simply state that results were comparable, and let readers decide for themselves whether or not the fit was "high quality". If the authors want to retain "high-quality fitting", then it needs a hyphen.

Page 17, line 23 – The distance between GRIP and GISP3 is ~30 km, it is not "a few km". The authors are correct that more cores closer together could be very useful.

**Clarity**
- Page 1, line 15 – "… parameter tuning leading to …" I think there should be a comma after "tuning. Without a comma it is unclear whether the phrase "leading to reproducible temperature estimates" refers to the new automated approach or to the old manual approach. I think the former, but some readers may stumble on this point.
- Page 1, line 16 –
  "… other ice core based temperature reconstruction methods …".
  Please try to avoid long strings of adjectives especially if you don't use hyphens to help readers understand the groupings.
  Better, with hyphens, "… other ice-core-based temperature-reconstruction methods …"
  Or better yet, unpacked,
  "… other temperature-reconstruction methods based on ice cores …".

- Page 2, line 3 - What is intended by "partly even centennial … variations"? This is not a English expression that I understand.
- Page 2, line 14 – There is no "Cuffey et al. (1997)" reference. The correct reference is "Cuffey and Clow (1997)".
- Page 4 – "g is the acceleration constant." I think it is standard practice to mention "gravity" in this context.
- Page 8, line 8 – "…which serve later on as …" Delete "on".
- Page 8, line 31 – "… of about -29.6$^o$C". Did you use -29.6$^o$C, or did you use something different? Why not just say "… of -29.6$^o$C"?

**Tables**
In Table 01, I don't see the point of saying that calculations ran over a weekend. Surely the days of the week are unimportant (are Saturdays really better than Wednesdays?). If the point is the execution time, then state "48 hours" or "N cpu cycles", or whatever is the appropriate number.

**References**

- Guillevic (2013). PhD thesis.  More information would be helpful – title, university, accessibility.
- Schwander et al. (1997) should have a hanging indent, not be fully indented.
- Severinghaus et al. (1998) should have hanging indent.  "Sowers" and "Alley" are mis-spelled.
- Spahni (2003)  Check how "CH 4" is presented in *GRL*.
- Steig et al. (1994).  Only the first word in the title ("Seasonal") should be capitalized.
- Werner et al. (2001)   Missing space in "… present and …"

---

## Author Response (AR2)

**Reply to reviewer #1:**

We thank reviewer #1 for the report. Hereafter we address the questions and comments mentioned by the reviewer. Reviewer comments are given in *italic letters* whereas our replies are given in normal letters.

*It needs to be made clear, right at the beginning, that d15N by itself is not sufficient to provide a temperature record. There must be a highly accurate independent accumulation rate record in addition. It is only the COMBINATION of d15N data and accumulation data that gives temperature, in the method that is described in the manuscript. The reason is very simple. A doubling of accumulation rate can cause a change in firn thickness*

10 *and hence d15N, with no temperature change whatsoever.*

*The same is true for d40Ar data, when it is used as the sole gas measurement.*

15 We fully agree with reviewer #1 that highly-accurate accumulation-rate data are needed to reconstruct surface-temperature histories from $\delta^{15}N$ and $\delta^{40}Ar$ data. We added this fact already in the first sentence of the abstract by saying:
"Greenland past temperature history can be reconstructed by forcing the output of a firn-densification and heat-diffusion model to fit multiple gas-isotope data ($\delta^{15}N$ or $\delta^{40}Ar$ or $\delta^{15}Nexcess$) extracted from ancient air in

20 Greenland ice-cores using published accumulation-rate (ACC) data sets."

Additionally, the abstract was changed for the dependencies of the reconstructed temperature by adding the accumulation explicitly, i.e. $T(\delta^{15}N, Acc)$. It reads now:

25 "We solve the inverse problem $T(\delta^{15}N, Acc)$ by using a combination of a Monte-Carlo-based iterative approach and the analysis of remaining mismatches between modelled and target data, based on cubic-spline-filtering of random numbers as well as the laboratory-determined temperature-sensitivity for nitrogen isotopes."

In the original text we already mentioned the necessity of the accumulation several times in chapter 2.

*Highly accurate d15Nexcess data, however, can provide a temperature record without any associated accumulation record. This distinction needs to be made clearer. In its present form the reader could be seriously misled by the manuscript. It borders on scientific dishonesty to persist in making this misleading presentation of the facts.*

We agree with the reviewer in part. Indeed the $\delta^{15}N_{excess}$ removes the gravitational signal and therefore temperature gradients, $\Delta T$, along the firn column can be directly estimated. However, we do not agree with the statement that no accumulation rate is necessary to reconstruct surface or bottom temperature because accumulation-rate changes initiate heat-diffusion changes. Therefore, firn models are needed as they describe the

40 changes of the firn column due to temperature and accumulation-rate variability and more important the associated heat diffusion through the firn to extract the related surface-temperature time-series (Landais, 2012). That means, also for $\delta^{15}N_{excess}$-based temperature reconstructions, highly accurate accumulation-rate data are needed. For instance, in the recent published study from Kobashi et al. (2017) where a $\delta^{15}N_{excess}$-based approach was chosen to reconstruct past temperature, the Goujon-firn-model (Goujon et al., 2003) was used for the

45 integration of $\Delta T(t)$ and even if it is not explicitly mentioned in the paper, accumulation-rate data were needed to run the firn model.
In contrast to this, Grachev and Severinghaus (2005) relates the $\Delta T(t)$ variability one to one to past surface-temperature changes with additionally applying a heat-model-based transfer correction of 2 K. In Landais (2012) it is argued that the use of a firn-densification and heat-diffusion model instead of the heat transfer correction

50 would probably lead to more accurate results. We support this statement.

*page 4 line 16 units for mass difference should be "kg per mol"*

We corrected for that and changed the new sentence to:

55 "g is the gravitational acceleration, $\Delta m$ the molar mass-difference between the heavy and light isotopes (equals $10^{-3}$ kg per mol for nitrogen) and R the ideal gas-constant."

*page 4 line 27 the language here is confusing. As I understand it, the Leuenberger, Boersma-Klein, and DeVries results were not the ones used in the current manuscript, so there is no place for them in the sentence describing the given equation. Please cut the mention of Leuenberger and Boersma-Klein and DeVries, as it makes it hard for the reader to know which measurement result is being presented in the given equation. There should be no ambiguity here, and there is indeed no excuse for creating ambiguity in a scientific paper. This issue was already raised by one of the reviewers in the first round of review, and the authors have apparently chosen to ignore that reviewer. This is unacceptable and unprofessional behavior and the authors are hereby warned that continuing to ignore the reviewers will result in rejection of the manuscript.*

We deleted the corresponding reference.

References:

Goujon, C., Barnola, J.-M. and Ritz, C.: Modeling the densification of polar firn including heat diffusion: Application to close-off characteristics and gas isotopic fractionation for Antarctica and Greenland sites, J. Geophys. Res. Atmos., 108(D24), n/a-n/a, doi:10.1029/2002JD003319, 2003.

Grachev, A. M. and Severinghaus, J. P.: A revised + 10±4°C magnitude of the abrupt change in Greenland temperature at the Younger Dryas termination using published GISP2 gas isotope data and air thermal diffusion constants, Quat. Sci. Rev., 24(5–6), 513–519, doi:10.1016/j.quascirev.2004.10.016, 2005.

Kobashi, T., Menviel, L., Jeltsch-Thömmes, A., Vinther, B. M., Box, J. E., Muscheler, R., Nakaegawa, T., Pfister, P. L., Döring, M., Leuenberger, M., Wanner, H. and Ohmura, A.: Volcanic influence on centennial to millennial Holocene Greenland temperature change, Sci. Rep., 7(1), 1441, doi:10.1038/s41598-017-01451-7, 2017.

Landais, A.: Stable Isotopes of N and Ar as Tracers to Retrieve Past Air Temperature from Air Trapped in Ice Cores, in Handbook of Environmental Isotope Geochemistry, edited by M. Baskaran, pp. 865–886, Springer Berlin Heidelberg, Berlin, Heidelberg., 2012.

**Reply to reviewer #3:**

We thank reviewer #3 for the detailed and helpful report. Hereafter we reply to the questions and comments mentioned by the reviewer. Reviewer comments are given in *italic letters* whereas our replies are given in normal letters.

*Although the manuscript appears to have been significantly improved since the original submission, I still found the manuscript to be dense and difficult to read. The point of the manuscript is to introduce a method to infer past temperatures, but the description of the method is difficult to follow. I think the material is appropriate for publication in Climate of the Past, but the presentation of the material can still be improved to make it more accessible to a broader range of climate scientists. Reaching readers is, after all, a primary goal of publishing. Here are issues that I had with Section 2:*
*• The authors want to describe a complicated procedure. The 4-point summary overview at the top of page 6 is a good start. However, in my view, too much detail is then provided before readers have been given a clear understanding of the whole process. Section 2.3 Reconstruction Approach actually contains many intermediate results, numerical values, and detailed discussion points, before the process is fully explained. When I see text like this, I really don't want to read it. I think the text would be easier to read if the authors were to restrict 2.3 to describing the fundamental concepts and assumptions of the approach. Then, e.g. in a subsequent Discussion section, they could explain how some of those results were developed or obtained, why they were needed, the numerical values that arose, and how they were used.*
*• Illustrations integrated with the description of the method could help readers to "see" and understand the procedure. For example, a flow chart could be very useful to help readers as they follow Section 2.3. However, there are no figures in the main text to help here; instead, the text depends heavily on figures and tables in the Supplement. If material is needed to explain the main points of a paper, then that material should be in the main body of the paper. A Supplement should be limited to material that is nice to have in order to supplement the primary material. It should not contain primary material that is essential in order for readers to understand the main text.*

In order to make the manuscript more readable, we follow the suggestion mentioned by the reviewer. We rearranged the order of sections to start with the fundamentals of the new method which is now section 2.1. To provide a guideline for an easier understanding of the description of our inversion approach, we created a flow chart (Fig. 01) as suggested by the reviewer. The new Fig. 01 contains besides the schematic description of the whole algorithm also the used variables. Additionally, we added a new table (Table 01) listing and describing all variables and acronyms used in the manuscript, which can be used together with Fig. 01 to follow the explanations in sections 2.1ff. To further clarify our descriptions we rearranged the figure order in the main and supplementary documents. Additionally, we moved the explanation of the adaption of the accumulation rates to the GICC05 time scale to a new supplement section S1, as we think that is not important for the understanding of the inversion algorithm itself. For the same reason, we moved the sensitivity experiments on the accumulation-rate data (low-pass-filtering and variability) to a new supplement section S2. To shorten the manuscript, we moved a part of our discussion (section 4.2 "High frequency step and final correction") to the supplement section S3 and only summarise the results of the cross-correlation experiments in the main text. Furthermore, we wrote a short remark right at the start of section 2.1 to inform the readers that the problem we deal with is indeed an inverse problem. We hope that with these changes the manuscript is better understandable.

*Reviewers of manuscripts are generally expected to identify two types of problems.*
*• The first is scientific failings, such as failure to correctly interpret and cite relevant background work, or omissions of key steps in descriptions or analysis, or errors in logical development of ideas or conclusions. I was pleased to see that many of the minor points raised by the two initial reviewers have been addressed.*

Thank you.

*• The second type of problem that reviewers are asked to identify is communication failings. Is a manuscript organized in a way that readers who are not closely connected with the work can easily understand the approach and grasp the aspects that are novel? If points were unclear to the reviewers, they will probably also be unclear to many other readers. So, I am a bit concerned that more effort appears to have gone into explaining how the initial two reviewers misunderstood the manuscript (15 pages), than has gone into making the manuscript clearer on those points to other readers. Section 4.3 is new in response, but it is only 3 pages long, i.e. only 20% of the length of the argument to the reviewers. I am concerned that the authors may not fully appreciate the challenges that outsiders can have in attempting to read their work.*

Yes, we got your point. Indeed, when re-reading our response it is obvious that we were very exhausting in describing what the reviewers potentially have misunderstood and not concentrating on improving why this is the case. We now rearranged sections; we added a new figure and table to follow our methodology as explained above.

*• In most inverse problems, non-uniqueness of the inferred model parameters grows rapidly with increasing uncertainties in either the data or the model physics. Section 4.3 addresses the new uncertainties (suggested by Reviewer #2) that could be associated individually with each of the imperfectly known real-world data sets, and with unknowns in the model physics. However, it might be prudent to remind potential users of the procedure why a formal inverse approach is necessary.*

Indeed this is a valid and important point. The problem that we deal with is an inverse problem, since the effect, observed as $\delta^{15}N$ variations, is dependent on its drivers, i.e. temperature and accumulation changes. Therefore, the temperature will be dependent on $\delta^{15}N$ and the accumulation-rate changes. The firn model is a non-linear transfer function of temperature and accumulation rate to firn states and relates then to $\delta^{15}N$ values. The fact that manual adjustments are time-consuming and yet not easy reproducible favours an automated inverse approach as implemented here. This last statement is already mentioned in the introduction and the main text.

But, we added the following short introduction to section 2.1:

"The problem that we deal with is an inverse problem, since the effect, observed as $\delta^{15}N$ variations, is dependent on its drivers, i.e. temperature and accumulation rate changes. Hence, the temperature that we would like to reconstruct depends on $\delta^{15}N$ and accumulation rate changes. To solve this inverse problem, the firn model, which is a non-linear transfer function of temperature and accumulation rate to firn states and relates to $\delta^{15}N$ values, is run iteratively to match the modelled and measured $\delta^{15}N$ values (or other gas species). The automated procedure is significantly more efficient and less time-consuming than a manual approach. The Holocene temperature-reconstruction is implemented by the following four steps (see Fig. 01):"

*Page 1, Line 16 – Results may be reproducible, but are they correct?*
*Quite apart from uncertainties in the input data or in the model physics, are there biases built into the automated procedure? This was a question that Reviewer #2 posed, and it is unclear to me that it has been answered satisfactorily.*

The first statement is indeed an interesting and valid comment. As it is shown in the paper, we are able to reconstruct the synthetic temperature targets by fitting the related $\delta^{15}N$ histories very accurately. This means that the methodology is working. Yet, another issue are the uncertainties in the input data or in the model physics. This, however, has not been the main goal of our investigation, but could be dealt with using our methodology with several different firn models. To our knowledge the presently available firn models (Goujon and Schwander model) are very well comparable. One first step to prove our methodology on real data has been done on data from two Daansgard-Oeschger events which demonstrated good agreement with previously determined reconstructed temperature and accumulation variations by Kindler et al. (2014).

*Page 9, Equation (8) –*
*Why is an L1 norm used instead of an L2 norm?*

During the development of the presented algorithm we investigated different minimization-criterions/cost-functions. As a result of these pre-examinations, no significant differences have been found between the use of a L1 or L2 norm for the fitting of Holocene gas-isotope data. The algorithm is implemented in such a way that the choice of the minimization-criterion can be done by the user in a very simple way by changing only one line of code in a sub-function. Also worthwhile mentioning is that for fitting of glacial $\delta^{15}N$ data we used a L2 norm weighted with the data uncertainties. To summarize: There is no special reason for the use of one norm instead of the other one. Both norms can be used in a sufficient way.

*Page 9, Line 8 –*
*"If the mismatch decreases compared to the prior input, the new input is saved and used as new guess."*
*I thought that in most Monte Carlo applications, there was also a probability that a result would not be*
5  *accepted, even if it had a lower mismatch. Why is that not done here?*

For the current version of the algorithm a new seed of the random generator is set each time before new random values are being generated. That means if the Monte-Carlo algorithm is used several times on the same target, the "way" to the final result is different for any case. This is done to prevent the algorithm for falling into the
10  same local minimum during the minimization and leads to a certain band of solutions (not discussed in the paper). Further, the fact that we do select only the solutions that improve our criteria of minimization corresponds to a probability function.

*In their response to reviewers (page 13), the authors argue that their paper is appropriate for publication in CP,*
15  *because another methods paper (Winstrup et al., 2012) was previously welcomed and published in CP. While the decision to publish or not resides with the editors, I personally think that it is obvious that Winstrup et al. is a methods paper, and the Winstrup et al. paper is much clearer and more accessible to readers than this manuscript in its current form. The editors must also consider these factors.*

20  We fully agree this was an unnecessary statement and of course the decision about an adequate manuscript is taken by the editor. Yet, we are convinced that our manuscript fits well into the Climate of the Past journal since it describes a method for investigating paleo proxy-records regarding temperature variations in the past.

*When acronyms are well established in the broad scientific literature, it is fine to use them. However, very little space is saved when authors introduce new acronyms for phrases that are relatively short and which are used relatively infrequently. Writing the phrase in full for clarity whenever it is used makes reading more efficient and produces happier readers, who don't need to search back through dense text to fine the meaning. For example, is using "cop" for "cut-off period" really necessary?*
*I think the manuscript would benefit from a table of variables (and acronyms). If reader are going to need to frequently look up meanings of variables and acronyms, it would at least make it easier for them if there was one clearly identified place to go for that information.*

We fully agree to the reviewer that used acronyms should be well established when used in a paper. For that reason we consequently changed the acronym "cop" to "cut-off-period" in the whole manuscript. Yet, we keep the abbreviation since it is used in Fig. 01 and Table 1.

*Hyphens*
*The manuscript is nearly devoid of hyphens, although in many instances, correctly used hyphens would eliminate minor textual stumbling blocks that can slow readers' grasp of the material. For example – • Page 3, line 1 – "... argon-isotope-based temperature reconstructions."*
*Or better,*
*"... temperature reconstructions based on argon isotopes."*
*• Page 3, line 24 – "... Holocene-like data ...."*
*• Page 6, line 19 – "low-pass filtered:*
*• Page 8, line 30 – "first-guess temperature"*
*• Page 6, line 21 - "... accumulation-rate data ..."*
*• Page 16, line 14 – "gas-isotope-based temperature reconstructions ...",*
*or better, "temperature reconstructions based on gas isotopes ...",*

We changed all the points as it was suggested by the reviewer. Also we reformulate sentences and add hyphens in the text when it was necessary.

*Data are plural*
*• Page 6, line 21 - "... data are linearly interpolated ..."*
*• Page 6, line 24 - "... data were linearly indeed reconstructed ..."*
*• Page 8, line 2 – "... data are filtered ..."*
*• Page 17, line 5 – "... gas-isotope data are calculated ..."*
*• Page 18, line 11 – "... data are used to run ..."*

We corrected all the points as it was suggested by the reviewer. We also corrected the wrong grammar according to this in the rest of the manuscript.

*Page 1, line 26 –*
*"This is shown by high quality fitting of NGRIP δ15N data for two Dansgaard-Oeschger events using the presented approach, leading to results comparable to other studies."*
*"High quality" is wordy and can appear to be an attempt to prejudice how readers will view the results. It can often be more convincing to simply state that results were comparable, and let readers decide for themselves whether or not the fit was "high quality". If the authors want to retain "high-quality fitting", then it needs a hyphen.*

We agree that the used phrase "high quality fitting…" may bias the reader and influence the judgement of the presented details. We changed the formulation of the sentence (p.1 l.26) to:
"This is shown by fitting of NGRIP $\delta^{15}$N data for two Dansgaard-Oeschger events using the presented approach, leading to results comparable to other studies."

*Page 17, line 23 – The distance between GRIP and GISP3 is ~30 km, it is not "a few km". The authors are correct that more cores closer together could be very useful.*

We corrected for that and reformulated the sentence to:
"For example, a $\delta^{15}$N-based temperature reconstruction from the combination of data from the GISP2 ice core with the "sister ice core" GRIP drilled 30 kilometres apart is likely one of the best ways to overcome potential cm-scale variability."

**Clarity**
*• Page 1, line 15 – "… parameter tuning leading to …" I think there should be a comma after "tuning. Without a comma it is unclear whether the phrase "leading to reproducible temperature estimates" refers to the new automated approach or to the old manual approach. I think the former, but some readers may stumble on this point.*

We corrected for that and added the comma as suggested by the reviewer:
"The presented approach is completely automated and therefore minimizes the "subjective" impact of manual parameter-tuning, leading to reproducible temperature-estimates."

*• Page 1, line 16 –*
*"… other ice core based temperature reconstruction methods …".*
*Please try to avoid long strings of adjectives especially if you don't use hyphens to help readers understand the groupings. Better, with hyphens, "… other ice-core-based temperature-reconstruction methods …" Or better yet, unpacked, "… other temperature-reconstruction methods based on ice cores …".*

We changed the sentence according to the suggestion by the reviewer:
"In contrast to many other temperature-reconstruction methods based on ice cores, the presented approach is completely independent from ice-core stable-water-isotopes, providing the opportunity to validate water-isotope-based reconstructions or reconstructions where water isotopes are used together with $\delta^{15}$N or $\delta^{40}$Ar."

*• Page 2, line 3 - What is intended by "partly even centennial … variations"? This is not a English expression that I understand.*

We corrected for that and changed the sentence to:
"Yet, high-resolution studies are still very sparse and therefore limit the investigation of decadal and even centennial climate variations over the course of the Holocene."

*• Page 2, line 14 – There is no "Cuffey et al. (1997)" reference. The correct reference is "Cuffey and Clow (1997)".*

We corrected the wrong reference:
"The studies of Cuffey et al. (1995), Cuffey and Clow (1997) and Dahl-Jensen et al. (1998) demonstrate the usefulness of inverting the measured borehole-temperature profile for surface-temperature-history estimates for the investigated drilling site using a coupled heat- and ice-flow model."

*• Page 4 – "g is the acceleration constant." I think it is standard practice to mention "gravity" in this context.*

We changed the terminology here to:
"g is the gravitational acceleration,…"

*• Page 8, line 8 – "…which serve later on as …" Delete "on".*

We corrected for that and deleted the "on":
"In order to develop and evaluate the presented algorithm, eight temperature scenarios were constructed and used to model synthetic $\delta^{15}$N data, which serve later as targets for the reconstruction."

*• Page 8, line 31 – "... of about -29.6oC". Did you use -29.6oC, or did you use something different? Why not just say "... of -29.6°C"?*

We changed the sentence to:

"To construct the first-guess temperature-input $T_{g,0}(t)$, a constant temperature of -29.6 °C is used for the complete Holocene section, which corresponds to the last value of the temperature spin-up (Fig. 02b)."

***Tables***

*In Table 01, I don't see the point of saying that calculations ran over a weekend. Surely the days of the week are unimportant (are Saturdays really better than Wednesdays?). If the point is the execution time, then state "48 hours" or "N cpu cycles", or whatever is the appropriate number.*

We fully agree to the reviewer that the presented algorithm is independent from the exact date of use. We wanted to provide an explanation for the significant higher execution time for three runs. As it is not important for the description of the algorithm we deleted the line "Comments" of that table together with the statement, "3 runs were conducted over weekend, which leads to a higher number of iterations;", in the description.

We corrected for that:

[revised manuscript text omitted]

---

## Author Response (AR3)

**Reply to reviewer #1:**

We thank reviewer #1 for the report. Hereafter we address the questions and comments mentioned by the reviewer. Reviewer comments are given in *italic letters* whereas our replies are given in normal letters. Changes in the revised manuscript are given in bold letters.

*On page 4, line 18: add Orsi et al. 2014 after Kindler et al. 2014*

We added the reference and changed the sentence to:
**"A relatively new method for ice-core-based temperature reconstructions uses the thermal fractionation of stable isotopes of air compounds (nitrogen and argon) within a firn layer of an ice sheet (Huber et al., 2006; Kindler et al., 2014; Kobashi et al., 2011; Orsi et al., 2014; Severinghaus et al., 1998, 2001)."**

*On page 4, line 34: the statement is made that, except for the spin-up, the method is completely independent of water isotopes. If this is really the case, why is the spin-up needed at all? It seems to me that your statement would be more accurate if you said something like "the method is mostly independent of water isotopes." It would also be useful to add a graph in the Supplement showing the resulting temperature histories when a water isotope spin-up was, and was not, used. That would give the reader a feeling for how independent the method really is.*

The spin-up is needed to bring the firn state to a well-defined starting point for the optimization. This is necessary as it takes possible memory effects (influence of earlier conditions) of firn states into account as mentioned in the paper at page 10 ("Surface-temperature spin-up"). Without an adequate spin-up it would require a significant extension of allowed amplitudes and time window of adjustment which would lead to an immense increase in computation time. The effect of a "wrong" spin-up can last for several millennia for large differences from the real case and is dependent on the amplitudes of temperature changes in this section. As the synthetic temperature and $\delta^{15}N$ target-scenarios are created to mimic measured Holocene data, a spin-up that includes the large temperature excursions leading into the Holocene and creating partly the falling slope of the isotope data in the early Holocene, is needed. It is not important from which proxy the temperature spin-up is calculated, only the "correctness" of the temperature signal is important. Because of the argumentation above, it is not expedient to try to fit the gas-isotope data without using a spin-up. Also if we would use an unknown spin-up approach, we have to extend the time domain in which we allow the minimization algorithm to work and would need extended target data for this time.

*On page 5, line 8: The argument is made that because both 15N and 40Ar are used, there is no possibility to compare one solution against the other. This statement is rather misleading. In fact, the reason that both isotope pairs were used by Orsi et al. was to eliminate the gravitational signal from the data, which substantially reduces the uncertainty in the final derived temperature. When using only one isotope pair, the uncertainty is much larger, due to the need to match the gravitational signal accurately. Unfortunately, the gravitational signal is affected by many non-temperature processes, such as changes in the convective zone thickness, imperfections in the accumulation data accuracy, imperfections in the firn thickness model, and uncertainties in the true lock-in density. These additional uncertainties amount to 2-3 deg C of extra uncertainty in many real-world cases (for example, see Kobashi et al. 2008)*

*In summary, the use of only one isotope pair is highly inferior to the use of two; and saying that it is an advantage to be able to compare one to the other is like saying that "half of a bicycle is better than a whole bicycle because it is possible to compare the performance of the rider when riding only the front wheel, with the performance of the rider when riding only the back wheel."*

*This sentence must therefore be cut prior to publication. Instead, the authors could say something like "The method presented here can be used when no 40Ar data are available, which is often the case because 40Ar is a more analytically challenging measurement and is not as commonly measured as 15N."*

We changed the corresponding statement as follows based on the reviewer's suggestion:

Old version reads as follows:
"As both methods rely on $\delta^{15}N$ together with $\delta^{40}Ar$, they do not offer the possibility to validate one isotope-based solution against the other. Also these two approaches can only be applied to ice cores where both isotope quantities are measured together with a sufficient precision."

New version in the revised text reads now:

**"The method presented here can be used when no $\delta^{40}$Ar data are available, which is often the case because $\delta^{40}$Ar is a more analytically challenging measurement and is not as commonly measured as $\delta^{15}$N and further allows a comparison among solutions obtained from any of the available isotope quantities."**

*Page 13, line 7: the word "synchronicity" should be replaced with "synchroneity". Synchronicity is a term from Jungian psychology, and I do not think the authors intended this meaning.*

We corrected for this and changed the sentence to:

10 **"This is needed to insure synchroneity between the high-frequency temperature variations $\Delta T(t)$ extracted from the mismatch $D_{\delta 15N,mc,fin}(t)$ on the ice-age scale and the smooth temperature solution $T_{mc,fin}(t)$."**

*Page 29, line 4: The convective zone (CZ) does not include a non-diffusive zone. In fact there is molecular*
15 *diffusion occurring in the CZ; it is simply overwhelmed by mixing due to convection. I suggest you replace the words "non-diffusive zone" with "well-mixed zone".*

We changed the sentence according to the reviewer advice.
**"Many studies have shown the existence of a well-mixed zone at the top of the diffusive-firn-column, called**
20 **convective zone (CZ)."**

*The authors are thanked for their close attention to my prior comments.*

25 Thanks belong to the reviewer who helped to improve our manuscript.

[revised manuscript text omitted]

---

## Author Response (AR4)

**Reply to the PAGES Data Review Team comment:**

We thank the PAGES Data Review Team for the report. Hereafter we address the comments mentioned by the reviewers. Reviewer comments are given in italic letters whereas our replies are given in normal letters. Changes in the revised manuscript are given in bold letters.

*(1) Add a separate "Data Availability" section as required by the publisher. Specify where all of the essential input and output data are archived, including formal Data Citations for each of the datasets (see below). This includes the ice accumulation and oxygen isotope data.*

*(2) For essential datasets used in the study but not already in a public repository, submit the data and related metadata to an established public data repository and cite the persistent identifier in "Data Availability".*

*(3) Prior to publication of this study, submit the primary original data or results of numerical modeling to a public repository and cite the corresponding persistent identifier in "Data Availability". This includes the final time series of d15N and surface temperatures and any other data that might be useful for future users to replicate the study outcomes and to readily compare the results with future studies. We also strongly encourage the authors to deposit their significant code into a suitable repository and to cite it using a Data Citation.*

We added a separate "Data Availability" section at the end of the manuscript. This section contains all required information as suggested according to the reviewer advice:

**"Data availability**

**The synthetic $\delta^{15}$N and temperature targets, the reconstructed $\delta^{15}$N and temperature data (using the synthetic $\delta^{15}$N as fitting-targets), and the used accumulation rates can be found in the data supplement of this paper available at Döring, M.; Leuenberger, M. C. (2018), PANGAEA, https://doi.pangaea.de/10.1594/PANGAEA.888997. The GISP2 $\delta^{18}O_{ice}$ data used in this study for calculating the temperature spin-up can be found in Grootes and Stuiver (1999). The source code for the inversion algorithm and additional auxiliary data are available upon request."**

Additionally, we added the data reference for the $\delta^{18}O_{ice}$ data directly in the section "2.3 Measurement, input data and time scale" in subsection "$\delta^{18}O_{ice}$ data" (p. 10, l. 5-10):

"Oxygen-isotope data from the GISP2 ice-core-water samples measured at the University of Washington's Quaternary Isotope Laboratory are used to construct the surface-temperature input of the model spin-up (12 yr to 35 kyr b2k, Grootes et al., 1993; Grootes and Stuiver, 1997; Meese et al., 1994; Steig et al., 1994; Stuiver et al., 1995; **data availability: Grootes and Stuiver, 1999**). The raw $\delta^{18}O_{ice}$ data are filtered and interpolated in the same way as the accumulation-rate data for the spin-up part."

Furthermore, we included the following references to the "References" section of the manuscript:

Döring, M., Leuenberger, M. C.: Synthetic and fitted d15N and temperature data and GISP2 accumulation rates (13.5-52497.5 yr b2k) on GICC05 time scale. PANGAEA, https://doi.pangaea.de/10.1594/PANGAEA.888997, 2018.

[revised manuscript text omitted]
 ($\text{xcf}_{max}$) and a minimum ($\text{xcf}_{min}$) at two certain lags ($\text{lag}_{max,\delta15N}$ at $\text{xcf}_{max,\delta15N}$ and $\text{lag}_{min,\delta15N}$ at $\text{xcf}_{min,\delta15N}$). Now, the same analysis is conducted for IF(t) versus the temperature mismatch $D_{T,hf}(t)$ (Fig. 03b), which shows an equal behaviour (two extrema, $\text{lag}_{max,T}$ at $\text{xcf}_{max,T}$ and $\text{lag}_{min,T}$ at $\text{xcf}_{min,T}$). Comparing the two cross correlations show that $\text{lag}_{max,\delta15N}$ equals the negative $\text{lag}_{min,T}$ and $\text{lag}_{min,\delta15N}$ corresponds to the negative $\text{
[revised manuscript text omitted]